# The Benefits of Balance:
# From Information Projections to Variance Reduction

**Lang Liu**[*]    **Ronak Mehta**[*]    **Soumik Pal**    **Zaid Harchaoui**

University of Washington

## Abstract

Data balancing across multiple modalities and sources appears in various forms in foundation models in machine learning and AI, *e.g.* in CLIP and DINO. We show that data balancing across modalities and sources actually offers an unsuspected benefit: variance reduction. We present a non-asymptotic statistical bound that quantifies this variance reduction effect and relates it to the eigenvalue decay of Markov operators. Furthermore, we describe how various forms of data balancing in contrastive multimodal learning and self-supervised clustering can be better understood, and even improved upon, owing to our variance reduction viewpoint.

## 1 Introduction

Deep neural networks have shown remarkable success at learning task-specific representations of data when provided supervision from massive amounts of labeled training examples. Recent trends, however, have shifted toward task-agnostic, universal representations that may be easily fine-tuned or even have zero-shot capabilities out of the box. Supervised learning, *stricto sensu*, is too limited a framework for these billion-parameter, data-hungry models, and a question at the heart of modern machine learning is learning from unlabeled, partially labeled, or weakly labeled data.

This need has paved the way for the current generation of self-supervised learning (SSL) approaches that circumvent the need for large amounts of "strong" labels. In SSL, a model is trained on a generic pseudo-task suited for unlabeled data, such as relating image-caption pairs or augmentations of the same image. Despite modern foundation models such as DINO [Caron et al., 2021] and CLIP [Radford et al., 2021] being trained in this fashion, many aspects of SSL remain mysterious.

In particular, the training process of self-supervised models often transcends the rules of the standard empirical risk minimization (ERM) toolkit. ERM combines two well-understood techniques: minibatch sampling and gradient-based optimization using backpropagation. On the other hand, SSL adds clever, yet less-understood techniques to the training pipeline. To illustrate this, consider a minibatch of independent and identically distributed (i.i.d.) training examples $(X_1, Y_1), \ldots, (X_n, Y_n) \sim P$, where $P$ is a joint probability measure on sample spaces $\mathcal{X} \times \mathcal{Y}$ (e.g. feature-label or image-caption pairs) and let $P_n = \frac{1}{n} \sum_{i=1}^n \delta_{(X_i, Y_i)}$ be the empirical distribution. For a model parameterized by $\theta \in \mathbb{R}^d$ with loss function $h_\theta$, a stochastic learning algorithm involves computing the minibatch loss

$$\mathbb{E}_{P_n}[h_\theta(X, Y)] = \frac{1}{n} \sum_{i=1}^n h_\theta(X_i, Y_i) \tag{1}$$

and backpropagating through it to produce a minibatch stochastic gradient estimate. The algorithm then proceeds with the stochastic gradient descent (SGD) or a variant thereof (e.g., Adam, SGD with momentum, etc). Self-supervised methods often modify this recipe by *intervening* on the optimization algorithm in a minibatch-specific way.

---

[*]These authors contributed equally to this work.

38th Conference on Neural Information Processing Systems (NeurIPS 2024).

For example, SwaV [Caron et al., 2020] passes the minibatch examples through the model's encoder and clusters output vectors to generate pseudo-labels for a prediction task. In teacher-student architectures such as BYOL [Grill et al., 2020] and DINO [Caron et al., 2021], the minibatch is passed through two networks, where the "student" network is updated via backpropagation and the "teacher" network is updated by cloning the student's weights in regular intervals. In CLIP [Radford et al., 2021], a model optimizes the sum of two cross-entropy losses, where the predicted class probabilities on example $i$ are generated by comparison to all other elements of the minibatch. While introducing such interventions into the procedure has clearly proven useful practically, it remains conceptually unclear what exactly is being optimized by the learning algorithm.

In this work, we aim to gain a better theoretical understanding of the objectives and algorithms underlying these empirically effective recipes. In particular, we want to shed a theoretical light on their precise benefits over traditional learning methods. We show that such recipes often enjoy an unsuspected benefit: reducing the variance of the empirical minibatch objective.

Concretely, we formalize the model updates described above as two phrases. Let $Z_1, \ldots, Z_n$ be a minibatch containing data points of arbitrary type (e.g. unlabeled images). In the first phase, this original data source is mapped (possibly using a model parameterized by $\theta$) to another minibatch $(X_1, Y_1), \ldots, (X_n, Y_n)$ of *derived* pairs in $\mathcal{X} \times \mathcal{Y}$. For example, in SwaV, each $Z_i$ is an image, and we derive $(X_i, Y_i)$ by setting $X_i = Z_i$ and letting $Y_i$ be the pseudo-label based on clustering the vector representations of the images. In CLIP, each $Z_i$ is an image-caption pair, and we derive $(X_i, Y_i)$ by simply letting $X_i$ be the image and $Y_i$ be the caption. Note that $Y_i$ is *not* a label in the traditional sense in neither of these examples. In the second phase, we use the model to compute a probability distribution $P_{n,\theta}$ over $\mathcal{X} \times \mathcal{Y}$, and perform a stochastic gradient update for the objective

$$\mathbb{E}_{P_{n,\theta}} [h_\theta(X, Y)]. \tag{2}$$

This reduces to empirical risk minimization on the minibatch objective (1) when $Z = (X, Y)$ (each data point is originally observed in $\mathcal{X} \times \mathcal{Y}$) and $P_{n,\theta} = P_n$ (the empirical distribution of the data is used, regardless of the model). Beyond this setting, one specific example of $P_{n,\theta}$ has been applied across various families of self-supervised learning (as detailed in Sec. 2), which we refer to as *data balancing* or simply *balancing*, the primary subject of this work.

For a probability measure $Q$ on $\mathcal{X} \times \mathcal{Y}$, let $Q_X$ and $Q_Y$ be the respective marginals on $\mathcal{X}$ and $\mathcal{Y}$ and let $Q_{X|Y}$ and $Q_{Y|X}$ denote the respective conditional distributions. Given fixed *target* marginal distributions $P_X$ on $\mathcal{X}$ and $P_Y$ on $\mathcal{Y}$, balancing refers to repeatedly applying the operations

$$R \mapsto \underset{Q:Q_X=P_X}{\arg\min} \ \mathrm{KL}(Q\|R) \quad \text{and} \quad R \mapsto \underset{Q:Q_X=P_Y}{\arg\min} \ \mathrm{KL}(Q\|R), \tag{3}$$

in an alternating fashion. After enough iterations, the resulting probability measure approximately marginalizes to $P_X$ and $P_Y$ in each variable. When $\mathcal{X}$ and $\mathcal{Y}$ are finite with $|\mathcal{X}| = m$ and $|\mathcal{Y}| = l$, these operations reduce to rescaling the rows of an $(m \times l)$-matrix by $P_X/R_X$ or its columns by $P_Y/R_Y$. This algorithm has a decades-old history and is known in other contexts as Sinkhorn-Knopp matrix scaling [Sinkhorn, 1967], iterative proportional or biproportional fitting [Johnston and Pattie, 1993], and raking-ratio estimation [Thompson, 2000]. The marginals $P_X$ and $P_Y$ can represent auxiliary information or inductive bias from users, such as the desire for balanced clusters.

Returning to $P_{n,\theta}$ in (2), we show in Sec. 2 that both self-labeling and contrastive approaches in SSL implicitly define $P_{n,\theta}$ by the following steps: 1) constructing a method-specific "initial" measure $P_n^{(0)}$ on $\mathcal{X} \times \mathcal{Y}$, then 2) applying $k$ iterations of the operations (3) to generate a sequence $P_n^{(0)}, \ldots, P_n^{(k)}$, and finally, 3) setting $P_{n,\theta} := P_n^{(k)}$. In other words, these methods embed a *learnable* balancing operation in their objectives. A natural question to consider is: if the marginals one uses accurately represent the ones of the true probability measure $P$ governing the data, are balanced quantities "better behaved" than their unbalanced counterparts? If so, in what way?

Inspired by this question, we fix the model parameter $\theta$ (thus dropping the subscript from the quantities above) and analyze the fluctuations of the unbalanced and balanced objectives. The formal problem statement is as follows. Let $P_n^{(0)} = P_n$ and $P_n^{(k)}$ denote the output of $k \geq 1$ iterations of data balancing (see Sec. 3 for the precise definition). Finally, letting $h : \mathcal{X} \times \mathcal{Y} \to \mathbb{R}$ be a fixed function of interest, we define the population parameter $\varphi$ and its $k$-step *balanced estimator* $\varphi_n^{(k)}$ by

$$\varphi := \mathbb{E}_P [h(X, Y)] \quad \text{and} \quad \varphi_n^{(k)} := \mathbb{E}_{P_n^{(k)}} [h(X, Y)]. \tag{4}$$

Our goal is to establish theoretical guarantees on the mean squared error (MSE) $\mathbb{E}_P[(\varphi_n^{(k)} - \varphi)^2]$ of estimating $\varphi$ using $\varphi_n^{(k)}$, with an informative dependence on the sample size $n$, number of iterations $k$, target marginals $(P_X, P_Y)$, and test function $h$. We are particularly interested in its comparison to the direct estimator based on the empirical measure $\varphi_n^{(0)} = \frac{1}{n} \sum_{i=1}^n h(X_i, Y_i)$, as to quantify the effect of the auxiliary information $(P_X, P_Y)$. Our analysis uncovers two surprising facts. Firstly, while originally proposed for a different purpose, balancing reduces the variance of the empirical estimate. Secondly, while the balancing iterations are nonlinear operations on the input measure, the variance reduction can be precisely quantified using the spectral decay of two linear Markov operators: the conditional means given $X$ and $Y$, respectively.

**Contributions.** In Sec. 2, we detail the mathematical connection between balancing and the modern representation learning techniques mentioned above. In Sec. 3, we prove a new upper bound on the MSE of the balancing estimator $\varphi_n^{(k)}$. The bound decomposes into an $O(n^{-1})$ first-order variance term and an $O(k^6 n^{-3/2})$ second-order term. The first-order term is shown to have a strict improvement over the empirical measure baseline with a fine-grained dependence on the spectra of two particular Markov operators. The second-order term can be used to compute the asymptotic variance reduction for statistical efficiency comparisons. Our proof technique relies on a recursion decomposition for balancing-based estimators, which may be of independent interest. In Sec. 4, we illustrate how insights from our analysis can be practically applied to CLIP-type objectives and evaluation setups.

## 2 Data Balancing in Practice

To demonstrate a precise connection to (2), we describe how a collection of training examples $Z_1, \ldots, Z_n$ observed in an original data space $\mathcal{Z}$ (e.g. grayscale images) is mapped to a probability measure $P_{n,\theta}$. Using the framework introduced in Sec. 1, this amounts to specifying four components: 1) the map from the original data into the derived sample spaces $\mathcal{X}$ and $\mathcal{Y}$, 2) the initial measure $P_n^{(0)}$, 3) the function $h$, and 4) the target marginals $(P_X, P_Y)$ for this measure to fit. From that point, the iterations of (3) produce $P_n^{(1)}, \ldots, P_n^{(k)}$, and we set $P_{n,\theta} := P_n^{(k)}$. For ease of presentation, we hide the dependence of $P_n^{(k)} = P_{n,\theta}$ and $h \equiv h_\theta$ on the model parameter $\theta$. See Fig. 1 for examples of different choices of the sample spaces $\mathcal{X}$ and $\mathcal{Y}$.

**Example 1: Self-Supervised Clustering.** Balancing is used in discriminative clustering and self-supervised clustering; see [Jones et al., 2022, Asano et al., 2020, Caron et al., 2020] for variations on this theme. We describe the swapped prediction task of Caron et al. [2020] for concreteness but emphasize that clustering of this form is used as an intermediate step (or as the task itself) in many SSL pseudo-tasks. At a high level, this approach involves passing elements of a minibatch through two encoders to generate vector representations. These representations are then clustered separately, and the features from one encoder predict the cluster label from the other encoders. Denote the encoders $f_{\theta_s} : \mathcal{Z} \to \mathbb{R}^r$ and $f_{\theta_t} : \mathcal{Z} \to \mathbb{R}^r$, colloquially known as the *student* and *teacher* networks, respectively. Here, we let $\{Z_i\}_{i=1}^n$ be a minibatch of $n$ images, with

$$\mathcal{X} = \{Z_1, \ldots, Z_n\} \quad \text{and} \quad \mathcal{Y} = \{1, \ldots, l\},$$

where $m = n$ and the elements of $\mathcal{Y}$ index learnable cluster representation vectors $c_1, \ldots, c_l \in \mathbb{R}^r$. Thus, we consider the overall parameter vector to be $\theta := (\theta_s, \theta_t, c_1, \ldots, c_l)$. Given temperature hyperparameters $\epsilon, \tau > 0$, the initial measure and loss function are given by the expressions

$$P_n^{(0)}(x, y) \propto e^{f_{\theta_s}(x)^\top c_y / \epsilon} \quad \text{and} \quad h(x, y) = \log \frac{e^{f_{\theta_t}(x)^\top c_y / \tau}}{\sum_{y'=1}^l e^{f_{\theta_t}(x)^\top c_{y'} / \tau}}.$$

Directly optimizing $\sum_{x,y} P_n^{(0)}(x, y) h(x, y)$ without any constraints would lead to collapse, so it is balanced before optimization. The target marginals $P_X$ and $P_Y$ are given by the discrete uniform measures on $\mathcal{X}$ and $\mathcal{Y}$. This formulation is often derived by solving an optimal transport problem with the Sinkhorn-Knopp algorithm to assign soft cluster labels, the iterative solution result from this procedure is precisely $P_n^{(k)}$. The intuition behind the choice of uniform marginal $P_X$ is that each data point has an equal amount of mass to be allotted, whereas $P_Y$ captures that the cluster sizes are equal. The number of iterations $k$ is selected based on optimization considerations.

**Example 2: Contrastive Learning.** Contrastive Language-Image Pre-Training [Radford et al., 2021], or CLIP, is an architecture with an image encoder and a text encoder that map to a joint

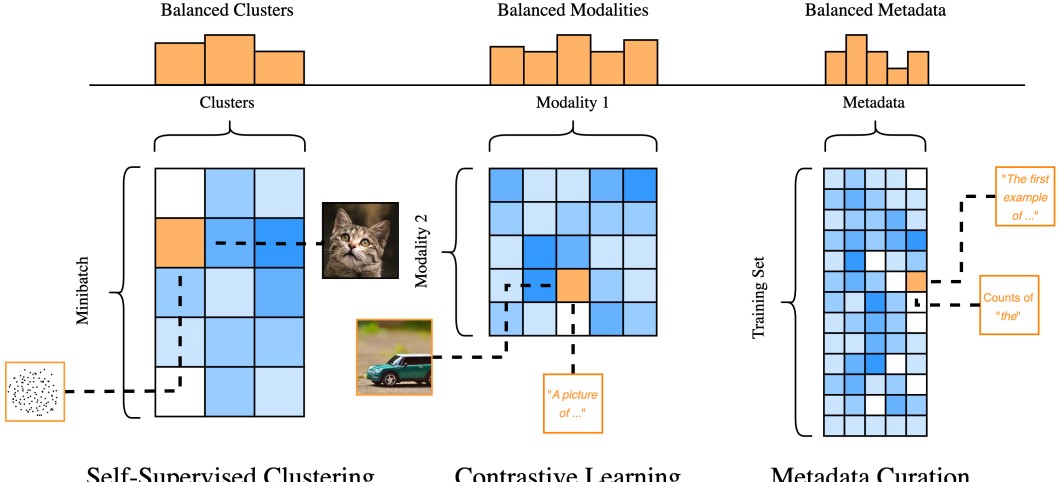

Figure 1: **Data Balancing Examples:** Each panel shows a possible distribution $Q$ on different choices of $(\mathcal{X}, \mathcal{Y})$. The orange histograms are the target marginal $P_Y$. **Left:** $Q(x, y)$ is the affinity of an image $x$ for cluster $y$. **Center:** $Q(x, y)$ is the similarity of an image $x$ to a text caption $y$. **Right:** $Q(x, y)$ is the proportion of substring matches between a text caption $x$ and a keyword $y$.

embedding space. Trained using image-caption pairs, the loss promotes representations such that images and text that are paired in the minibatch are close, whereas those that are not paired are far. The latter aspect (promoting dissimilarity of unpaired images/text) is what prevents collapse in this framework. To our knowledge, our interpretation of the CLIP objective as an implicit data balancing procedure is novel. Under this interpretation, we demonstrate that the objective is in fact a nonlinear function of $P_{n,\theta}$, whereas its gradient will have a linear form similar to (2). In this case, each $Z_i = (X_i, Y_i)$, where $X_i$ is an image and $Y_i$ is an associated caption. We have that

$$\mathcal{X} = \{X_1, \ldots, X_n\} \quad \text{and} \quad \mathcal{Y} = \{Y_1, \ldots, Y_n\},$$

so that $m = n$. Consider an image encoder $f_{\theta_I} : \mathcal{X} \mapsto \mathbb{R}^r$ and text encoder $f_{\theta_T} : \mathcal{Y} \mapsto \mathbb{R}^r$ with parameter vector $\theta = (\theta_I, \theta_T)$. The initial, unnormalized measure and the (in this case, vector-valued) function $h$ are chosen based on these encoded representations:

$$P_n^{(0)}(x, y) \propto e^{f_{\theta_I}(x)^\top f_{\theta_T}(y)} \quad \text{and} \quad h(x, y) = \nabla_\theta (f_{\theta_I}(x)^\top f_{\theta_T}(y)). \tag{5}$$

While we usually interpret $h$ as a loss function, we will show below that the CLIP loss depends nonlinearly on $P_{n,\theta}$, while the gradient has a linear dependence. If we believe, as in Example 1, that the target marginals $(P_X, P_Y)$ of the images and the text should be roughly uniform, we can apply the balancing iterations (3) with the target marginals being the uniform distributions over $\mathcal{X}$ and $\mathcal{Y}$, respectively. Because there is no preference for starting the iterations with the $\mathcal{X}$ or $\mathcal{Y}$ dimension first, we may consider both orderings. Let $Q_n^{(1)}$ be one iteration of balancing in the $\mathcal{Y}$ dimension and $R_n^{(1)}$ represent one such iteration in the $\mathcal{X}$ dimension. Then the original CLIP objective $L_n^{\text{CLIP}}$ can be recovered (up to an additive constant) as

$$L_n^{\text{CLIP}} := -\frac{1}{2} \sum_{i=1}^n \left[ \log \frac{P_n^{(0)}(X_i, Y_i)}{\sum_x P_n^{(0)}(x, Y_i)} + \log \frac{P_n^{(0)}(X_i, Y_i)}{\sum_y P_n^{(0)}(X_i, y)} \right]$$

$$= -\frac{1}{2} \sum_{i=1}^n [\log Q_n^{(1)}(X_i, Y_i) + \log R_n^{(1)}(X_i, Y_i)] - \log n. \tag{6}$$

The measure $P_{n,\theta} = \frac{1}{2} Q_n^{(1)} + \frac{1}{2} R_n^{(1)}$ is constructed in this case by averaging the outputs of one iteration of balancing under each modality. Taking the gradient of (6) with respect to $\theta$ (whose dependence is contained in $(Q_n^{(1)}, R_n^{(1)})$) recovers the expression for $h$ in (5). The objective is often interpreted as an average of cross-entropy loss terms, each representing the prediction of one modality's original pair from the other. In our formulation, $L_n^{\text{CLIP}}$ can also be viewed as an average negative log-likelihood under the $Q_n^{(1)}$ and $R_n^{(1)}$. It is also of interest to study the effect of using $Q_n^{(k)}$ and $R_n^{(k)}$ for $k \geq 0$ in general, as we show in Sec. 4.

**Example 3: Metadata Curation.** Here, we consider balancing an entire training set, as opposed to a particular minibatch. At the billion-parameter scale, dataset design can be the primary factor that differentiates performance between foundation models [Fang et al., 2013, Xu et al., 2024, Gadre et al., 2023]. One general approach used in both the original CLIP dataset [Radford et al., 2021] and an open-source replication [Xu et al., 2024] is metadata curation, wherein a text dataset (possibly captions for images) is synthesized using a list of keywords $\{y_1, \ldots, y_l\}$ so that

$$\mathcal{X} = \{Z_1, \ldots, Z_n\}, \quad \mathcal{Y} = \{y_1, \ldots, y_l\},$$

meaning that $m = n$. The keywords are matched to texts within $\mathcal{X}$ via substring matching. While the approach of Xu et al. [2024] (dubbed MetaCLIP) pools all matched keywords on every text to measure the "distribution" of keywords, we consider a version in which each text $Z_i$ can only be labeled with a single keyword $y_j$. This allows for a true joint probability measure on $\mathcal{X} \times \mathcal{Y}$. The marginal distribution of observed keywords is initially long-tailed (see Fig. 4) (e.g., "the" will match many more texts than "xylophone"). In both Radford et al. [2021] and Xu et al. [2024], the data are resampled so that this distribution of keywords over matches is closer to uniformity, i.e. keywords with many matches have their associated texts downsampled during the dataset creation process. While the probability measure may not be computed explicitly (due to scale), this adjustment of the keyword distribution can be viewed as a single iteration of balancing (3) applied to the $\mathcal{Y}$ marginal. For tasks such as language modeling, we have

$$P_n^{(0)}(x, y) = P_n(x, y) \quad \text{and} \quad h(x, y) = \ell_\theta(x), \tag{7}$$

where $\ell_\theta(x)$ denotes the loss of a model evaluated at a single text $x \in \mathcal{X}$ (notice that the keyword is not used). We elucidate this connection by applying direct balancing on a subset of the ImageNet-Captions dataset in Sec. 4, observing the effect on downstream model performance.

Motivated by these scenarios, we address the statistical problem outlined in Sec. 1 by analyzing balancing-based estimators. We then return to examples mentioned above in Sec. 4, illustrating how the theoretical analysis can be translated to algorithmic variants.

## 3 Theoretical Analysis of Variance Reduction

We now present theoretical guarantees on the mean squared error (MSE) of the data-balanced estimator $\varphi_n^{(k)}$ and highlight relevant points in the proofs. For readers' convenience, a notation table (Tab. 1) is in Appx. A. We first give context on the main innovations of the analysis and then outline its high-level steps. These innovations include relating the nonlinear iterations of balancing over probability measures to linear operators on a vector space and using a singular value decomposition of these operators to quantify their effect after a finite number of iterations. Furthermore, by scaling the number of iterations appropriately, we can characterize the estimator using the limit of balancing iterations, which is an object of interest in applications including optimal transport.

**Preliminaries.** Recall the setting introduced in Sec. 1, in which we consider sample spaces $(\mathcal{X}, \mathcal{Y})$, along with true and unknown joint distribution $P$ on $\mathcal{X} \times \mathcal{Y}$ with known marginals $(P_X, P_Y)$. For ease of presentation, we assume that $|\mathcal{X}| = |\mathcal{Y}| = m$, although the arguments do not rely on equal support sizes. We make the following assumption throughout, which is usually satisfied by the desired marginals $P_X$ and $P_Y$, such as in the uniform cases discussed in Sec. 2: the target marginals $P_X(x) > 0$ and $P_Y(y) > 0$ for all $x \in \mathcal{X}$ and $y \in \mathcal{Y}$. We define $P_n^{(0)} = P_n$ as the empirical measure and for $k \geq 1$ construct

$$P_n^{(k)}(x, y) := \begin{cases} \frac{P_X(x)}{P_{n,X}^{(k-1)}(x)} \cdot P_n^{(k-1)}(x, y) & k \text{ odd} \\ \frac{P_Y(y)}{P_{n,Y}^{(k-1)}(y)} \cdot P_n^{(k-1)}(x, y) & k \text{ even} \end{cases}. \tag{8}$$

By direct computation, we see that the iterations in (8) are equivalent to applying (3) for $k$ odd and even, respectively. See Fig. 2 (left) for a visualization of this procedure. The iterations are well-defined for all $k$ under the event that $\mathrm{Supp}(P_{n,X}) = \mathrm{Supp}(P_X)$ and $\mathrm{Supp}(P_{n,Y}) = \mathrm{Supp}(P_Y)$, i.e., all observed row counts and column counts are non-empty.[2]

---

[2]Due to this technical consideration, we define $P_n^{(k)}$ to be the empirical measure $P_n$ when this condition is not satisfied, which we show occurs with low-probability. See Appx. D.4 for details.

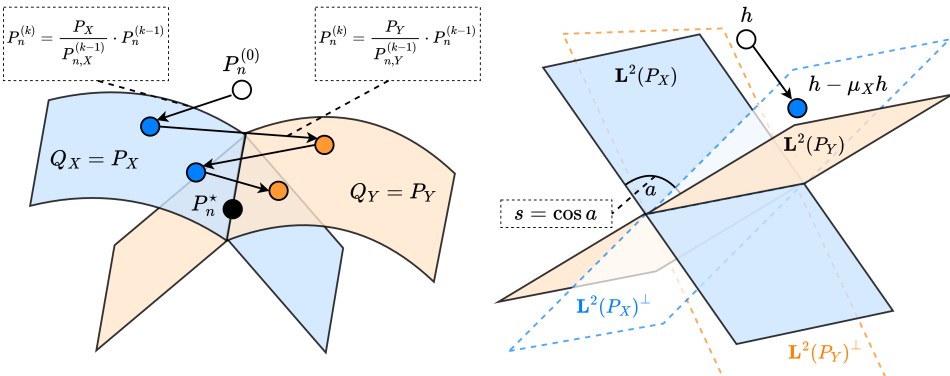

Figure 2: **Data Balancing.** Nonlinear and linear operators associated with each iteration of (8). **Left:** Visualization of the exact iterations of (8) in the space of probability measures. The blue set contains joint distributions with $\mathcal{X}$-marginal equal to $P_X$, whereas the orange set contains joint distributions with $\mathcal{Y}$-marginal equal to $P_Y$. **Right:** Visualization of $\mathbf{L}^2(P)$, the operators defining (11), and the singular values given in (13).

To provide background, the scheme of alternating the operators (8) is often seen as an iterative algorithm to solve the problem

$$\min_{Q \in \Pi(P_X, P_Y)} \mathrm{KL}(Q \| P_n^{(0)}), \tag{9}$$

where $\Pi(P_X, P_Y)$ denotes the set of probability measures on $\mathcal{X} \times \mathcal{Y}$ that marginalize to $P_X$ and $P_Y$ in each variable and $\mathrm{KL}(\cdot\|\cdot)$ denotes the Kullback-Leibler divergence. The iterations (8) are based on the alternating minimization approach of solving

$$P_n^{(k)}(x,y) := \begin{cases} \arg\min_{\{Q:Q_X=P_X\}} \mathrm{KL}(Q \| P_n^{(k-1)}) & k \text{ odd} \\ \arg\min_{\{Q:Q_Y=P_Y\}} \mathrm{KL}(Q \| P_n^{(k-1)}) & k \text{ even} \end{cases},$$

which inspires the viewpoint of balancing as alternating *information projections*. As we show in Appx. C, the iterations of (8) can equivalently be defined using the KL, reverse KL, or $\chi^2$-divergences. This viewpoint is relevant as previously, efforts have been made (e.g. in Bickel et al. [1991]) to analyze the variance reduction afforded by the solution to (9) directly. However, quantifying the variance reduction (in terms of properties of $P$) using this approach is challenging, as there is no closed-form expression for the solution of (9). A key mathematical outcome of our analysis is that the closed-form expressions of the projections (8) can be used to compute the reduction in mean squared error at each iteration. Thus, by letting $k \equiv k(n) \to \infty$ (scaled appropriately against $n$), we can determine the reduction for the solution of (9) for large $n$. This is the subject of Thm. 1.

**From Information Projections to Orthogonal Projections.** First, we will show that the variance reduction resulting from each nonlinear iteration of (8) is associated with a linear operator applied to $h$. Thus, instead of analyzing the alternating information projections over probability measures, we may use familiar tools to understand alternating orthogonal projections in a vector space. To define them, we first let $\mathbf{L}^2(P)$ to be the set of functions $h : \mathcal{X} \times \mathcal{Y} \to \mathbb{R}$ satisfying $\mathbb{E}_P\left[h^2(X,Y)\right] < \infty$. Even though $\mathcal{X} \times \mathcal{Y}$ is finite, working within $\mathbf{L}^2(P)$ will be analytically convenient. Let $\mathbf{L}^2(P_X)$ be the subspace of $\mathbf{L}^2(P)$ containing functions that only depend on the first argument $x \in \mathcal{X}$ and define $\mathbf{L}^2(P_Y)$ analogously. These are the solid-colored subspaces in Fig. 2 (right). Next, let $\mu_X : \mathbf{L}^2(P) \to \mathbf{L}^2(P_X)$ and $\mu_Y : \mathbf{L}^2(P) \to \mathbf{L}^2(P_Y)$ be defined as, for any $h \in \mathbf{L}^2(P)$,

$$\mu_X h = \arg\min_{f \in \mathbf{L}^2(P_X)} \mathbb{E}_P\left[(h(X,Y) - f(X))^2\right] \implies [\mu_X h](x,y) := \mathbb{E}_P\left[h(X,Y)|X\right](x)$$

The operator $\mu_X$ is an orthogonal projection onto $\mathbf{L}^2(P_X)$. The orthogonal projection operator $\mu_Y$ onto $\mathbf{L}^2(P_Y)$ is defined analogously. We may also define the conditional *debiasing* operators $\mathcal{C}_X = I - \mu_X$ and $\mathcal{C}_Y = I - \mu_Y$, which each project onto the orthogonal complements of $\mathbf{L}^2(P_X)$ and

$\mathbf{L}^2(P_Y)$, visualized as subspaces with dotted border in Fig. 2 (right). To understand the importance of the conditional mean and debiasing operators, we give a recursive formula that forms the backbone of our analysis. Define $\mu_k = \mu_X$ for $k$ odd and $\mu_k = \mu_Y$ for $k$ even, and define $\mathcal{C}_k$ similarly. Thus, by using the notation $Q(h) := \mathbb{E}_Q[h(X, Y)]$, we have by linearity of expectation that

$$
\begin{aligned}
[P_n^{(k)} - P](h) &= [P_n^{(k)} - P](\mathcal{C}_k h) + \overbrace{[P_n^{(k)} - P](\mu_k h)}^{=0} \\
&= [P_n^{(k-1)} - P](\mathcal{C}_k h) + [P_n^{(k)} - P_n^{(k-1)}](\mathcal{C}_k h) \\
&= \underbrace{[P_n^{(0)} - P](\mathcal{C}_1 \ldots \mathcal{C}_k h)}_{\text{first-order term}} + \underbrace{\sum_{\ell=1}^{k} [P_n^{(\ell)} - P_n^{(\ell-1)}](\mathcal{C}_\ell \ldots \mathcal{C}_k h)}_{\text{higher-order terms}}.
\end{aligned}
\tag{10}
$$

To justify the first line, we discuss the case when $k$ is odd. Notice that $\mu_X h$ is only a function of $X$, so its expectation only depends on $P_X$ that is equal to $P_{n,X}^{(k)}$ (the $\mathcal{X}$-marginal of $P_n^{(k)}$) by (8). The last line follows by unrolling the previous step $k - 1$ times. This recursive expansion is proven formally in Prop. 15 in Appx. D. Given the expansion, the mean squared error can be computed by taking the expectation of squared (10). We show that the second moment of the first-order term in (10) is equal to $\sigma_k^2/n$ where

$$
\sigma_0^2 := \mathbb{V}\mathrm{ar}(h) \text{ and } \sigma_k^2 := \mathbb{V}\mathrm{ar}(\mathcal{C}_1 \ldots \mathcal{C}_k h) \text{ for } k \geq 1,
\tag{11}
$$

and all other terms are $O(k^6 n^{-3/2})$. Thus, by exactly computing the constant in the dominating term, we may quantify the asymptotic variance reduction. Our first main result concerns the higher-order terms and shows that it is indeed dominated by the first-order term. Note that the empirical mean $\varphi_n^{(0)} = \frac{1}{n}\sum_{i=1}^{n} h(X_i, Y_i)$ is unbiased, and so its MSE is equal to $\sigma_0^2/n$. Define in addition

$$
p_\star := \min\{\min_x P_X(x), \min_y P_Y(y)\}
$$

which measures the non-uniformity of the target marginals. We have that $p_\star$ is positive because both $P_X$ and $P_Y$ are positive. We now state the first main result.

**Theorem 1.** *For a sequence of data balancing estimators $(\varphi_n^{(k)})_{k \geq 1}$ as defined in (4), there exists an absolute constant $C > 0$ and distribution dependent constant $s \in [0, 1)$ and such the following holds for $\sigma_{gap}^2 = \sigma_0^2 - \sigma_k^2$: For $n \geq C[\log_2(2n/p_\star) + m\log(n+1)]/p_\star^2$ and $k \geq 1$, we have*

$$
\mathbb{E}_P\left[(\varphi_n^{(k)} - \varphi)^2\right] \leq \frac{\sigma_0^2 - \sigma_{gap}^2}{n} + O\left(\frac{s^k}{n}\right) + \tilde{O}\left(\frac{k^6}{n^{3/2}}\right).
\tag{12}
$$

The quantities $\sigma_{\text{gap}}^2$ and $s$ are quantified toward the end of this section and are dependent on eigendecays of the conditional mean operators for each variable under $P$. Furthermore, $\sigma_{\text{gap}}^2 > 0$ except for the pathological case of $\mu_X h$ being a constant function. Showing Thm. 1 boils down to showing that the higher-order term in (10) is $O(n^{-1})$ with high probability. Using the expression (8) and assuming that $\ell \geq 1$ is odd, we see that

$$
[P_n^{(\ell)} - P_n^{(\ell-1)}](\mathcal{C}_\ell \ldots \mathcal{C}_k h) = \sum_{x,y}\left[\frac{P_X(x)}{P_{n,X}^{(\ell-1)}(x)} - 1\right] \cdot [\mathcal{C}_\ell \ldots \mathcal{C}_k h](x, y) P_n^{(\ell-1)}(x, y).
$$

The first (blue) term in the product quantifies the disagreement between the $\mathcal{X}$-marginal of $P_n^{(\ell-1)}$ and the true marginal, which can be bounded in terms of $\mathrm{KL}(P_{n,X}^{(0)} \| P_X)$ and is shown to be $O(n^{-1/2})$ with high probability via techniques from information theory. The second (orange) term can be unrolled recursively in a similar fashion to (10) itself, which will consequently be $O(n^{-1/2})$ as well; this is the most technical part of the analysis (see Appx. D.3). Our analysis also yields a bound for the sensitivity of balancing to misspecified marginals; see Appx. D.5.

Given Thm. 1, a natural next step is to quantify the gap between $\sigma_0^2$ and $\sigma_k^2$, which requires finer-grained properties of $\mathcal{C}_X$ and $\mathcal{C}_Y$. Notably, we show that as $k \to \infty$, $\sigma_k^2$ approaches a limiting value. Thus, via (12), by using $k = o(n^{1/12})$ obtains asymptotic variance of the solution to (9). This contrasts with Albertus and Berthet [2019], in which the dependence of a quantity similar to (12) is exponential in $k$, meaning that $k = o(\log(n))$ is required for convergence under this argument.

**From Orthogonal Projections to Variance Reduction.** We now clarify what is precisely meant by the "spectrum" of the conditional mean operators $\mu_X$ and $\mu_Y$. As proven using a *singular value decomposition* (Prop. 3) in Appx. B.1, there exists a basis $\{\alpha_j\}_{j=1}^m$ of $\mathbf{L}^2(P_X)$, a basis $\{\beta_j\}_{j=1}^m$ of $\mathbf{L}^2(P_Y)$, and real values $\{s_j\}_{j=1}^m$, that satisfy

$$\mu_Y \alpha_j = s_j \beta_j \text{ and } \mu_X \beta_j = s_j \alpha_j \text{ for } j \in \{1, \ldots, m\}. \tag{13}$$

Furthermore, $\alpha_1 = \mathbf{1}_{\mathcal{X}}$ and $\beta_1 = \mathbf{1}_{\mathcal{Y}}$ leading to the equality $\langle f, \alpha_1 \rangle_{\mathbf{L}^2(P_X)} = \mathbb{E}_{P_X}[f(X)]$. Finally, $s_1 = 1$ and $s_j$ is non-negative and non-increasing in $j$. For a concrete example, consider $m = 2$, in which case $P$ can be written as a matrix in $\mathbb{R}^{2 \times 2}$ and elements of $\mathbf{L}^2(P_X)$ and $\mathbf{L}^2(P_X)$ are vectors in $\mathbb{R}^2$. Then, in the case of uniform marginals, we can verify directly that (13) can be satisfied by setting

$$\alpha_1 = \beta_1 = \begin{bmatrix} 1 \\ 1 \end{bmatrix}, \alpha_2 = \beta_2 = \begin{bmatrix} 1 \\ -1 \end{bmatrix}, \text{ and } P = \frac{1}{4}\begin{bmatrix} 1+s & 1-s \\ 1-s & 1+s \end{bmatrix} \tag{14}$$

for $s = s_2$ (the second largest singular value). Thus, as $s \to 1$, the distribution becomes "fully dependent" as $Y$ and $X$ are completely determined by one another. As $s \to 0$, $P$ approaches the product measure. Geometrically, because $\alpha_1 = \beta_1$, we know that the angle $a$ between the subspaces $\mathbf{L}^2(P_X)$ and $\mathbf{L}^2(P_Y)$ is given by the angle between $\alpha_2$ and $\beta_2$. By computing their inner product in $\mathbf{L}^2(P)$, we have that $\langle \alpha_2, \beta_2 \rangle_{\mathbf{L}^2(P)} = \langle P, \alpha_2 \beta_2^\top \rangle = s = \cos a$. Thus, $s = 0$ indicates orthogonality of these subspaces, alluding to the independence of $X$ and $Y$ (see the right panel of Fig. 2).

Returning to $m \geq 2$, we consider the following as a sufficient condition for variance reduction: the operators $\mu_X$ and $\mu_Y$ have a positive spectral gap, i.e., $s_2 < s_1$. Note that this assumption is satisfied when $P(x, y) > 0$ for all $(x, y) \in \mathcal{X} \times \mathcal{Y}$ by the Perron–Frobenius Theorem [Horn and Johnson, 2013, Chapter 8]. Using the intuition from Fig. 2, this rules out pathological cases such as $Y$ being a deterministic function of $X$. Under the spectral gap condition, the singular values $\{s_j\}_{j=2}^m$ that are strictly less than 1 will determine a geometric rate of decay in variance given in Cor. 2. The left and right singular functions $\alpha_j : \mathcal{X} \to \mathbb{R}$ and $\beta_j : \mathcal{Y} \to \mathbb{R}$ will define a useful coordinate system to represent projections of $h$ when analyzing $\varphi_n^{(k)}$.

Indeed, let $\bar{h} = P(h)$ be the centered test function. Because $\mu_X \bar{h} \in \mathbf{L}^2(P_X)$ and $\mu_Y \bar{h} \in \mathbf{L}^2(P_Y)$, we may decompose this function on the two bases to write

$$\mu_X \bar{h} = \sum_{j=1}^m u_j \alpha_j \quad \text{and} \quad \mu_Y \bar{h} = \sum_{j=1}^m v_j \beta_j. \tag{15}$$

Cor. 2 below relates the (normalized) variance $\sigma_k^2$ of the first-order term to the one of the sample mean $\varphi_n^{(0)}$. In fact, it shows that the variance reduction $\sigma_0^2 - \sigma_k^2$ decays geometrically to the quantity

$$\sigma_{\text{gap}}^2 := \sum_{j=2}^m \left[ u_j^2 + \frac{(v_j - s_j u_j)^2}{1 - s_j^2} \right].$$

For simplicity, we only present the result for $k$ even, i.e., $\sigma_{2t}^2$.

**Corollary 2.** *The variance reduction achieved by $t + 1$ iterations of the $\mathcal{C}_Y \mathcal{C}_X$ operator can be quantified as*

$$\sigma_0^2 - \sigma_{2(t+1)}^2 = \sigma_{gap}^2 - \sum_{j=2}^m \frac{s_j^2 (v_j - s_j u_j)^2}{1 - s_j^2} s_j^{4t} = \sum_{j=2}^m \left[ u_j^2 + (1 - s_j^{4t+2}) \frac{(v_j - s_j u_j)^2}{1 - s_j^2} \right].$$

Intuitively, the operators $\mathcal{C}_X$ and $\mathcal{C}_Y$ are the main sources of the variance reduction via orthogonality. Since $\alpha_1 = \mathbf{1}_{\mathcal{X}}$, we can see that the reduction will always be strictly positive as long as $\mu_X \bar{h}$ is not a constant function. Finally, using $s := s_2 \geq s_j$ for $j \geq 2$ gives the second term in Thm. 1.

## 4 Numerical Illustrations

We illustrate how data balancing manifests in the motivating examples mentioned in Sec. 2 with experiments with CLIP-type models. We focus here on zero-shot image classification tasks. Details on these experiments, and additional ones including linear probing and zero-shot retrieval, as well as an empirical investigation of the sensitivity to misspecified marginals, are all contained in Appx. E. Code to reproduce the data and experiments can be found at https://github.com/ronakdm/balancing.

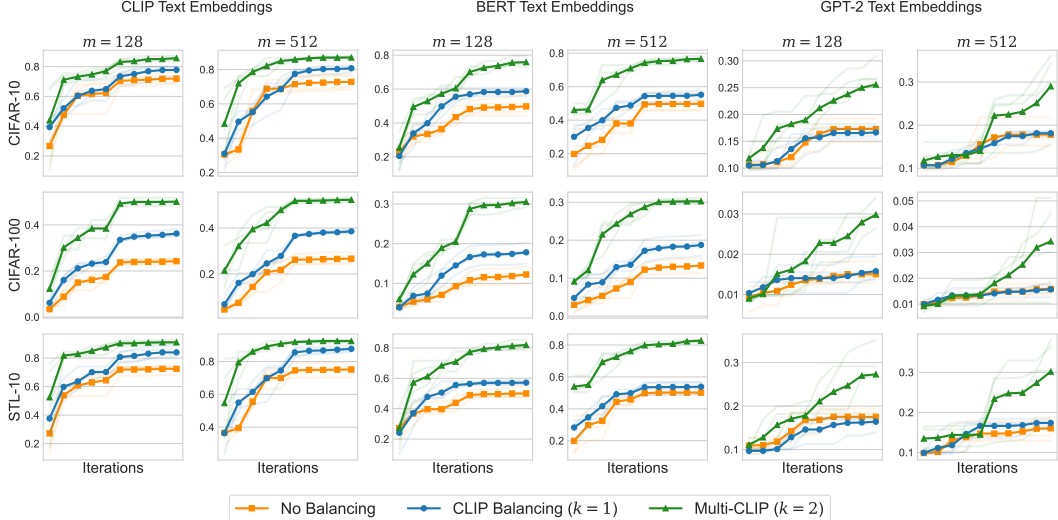

Figure 3: **Zero-Shot Classification Performance across Embeddings, Batch Sizes, and Objectives.** The three vertical panels describe different choices of the text encoder $f_{\theta_T}$ which increases in quality from left to right; that is, pre-trained GPT-2, BERT, and CLIP embeddings, respectively. Within each vertical panel, examples include batch sizes $m = 128$ and $m = 512$. Rows indicate various evaluation datasets from CIFAR-10, CIFAR-100, and STL-10. The $y$-axis of each plot indicates average per-class recall, whereas the $x$-axis indicates training iterations at the given batch size.

**Model, Datasets, and Evaluation.** Throughout, we consider training variants of CLIP models (see Sec. 2), which require a dataset of image-caption pairs. For the training set, we use the ImageNet-Captions dataset [Fang et al., 2013], which pairs images from ImageNet [Deng et al., 2009] that were taken from Flickr with their original captions. In the notation of Sec. 2, the model is specified by selecting an image encoder $f_{\theta_I}$ and a text encoder $f_{\theta_T}$. In all cases, we use a fixed image/text encoder as a base vector representation and compose it with a trainable feed-forward neural network, i.e., $f_\theta = f_\theta^{\text{head}} \circ f^{\text{base}}$. We fix the base image encoder as CLIP ViT-B/32 architecture pre-trained on LAION-2B [Schuhmann et al., 2022], and vary the base text encoder across embedding models of varying quality: GPT-2 [Radford et al., 2019], BERT [Devlin et al., 2019], and CLIP-based encodings. When two CLIP encoders are used for the base image/text vector representation, they are taken from separate CLIP models (i.e. the base representations are not dependent). We evaluate models based on zero-shot classification performance using the standard CLIP inference procedure: for any image $x$, a label $c \in \{1, \ldots, C\}$ is predicted by associating to each $c$ a natural language prompt $y_c$, and predicting the scores $s(x) = (s_1(x), \ldots, s_C(x))$, with

$$s_c(x) = \frac{e^{\left\langle f_{\theta_I}(x), f_{\theta_T}(y_c) \right\rangle / \tau}}{\sum_{c'=1}^{C} e^{\left\langle f_{\theta_I}(x), f_{\theta_T}(y_{c'}) \right\rangle / \tau}} \tag{16}$$

for a temperature $\tau > 0$. Multiple prompting strategies can be used depending on the evaluation dataset, for which we average embeddings before applying (16). We use the public CLIP Benchmark repository, using the datasets CIFAR-10, CIFAR-100, STL-10, with their default caption sets.

**Data Balancing Effects.** Fig. 3 shows the zero-shot classification performance (in terms of average per-class recall) of variants depending on whether the *contrastive learning* objective from Sec. 2 is used or not. One iteration of balancing already leads to improvement in terms of downstream performance. Multiple balancing iterations lead to further improvements. See Appx. E for more details on this experiment, and for analogous ones with linear probing and zero-shot retrieval.

Fig. 4 then shows how balancing can be used to adjust an entire pre-training set to given marginals based on metadata, as described in Sec. 2 in the *metadata curation* example. After balancing, the target marginal has less than 2 orders of difference. In terms of downstream performance, data balancing leads to some improvement in the smaller batch regime ($m = 512$) when curating the dataset. See Appx. E for more details on this experiment.

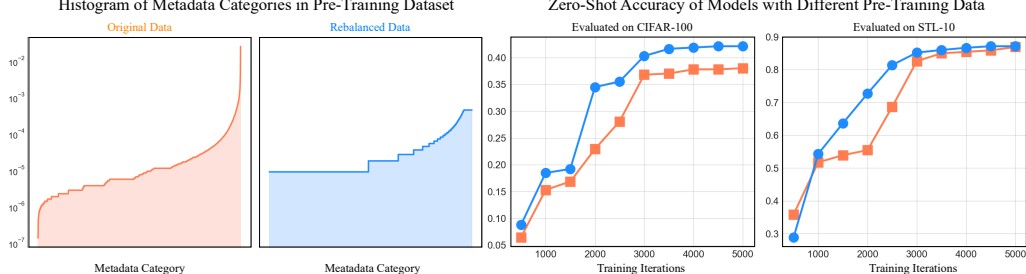

Figure 4: **Balancing and Metadata Curation.** Depiction of balancing and metadata curation (Example 3 in Sec. 2) on ImageNet-Captions dataset, in which $\mathcal{X}$ represents image-caption pairs and $\mathcal{Y}$ represents keywords. **Left:** Observed marginal $P_{n,Y}$ (orange) and $P_Y$ (blue), which are sorted by order of increasing probability. **Right:** Zero-shot evaluation of an embedding model trained using the standard CLIP loss original versus the balanced training set.

**Related Work** Self-supervised learning has witnessed a surge of recent interest as datasets and computing hardware allow for larger, more capable models (see Balestriero et al. [2023] and references therein). While we highlight in this paper the connections between data balancing and contrastive learning [Radford et al., 2021], we acknowledge that data balancing can also be related to "self-distillation" approaches more broadly [Grill et al., 2020, Chen and He, 2021, Oquab et al., 2024].

Historical motivations for data balancing include census or survey data, in which $P_n$ is a cross-tabulation of (a limited number of) paired observations and the target marginals were estimated from large amounts of unpaired observations [Deming and Stephan, 1940, Ireland and Kullback, 1968]. This situation is not unlike the present day—yet at a different scale—in which the amount of unstructured single-modality data (such as images) still dwarfs the amount of high-quality multimodal data [Gadre et al., 2023]. Bickel et al. [1991] proved classical asymptotic results on balancing estimators. Linear operators similar to the ones we use in Sec. 3 also appear in their analysis. More recently, Albertus and Berthet [2019] studied such estimators from an asymptotic empirical process viewpoint. Our theoretical results significantly improve on those from Albertus and Berthet [2019] primarily in the dependence of the number of iterations $k$ on the sample size $n$ to achieve convergence guarantees (from logarithmic to polynomial).

Matrix scaling is a popular algorithm for solving entropy-regularized optimal transport (EOT). We refer to Peyré and Cuturi [2019] for a survey. See also Courty et al. [2017], Shen et al. [2018], Peng et al. [2019] for interesting methods based on EOT in machine learning. Entropy-regularized optimal transport was one of the original inspirations for SSL techniques such as SwaV (see Sec. 2). While EOT is itself a deterministic optimization problem, a related statistical problem is the large-sample limits of EOT solutions when the marginal measures are estimated from data [Mena and Niles-Weed, 2019, Genevay et al., 2019, Klatt et al., 2020]. We emphasize that, while this line of work shares the matrix scaling algorithm with our setting, the statistical problem is entirely distinct; in statistical EOT, the target marginal distributions are computed from observations of independent, unpaired data, and the initial measure can be computed from the cost function. In our setting, the data are dependent, forming the random initial measure $P_n$, whereas $P_X$ and $P_Y$ are fixed auxiliary information.

## 5 Conclusion

We showed how several disparate techniques used towards the training of foundation models are instances of a data balancing algorithm, which has the unsuspected benefit of reducing the variance of learning objectives involving multiple sources of data. We proved a new non-asymptotic bound on the mean-squared error of balanced estimators as they adjust to the given marginals. We also highlight the key roles of conditional expectation operators in quantifying that variance reduction effect. Finally, we translated the marginal balancing interpretation of several training practices for foundation models into algorithmic variants that warrant further investigation. Exploring variants incorporating prior information on the data sources is also an interesting venue for future work.

**Acknowledgements** The authors are grateful to G. Ilharco, M. Wortsman, K. Pillutla, L. Schmidt, and J. Wellner for fruitful discussions related to this work. This work was supported by NSF DMS-2023166, CCF-2019844, DMS-2134012, PIMS 20240827-PRN01, NIH, and IARPA 2022-22072200003. Part of this work was done while L. Liu was with the University of Washington, and while R. Mehta and Z. Harchaoui were visiting the Simons Institute for the Theory of Computing.

**Broader Impact** While this paper is of a theoretical nature, the web-scale pre-training sets used to train foundation models can affect not only the biases of the models themselves but also the behavior of individuals who interact with them. In the case of representation learning, unrefined Internet data may lead to non-uniform performance among protected attributes such as gender, age, etc. For generative models, individuals of all ages may be influenced by harmful images or textual output. Studying the relationship between the balancing procedures considered in this paper and more holistic model evaluations presents a valuable direction for follow-up work.

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

# Appendix

## Table of Contents

# A  Notation

| Symbol | Description |
|---|---|
| $\mathcal{X}, \mathcal{Y}$ | Sample spaces for two data sources. |
| $m, l$ | Support sizes $m = \|\mathcal{X}\|$ and $l = \|\mathcal{Y}\|$. We sometimes assume $m = l$ for ease of presentation. |
| $P$ | Probability measure on $\mathcal{X} \times \mathcal{Y}$ (the data-generating distribution). |
| $n$ | Sample size. |
| $(X_1, Y_1), \ldots, (X_n, Y_n)$ | Independent and identically distributed sample from $P$. |
| $P_n$ | Empirical measure of $\{(X_i, Y_i)\}_{i=1}^n$. |
| $Q_X, Q_Y$ | Marginals of measure $Q$ on $\mathcal{X} \times \mathcal{Y}$, e.g. $R_X, P_Y, P_{n,X}$, etc. |
| $\mathrm{Supp}(Q)$ | For measure $Q$ over $\mathcal{Z}$, the set of values $z \in \mathcal{Z}$ such that $Q(z) > 0$. |
| $Q(h)$ | The expected value of $h$ under $Q$, or $\mathbb{E}_Q[h(X, Y)]$. |
| $(P_n^{(k)})_{k \geq 1}$ | Sequence of iterations of (8). |
| $k$ | Iteration count of (8). |
| $\mathcal{S}$ | The event $\{\mathrm{Supp}(P_{n,X}) = \mathrm{Supp}(P_X) \text{ and } \mathrm{Supp}(P_{n,Y}) = \mathrm{Supp}(P_Y)\}$. |
| $h$ | Test function $h : \mathcal{X} \times \mathcal{Y} \to \mathbb{R}$ of interest. |
| $\varphi$ | The estimand $\sum_{x,y} h(x, y) P(x, y)$. |
| $\varphi_n^{(k)}$ | The estimator $\sum_{x,y} h(x, y) P_n^{(k)}(x, y)$, when well-defined. |
| $\tilde{\varphi}_n^{(k)}$ | The estimator $\tilde{\varphi}_n^{(k)} := \varphi_n^{(k)} \mathbb{1}_{\mathcal{S}} + \varphi_n^{(0)} \mathbb{1}_{\mathcal{S}^c}$. |
| $\mathbb{G}_n^{(k)}(h)$ | Normalized error $\sqrt{n}(\tilde{\varphi}_n^{(k)} - \varphi)$. |
| $V_n^{(k)}(h)$ | Remainder defined in Prop. 15. |
| $\bar{h}$ | Centered function $h - \mathbb{E}_P[h]$. |
| $\sigma_k^2$ | Variance term $\mathbb{E}_P[(\mathcal{C}_1, \ldots \mathcal{C}_k h)^2]$. |
| $p_\star$ | $\min\{\min_x P_X(x), \min_y P_Y(y)\}$. |
| $\mathbf{L}^2(P)$ | Functions $h : \mathcal{X} \times \mathcal{Y} \to \mathbb{R}$ (as $\mathcal{X} \times \mathcal{Y}$ is finite). |
| $\mathbf{L}^2(P_X), \mathbf{L}^2(P_Y)$ | Subspaces of $\mathbf{L}^2(P)$ containing functions only of $x \in \mathcal{X}$ and $y \in \mathcal{Y}$, respectively. |
| $\mu_X, \mu_Y$ | Conditional expectation operators $[\mu_X h](x) := \mathbb{E}_P[h(X, Y)\|X](x)$ and $[\mu_Y h](y) := \mathbb{E}_P[h(X, Y)\|Y](y)$. |
| $\mathcal{C}_X, \mathcal{C}_Y$ | Debiasing/centering operators $\mathcal{C}_X = I - \mu_X$ and $\mathcal{C}_Y = I - \mu_Y$. |
| $\mu_k, \mathcal{C}_k$ | $(\mu_X, \mathcal{C}_X)$ for $k$ odd and $(\mu_Y, \mathcal{C}_Y)$ for $k$ even. |
| $\{s_j\}_{j=1}^m$ | Singular values in Prop. 3. |
| $\{\alpha_j\}_{j=1}^m, \{\beta_j\}_{j=1}^m$ | Bases for $\mathbf{L}^2(P_X)$ and $\mathbf{L}^2(P_Y)$ in Prop. 3. |

Table 1: Notation used throughout the paper.

# B  Linear Operators and Variance Reduction

This section is dedicated to establishing the variance reduction result in Cor. 2 by employing properties of the Markov operators introduced in Sec. 3. In the first part, we establish Prop. 3, the singular value decomposition that defines the quantities appearing in Cor. 2. In the second part, we quantify the difference between $\sigma_0^2$ and $\sigma_k^2$ for even and odd iterations of $k$.

## B.1  Singular Value Decomposition

Recall the conditional mean operators $\mu_X$ and $\mu_Y$ from Sec. 3,

$$[\mu_X h](x) := \mathbb{E}[h(X, Y)|X](x) \text{ and } [\mu_Y h](y) := \mathbb{E}[h(X, Y)|Y](y),$$

with the corresponding debiasing (a.k.a. centering) operators defined by $\mathcal{C}_X = I - \mu_X$ and $\mathcal{C}_Y = I - \mu_Y$.

**Proposition 3.** *There exists a basis $\{\alpha_j\}_{j=1}^m$ of $\mathbf{L}^2(P_X)$, a basis $\{\beta_j\}_{j=1}^m$ of $\mathbf{L}^2(P_Y)$, and real values $\{s_j\}_{j=1}^m$, which satisfy:*

$$\mu_Y \alpha_j = s_j \beta_j \text{ and } \mu_X \beta_j = s_j \alpha_j \text{ for } j \in \{1, \ldots, m\}, \tag{17}$$

$\alpha_1 = \mathbf{1}_{\mathcal{X}}$, $\beta_1 = \mathbf{1}_{\mathcal{Y}}$, $s_1 = 1$ and $s_j$ is non-negative and non-increasing in $j$.

*Proof.* When $\mu_X$ is restricted to $\mathbf{L}^2(P_Y)$ and $\mu_Y$ is restricted to $\mathbf{L}^2(P_X)$, these operators are in fact adjoint in $\mathbf{L}^2(P)$, as by the tower property we have the relation

$$\langle f, \mu_X g \rangle_{\mathbf{L}^2(P_X)} = \mathbb{E}\left[f(X)\mathbb{E}\left[g(Y)|X\right]\right] = \mathbb{E}\left[\mathbb{E}\left[f(X)|Y\right]g(Y)\right] = \langle \mu_Y f, g \rangle_{\mathbf{L}^2(P_Y)}.$$

Since $\mu_Y : \mathbf{L}^2(P_X) \to \mathbf{L}^2(P_Y)$ is a compact linear operator, by Gohberg et al. [1990, Section IV.1 Theorem 1.1] and Gohberg et al. [1990, Section IV.1 Corollary 1.2], we have that $\mu_Y$ admits a singular value decomposition satisfying (17). Next, we show that $s_1 \leq 1$ and that $\mathbf{1}_{\mathcal{X}}$ is an eigenvector of $\mu_X \mu_Y : \mathbf{L}^2(P_X) \to \mathbf{L}^2(P_X)$ with eigenvalue 1, which confirms that $s_1 = 1$ and $\alpha_1 = \mathbf{1}_{\mathcal{X}}$ by the definition of singular values (arguing symmetrically achieves $\beta_1 = \mathbf{1}_{\mathcal{Y}}$). By the variational representation of singular values [Gohberg et al., 1990, Section IV.1 Equation (2)], we have that

$$\sup_{f:\|f\|_{\mathbf{L}^2(P_X)}=1} \|\mu_Y f\|_{\mathbf{L}^2(P_Y)} = s_1.$$

Consider any $f \in \mathbf{L}^2(P_X)$ such that $\|f\|_{\mathbf{L}^2(P_X)} = 1$. Define the conditional probability $P_{X|Y}(x|y) = P(x,y)/P_Y(y)$ which is well-defined by assumption. Then, by the Cauchy-Schwarz inequality in $\mathbf{L}^2(P_{X|Y})$,

$$\begin{aligned}
\|\mu_Y f\|^2_{\mathbf{L}^2(P_Y)} &= \sum_{y \in \mathcal{Y}} \left( \sum_{x \in \mathcal{X}} f(x) P_{X|Y}(x|y) \right)^2 P_Y(y) \\
&\leq \sum_{y \in \mathcal{Y}} \sum_{x \in \mathcal{X}} f^2(x) P_{X|Y}(x|y) P_Y(y) \\
&= \sum_{x \in \mathcal{X}} f^2(x) \sum_{y \in \mathcal{Y}} P(x,y) \\
&= \|f\|^2_{\mathbf{L}^2(P_X)} = 1.
\end{aligned}$$

This proves that $s_1 \leq 1$. For equality, notice that $\mu_X \mu_Y \mathbf{1}_{\mathcal{X}} = \mu_X \mathbf{1}_{\mathcal{Y}} = \mathbf{1}_{\mathcal{X}}$, completing the proof. $\qquad\square$

### B.2 Proof of Main Results

From Prop. 3, we establish two bases $\{\alpha_j\}_{j=1}^m$ and $\{\beta_j\}_{j=1}^m$ of $\mathbf{L}^2(P_X)$ and $\mathbf{L}^2(P_Y)$, respectively. These bases span the range of the operators $\mu_X$ and $\mu_Y$. We will consider the repeated application of the operator $\mathcal{C}_Y \mathcal{C}_X$, a sequence of two centering operations on some function $h \in \mathbf{L}^2(P)$, and compare

$$\mathbb{E}\left[((\mathcal{C}_Y \mathcal{C}_X)^t \bar{h})^2\right] \text{ against } \mathbb{E}\left[\bar{h}^2\right]$$

for $\bar{h} = h - \mathbb{E}_P[h]$. We establish the main result by measuring the reduction in variance from a single application, in terms of the coordinates of the function of interest on each of the two subspaces. We will then observe how these coordinates change iteration-to-iteration to give the final result.

**Lemma 4.** *For any $h \in \mathbf{L}^2(P)$ such that $\mathbb{E}_P[h] = 0$, let*

$$\mu_X h = \sum_{j=1}^m u_j \alpha_j \text{ and } \mu_Y h = \sum_{j=1}^m v_j \beta_j.$$

*Then, we have that*

$$\mathbb{E}\left[(\mathcal{C}_Y \mathcal{C}_X h)^2\right] = \mathbb{E}\left[h^2\right] - \sum_{j=2}^m u_j^2 - \sum_{j=2}^m (v_j - s_j u_j)^2.$$

*Proof.* By orthogonality, we have that

$$\begin{aligned}
\mathbb{E}\left[(\mathcal{C}_Y \mathcal{C}_X h)^2\right] &= \mathbb{E}\left[((I - \mu_Y)\mathcal{C}_X h)^2\right] \\
&= \mathbb{E}\left[(\mathcal{C}_X h)^2\right] - 2\mathbb{E}\left[(\mathcal{C}_X h)(\mu_Y \mathcal{C}_X h)\right] + \mathbb{E}\left[(\mu_Y \mathcal{C}_X h)^2\right] \\
&= \mathbb{E}\left[(\mathcal{C}_X h)^2\right] - 2P_Y((\mu_Y \mathcal{C}_X h)^2) + P_Y((\mu_Y \mathcal{C}_X h)^2) \\
&= \mathbb{E}\left[(\mathcal{C}_X h)^2\right] - P_Y((\mu_Y \mathcal{C}_X h)^2) \\
&= \mathbb{E}\left[h^2\right] - P_X((\mu_X h)^2) - P_Y((\mu_Y \mathcal{C}_X h)^2).
\end{aligned}$$

Because $P(h) = 0$, it holds by the tower property of conditional expectation that $P_X(\mu_X h) = 0$, which implies that

$$u_1 = \langle \mu_X h, \alpha_1 \rangle_{\mathbf{L}^2(P_X)} = 0 \implies P_X((\mu_X h)^2) = \sum_{j=2}^m u_j^2.$$

For the second term, observe that $P_X(\mathcal{C}_X h) = 0$, so it holds by the tower property that $P_Y(\mu_Y \mathcal{C}_X h) = 0$, so

$$P_Y((\mu_Y \mathcal{C}_X h)^2) = \sum_{j=2}^m \left( \langle \mu_Y \mathcal{C}_X h, \beta_j \rangle_{\mathbf{L}^2(P_Y)} \right)^2.$$

Next, we compute the term in the square by applying Prop. 3:

$$
\begin{aligned}
\langle \mu_Y \mathcal{C}_X h, \beta_j \rangle_{\mathbf{L}^2(P_Y)} &= \langle \mu_Y h, \beta_j \rangle_{\mathbf{L}^2(P_Y)} - \langle \mu_Y \mu_X h, \beta_j \rangle_{\mathbf{L}^2(P_Y)} \\
&= v_j - \left\langle \mu_Y \sum_{k=1}^m u_k \alpha_k, \beta_j \right\rangle_{\mathbf{L}^2(P_Y)} \\
&= v_j - \left\langle \sum_{k=1}^m u_k s_k \beta_k, \beta_j \right\rangle_{\mathbf{L}^2(P_Y)} \\
&= v_j - s_j u_j,
\end{aligned}
$$

which completes the proof. $\qquad\square$

Lem. 4 ensures that we have a reduction on each iteration, with a formula that depends on the coordinates of the function on each subspace. Because these coordinates change every iteration, we track them in the next lemma. Define $h_0 = \bar{h}$ and $h_{t+1} = (\mathcal{C}_Y \mathcal{C}_X) h_t$, along with the constants $\{u_{t,j}\}_{j=1}^m$ and $\{v_{t,j}\}_{j=1}^m$ given by

$$\mu_X h_t = \sum_{j=1}^m u_{t,j} \alpha_j \text{ and } \mu_Y h_t = \sum_{j=1}^m v_{t,j} \beta_j.$$

We have the following.

**Lemma 5.** *For all $t \geq 0$, it holds that*

$$
\begin{aligned}
u_{t+1,j} &= s_j^2 u_{t,j} - s_j v_{t,j}, \\
v_{t+1,j} &= 0.
\end{aligned}
$$

*Proof.* Fix any $j \in [m]$, and use Prop. 3 to write

$$
\begin{aligned}
u_{t+1,j} &= \langle \mu_X \mathcal{C}_Y \mathcal{C}_X h_t, \alpha_j \rangle_{\mathbf{L}^2(P_X)} \\
&= \langle \mu_X (I - \mu_X - \mu_Y + \mu_Y \mu_X) h_t, \alpha_j \rangle_{\mathbf{L}^2(P_X)} \\
&= \langle \mu_X \mu_Y \mu_X h_t, \alpha_j \rangle_{\mathbf{L}^2(P_X)} - \langle \mu_X \mu_Y h_t, \alpha_j \rangle_{\mathbf{L}^2(P_X)} \\
&= \left\langle \mu_X \mu_Y \sum_{k=1}^m u_{t,k} \alpha_k, \alpha_j \right\rangle_{\mathbf{L}^2(P_X)} - \left\langle \mu_X \sum_{k=1}^m v_{t,k} \beta_k, \alpha_j \right\rangle_{\mathbf{L}^2(P_X)} \\
&= s_j^2 u_{t,j} - s_j v_{t,j},
\end{aligned}
$$

which proves the first part of the claim. For the second part, note that $\mu_Y \mathcal{C}_Y = 0$, so $\langle \mu_Y \mathcal{C}_Y \mathcal{C}_X h_t, \alpha_j \rangle_{\mathbf{L}^2(P_Y)} = 0$. $\qquad\square$

Using Lem. 4 and Lem. 5, we can simply accumulate the reduction incurred on every iteration.

**Proposition 6.** *Define the constants $(u_j)_{j=1}^m$ and $(v_j)_{j=1}^m$ by*

$$\mu_X \bar{h} = \sum_{j=1}^m u_j \alpha_j \text{ and } \mu_Y \bar{h} = \sum_{j=1}^m v_j \beta_j.$$

*Then, we may quantify the variance reduction achieved by $t+1$ iterations of the $\mathcal{C}_Y \mathcal{C}_X$ operator as*

$$\mathbb{E}\left[\bar{h}^2\right] - \mathbb{E}\left[((\mathcal{C}_Y \mathcal{C}_X)^{t+1}\bar{h})^2\right] = \sum_{j=2}^m \left\{ u_j^2 + (v_j - s_j u_j)^2 \left[1 + \frac{s_j^2(1 - s_j^{4t})}{1 - s_j^2}\right] \right\}$$

$$\to \sum_{j=2}^m \left[ u_j^2 + \frac{(v_j - s_j u_j)^2}{1 - s_j^2} \right]$$

*as $t \to \infty$.*

*Proof.* Apply Lem. 4 $(t+1)$-times so that

$$\mathbb{E}\left[((\mathcal{C}_Y \mathcal{C}_X)^{t+1}\bar{h})^2\right] = \mathbb{E}\left[\bar{h}^2\right] - \sum_{j=2}^m \sum_{\tau=0}^t \left[(1 + s_j^2)u_{\tau,j}^2 + v_{\tau,j}^2 - 2s_j u_{\tau,j}v_{\tau,j}\right]$$

$$= \mathbb{E}\left[\bar{h}^2\right] - \sum_{j=2}^m \left[ v_{0,j}^2 - 2s_j u_{0,j}v_{0,j} + \sum_{\tau=0}^t (1 + s_j^2)u_{\tau,j}^2 \right]$$

as by Lem. 5, we have that $v_{\tau,j} = 0$ for $\tau > 0$. Next, we unroll the definition of $u_{\tau,j}$ so that

$$\begin{aligned} u_{\tau,j} &= s_j^2 u_{\tau-1,j} - s_j v_{\tau-1,j} \\ &= s_j^2 (s_j^2 u_{\tau-2,j} - s_j v_{\tau-2,j}) - s_j v_{\tau-1,j} \\ &= s_j^{2\tau-2}(s_j^2 u_{0,j} - s_j v_{0,j}) \end{aligned}$$

for $\tau > 0$, yielding

$$\mathbb{E}\left[\bar{h}^2\right] - \mathbb{E}\left[((\mathcal{C}_Y \mathcal{C}_X)^{t+1}\bar{h})^2\right]$$

$$= \sum_{j=2}^m \left[ u_{0,j}^2 + (v_{0,j} - s_j u_{0,j})^2 + (1 + s_j^2)(s_j^2 u_{0,j} - s_j v_{0,j})^2 \sum_{\tau=1}^t (s_j^4)^{\tau-1} \right]$$

$$= \sum_{j=2}^m \left[ u_{0,j}^2 + (v_{0,j} - s_j u_{0,j})^2 + (1 + s_j^2)(s_j^2 u_{0,j} - s_j v_{0,j})^2 \sum_{\tau=0}^{t-1} (s_j^4)^{\tau} \right]$$

$$= \sum_{j=2}^m \left[ u_{0,j}^2 + (v_{0,j} - s_j u_{0,j})^2 + \frac{s_j^2(1 + s_j^2)(v_{0,j} - s_j u_{0,j})^2(1 - s_j^{4t})}{1 - s_j^4} \right]$$

$$= \sum_{j=2}^m \left[ u_{0,j}^2 + (v_{0,j} - s_j u_{0,j})^2 + \frac{s_j^2(v_{0,j} - s_j u_{0,j})^2(1 - s_j^{4t})}{1 - s_j^2} \right].$$

Substitute $u_{0,j} = u_j$ and $v_{0,j} = v_j$ to complete the proof. $\qquad \square$

We also present the corresponding result for $k$ odd. The proof follows similarly by repeated application of the operator $\mathcal{C}_Y \mathcal{C}_X$. However, the iterations will be compared to $\sigma_1^2 = \mathbb{E}_P\left[(\mathcal{C}_X \bar{h})^2\right]$, as we consider $\mathcal{C}_X \bar{h}$ as the "first" iteration to this process.

**Proposition 7.** *Define the constants $(u_j)_{j=1}^m$ by*

$$\mu_Y \mathcal{C}_X \bar{h} = \sum_{j=1}^m u_j \beta_j.$$

*Then, we may quantify the variance reduction achieved by $t+1$ iterations of the $\mathcal{C}_X \mathcal{C}_Y$ operator as*

$$\mathbb{E}\left[(\mathcal{C}_X \bar{h})^2\right] - \mathbb{E}\left[((\mathcal{C}_X \mathcal{C}_Y)^{t+1} \mathcal{C}_X \bar{h})^2\right] = \sum_{j=2}^{m} \left\{ u_j^2 + (s_j u_j)^2 \left[1 + \frac{s_j^2(1 - s_j^{4t})}{1 - s_j^2}\right]\right\}$$

$$\to \sum_{j=2}^{m} \left(\frac{1 + s_j^2}{1 - s_j^2}\right) u_j^2$$

*as $t \to \infty$.*

In order to have full monotonicity, we also need that $\sigma_0^2 \geq \sigma_1^2$. This follows by orthogonality, as

$$\sigma_0^2 = \mathbb{E}\left[\bar{h}^2\right] = \mathbb{E}\left[(\mathcal{C}_X \bar{h})^2\right] + \mathbb{E}\left[(\mu_X \bar{h})^2\right] = \sigma_1^2 + \mathbb{E}\left[(\mu_X \bar{h})^2\right] \geq \sigma_1^2. \tag{18}$$

Thus, we can combine Prop. 7 and (18) to fully quantify the relationship between $\sigma_0^2$ and $\sigma_k^2$ for $k$ odd.

## C   From Information Projections to Data Balancing

This section is dedicated to deriving three representations of the balancing procedure as projections in various statistical divergences, as shown in Fig. 2.

We consider two sets of probability measures denoted by $\Pi_X = \{Q : Q_X = P_X\}$ and $\Pi_Y = \{Q : Q_Y = P_Y\}$. The marginal matching steps are written as projections in terms of a statistical divergence $D$ (precisely, an $f$-divergence) in the form

$$\frac{P_X}{P_{n,X}^{(k-1)}} \otimes P_n^{(k-1)} = \underset{Q \in \Pi_X}{\arg\min}\, D(Q \| P_n^{(k-1)}), \quad \frac{P_Y}{P_{n,Y}^{(k-1)}} \otimes R = \underset{Q \in \Pi_Y}{\arg\min}\, D(Q \| P_n^{(k-1)}).$$

We provide the derivations for three common choices of $D$: Kullback-Leibler (KL), reverse KL, and $\chi^2$. Using this viewpoint, and simply assuming the positivity of the marginal measures $P_X$ and $P_Y$, we derive an upper bound in Prop. 14 that is *constant* in $k$. This is an improvement over the recent work of Albertus and Berthet [2019], in which they show an upper bound that scales *exponentially* in $k$.

The KL representation will be used in the proof of Prop. 14, which (recalling the sequence $(P_n^{(k)})_{k \geq 1}$ from (8)), controls the error between $P_{n,Y}^{(k)}$ and $P_Y$ for $k$ odd and $P_{n,X}^{(k)}$ and $P_X$ for $k$ even.

### C.1   Balancing as Information Projections

#### C.1.1   Projection in KL-Divergence

**Proposition 8.** *Assume that $P_X \ll R_X$ and $P_Y \ll R_Y$, and define*

$$Q^\star := \underset{Q \in \Pi_X}{\arg\min}\, \mathrm{KL}(Q \| R), \quad P^\star := \underset{Q \in \Pi_Y}{\arg\min}\, \mathrm{KL}(Q \| R). \tag{19}$$

*Then, it holds that*

$$Q^\star(x, y) = \begin{cases} P_X(x) R_{Y|X}(y|x) & \text{if } R_X(x) > 0 \\ 0 & \text{if } R_X(x) = 0 \end{cases} \tag{20}$$

*and*

$$P^\star(x, y) = \begin{cases} P_Y(y) R_X(x|y) & \text{if } R_Y(y) > 0 \\ 0 & \text{if } R_Y(y) = 0 \end{cases}. \tag{21}$$

*Proof.* In the case that $Q(x,y) = 0$, we apply the convention that $0 \log 0 = 0$. Consider the case $Q^\star$, the projection of $R$ onto $\Pi_X$. Write

$$\mathrm{KL}(Q\|R) = \sum_{x\in\mathcal{X}}\sum_{y\in\mathcal{Y}} Q(x,y) \log \frac{Q_{Y|X}(y|x)Q_X(x)}{R_{Y|X}(y|x)R_X(x)}$$

$$= \sum_{x\in\mathcal{X}} Q_X(x) \left[ \sum_{y\in\mathcal{Y}} Q_{Y|X}(y|x) \log \frac{Q_{Y|X}(y|x)Q_X(x)}{R_{Y|X}(y|x)R_X(x)} \right]$$

$$= \sum_{x\in\mathcal{X}} Q_X(x) \left[ \sum_{y\in\mathcal{Y}} Q_{Y|X}(y|x) \log \frac{Q_{Y|X}(y|x)}{R_{Y|X}(y|x)} + \sum_{y\in\mathcal{Y}} Q_{Y|X}(y|x) \log \frac{Q_X(x)}{R_X(x)} \right]$$

$$= \sum_{x\in\mathcal{X}} Q_X(x) \left[ \sum_{y\in\mathcal{Y}} Q_{Y|X}(y|x) \log \frac{Q_{Y|X}(y|x)}{R_{Y|X}(y|x)} \right] + \sum_{x\in\mathcal{X}} Q_X(x) \log \frac{Q_X(x)}{R_X(x)}$$

$$= \sum_{x\in\mathcal{X}} Q_X(x) \mathrm{KL}(Q_{Y|X}(\cdot|x)\|R_{Y|X}(\cdot|x)) + \mathrm{KL}(Q_X\|R_X)$$

$$= \sum_{x\in\mathcal{X}} P_X(x) \mathrm{KL}(Q_{Y|X}(\cdot|x)\|R_{Y|X}(\cdot|x)) + \mathrm{KL}(P_X\|R_X),$$

where the last line is due to the marginal constraint $Q \in \Pi_X$. For the above to be well defined, we need that $P_X \ll R_X$ so that $\mathrm{KL}(P_X\|R_X) < +\infty$. The above is minimized when $Q_{Y|X}(y|x) = R_{Y|X}(y|x)$ for all $(x,y) \in \mathcal{X} \times \mathcal{Y}$ such that $Q_X(x) = P_X(x) > 0$. The case of $P^\star$ follows analogously when using that $P_Y \ll R_Y$. $\qquad\square$

### C.1.2 Projection in Reverse KL-Divergence

**Proposition 9.** *Assume that $P_Y \ll R_X$ and $P_Y \ll R_Y$, and define*

$$Q^\star := \arg\min_{Q\in\Pi_X} \mathrm{KL}(R\|Q), \quad P^\star := \arg\min_{Q\in\Pi_Y} \mathrm{KL}(R\|Q). \tag{22}$$

*Then, it holds that*

$$Q^\star(x,y) = \begin{cases} P_X(x)R_{Y|X}(y|x) & \text{if } R_X(x) > 0 \\ 0 & \text{if } R_X(x) = 0 \end{cases} \tag{23}$$

*and*

$$P^\star(x,y) = \begin{cases} P_Y(y)R_X(x|y) & \text{if } R_Y(y) > 0 \\ 0 & \text{if } R_Y(y) = 0 \end{cases}. \tag{24}$$

*Proof.* In the case that $R(x,y) = 0$, we apply the convention that $0 \log 0 = 0$. Note that minimizing $\mathrm{KL}(R\|Q)$ over $Q$ is equivalent to minimizing $-\sum_{x,y} R(x,y) \log Q(x,y)$ (i.e. the cross entropy). Consider the case $Q^\star$, the projection of $R$ onto $\Pi_X$. Because $R \ll Q$ for $\mathrm{KL}(R\|Q) < +\infty$ to hold, we have that $R(x) > 0 \implies Q(x) > 0$, so that $Q_{Y|X}(y|x)$ is well-defined. Write

$$-\sum_{x,y} R(x,y) \log Q(x,y)$$

$$= -\sum_{x\in\mathcal{X}} R_X(x) \log Q_X(x) - \sum_{x\in\mathcal{X}} R(x) \sum_{y\in\mathcal{Y}} R_{Y|X}(y|x) \log Q_{Y|X}(y|x)$$

$$= -\sum_{x\in\mathcal{X}} R_X(x) \log P_X(x) + \sum_{x\in\mathcal{X}} R_X(x) \left[ -\sum_{y\in\mathcal{Y}} R_{Y|X}(y|x) \log Q_{Y|X}(y|x) \right].$$

The second first term does not depend on $Q$ due to the marginal constraint $Q \in \Pi_X$. The second term is the expectation of the cross entropy from $R_{Y|X}$ to $Q_{Y|X}$ over $R_X$, which is minimized if $R_{Y|X} = Q_{Y|X}$. We have specified $Q_{Y|X}$ and $Q_X$, completing the proof. $\qquad\square$

### C.1.3 Projection in $\chi^2$-Divergence

Let $\mathbf{1}$ denote the function that is identically equal to 1. Consider the following optimization problem, which is the subject of the subsequent lemmas:

$$\min_{\xi \in \mathcal{A}_X} \|\mathbf{1} - \xi\|_{\mathbf{L}^2(R)}^2, \tag{25}$$

where

$$\mathcal{A}_X := \left\{ f : \mathcal{X} \times \mathcal{Y} \to \mathbb{R} \text{ satisfying } \sum_{y \in \mathcal{Y}} f(x,y) R(x,y) = P_X(x) \text{ for any } x \in \mathcal{X} \right\}.$$

**Lemma 10.** *Assume that $P_X \ll R_X$, and define The problem (25) is feasible, and its solution can be written as*

$$\xi^\star = \mathcal{C}_X^R(\mathbf{1} - f) + f$$

*for any $f \in \mathbf{L}^2(R)$, where the linear operator $\mathcal{C}_X^R$ is specified by*

$$[\mathcal{C}_X^R g](x,y) = g(x,y) - \sum_{y' \in \mathcal{Y}} g(x,y') R_{Y|X}(y'|x).$$

*Proof.* First, we establish feasibility by letting

$$f(x,y) := \begin{cases} P_X(x)/R_X(x) & \text{if } R_X(x) > 0 \\ 1 & \text{otherwise} \end{cases}.$$

This function does not depend on the second input $y$. Because we assumed that $P_X \ll R_X$, we have that the terms of $f(x,y)$ for which $R_X(x) = 0$ do not affect whether $\sum_{y \in \mathcal{Y}} f(x,y) R(x,y) = P_X(x)$, because $P_X(x) = 0$ in these cases. In the remainder of this proof, we will show that (25) is an affine projection problem, and find its solution by converting it to a subspace projection problem. Indeed, consider $f_1, \ldots, f_r \in \mathcal{A}_X$, and $\alpha_1, \ldots, \alpha_r \in \mathbb{R}$ such that $\sum_{j=1}^r \alpha_j = 1$. Then,

$$\sum_{y \in \mathcal{Y}} \left[ \sum_{j=1}^r \alpha_j f_j(x,y) \right] \cdot R(x,y) = \sum_{j=1}^r \alpha_j \left[ \sum_{y \in \mathcal{Y}} f_j(x,y) R(x,y) \right] = P_X(x),$$

indicating that $\sum_{j=1}^r \alpha_j f_j(x,y) \in \mathcal{A}_X$ and $\mathcal{A}_X$ is an affine subset of $\mathbf{L}^2(R)$. Define

$$\mathcal{S}_X := \left\{ g : \mathcal{X} \times \mathcal{Y} \to \mathbb{R} \text{ satisfying } \sum_{y \in \mathcal{Y}} g(x,y) R(x,y) = 0 \text{ for any } x \in \mathcal{X} \right\}.$$

Then, for any $f \in \mathcal{A}_X$, we have that $g \in \mathcal{S}_X$ if and only if $g + f \in \mathcal{A}_X$. Taking any $f \in \mathcal{A}_X$, letting $\phi^\star$ be the solution of

$$\min_{\phi \in \mathcal{S}_X} \|\mathbf{1} - f - \phi\|_{\mathbf{L}^2(R)}^2, \tag{26}$$

we will have that $\phi^\star + f$ will be the solution of (25). The remainder of the proof is showing that $\phi^\star = \mathcal{C}_X^R(\mathbf{1} - f)$.

First, define the operator $\mu_X^R$ by $[\mu_X g](x,y) = \sum_{y' \in \mathcal{Y}} g(x,y') R_{Y|X}(y'|x)$, and note (by factoring out $R_X(x)$) that $g \in \mathcal{S}_X$ if and only if $\mu_X^R g = 0$. In addition, $\mu_X^R g$ is linear and idempotent as $\mu_X^R \mu_X^R g = \mu_X^R g$, so it is a projection operator in $\mathbf{L}^2(R)$. Thus, $\mathcal{S}_X$ is the orthogonal complement of $\text{range}(\mu_X^R)$, and the solution of (26) is given by $(I - \mu_X^R)(\mathbf{1} - f) = \mathcal{C}_X^R(\mathbf{1} - f)$, because $\mathcal{C}_X^R = I - \mu_X^R$. The claim is proved. $\square$

**Lemma 11.** *Assume that $P_X \ll R_X$. Define*

$$Q^\star := \arg\min_{Q \in \Pi_X} \chi^2(Q \| R). \tag{27}$$

*and let $\xi^\star$ be the solution of problem (25). Then,*

$$Q^\star(x,y) = \xi^\star(x,y) R(x,y) = \begin{cases} P_X(x) R_{Y|X}(y|x) & \text{if } R_X(x) > 0 \\ 0 & \text{if } R_X(x) = 0 \end{cases}. \tag{28}$$

*Proof.* First, by reparametrizing the problem (27) as finding $\xi$ such that $Q(x,y) = \xi(x,y)R(x,y)$, we can compute its solution by solving

$$\min_{\xi \in \mathcal{A}_X, \xi \geq 0} \|\mathbf{1} - \xi\|_{\mathbf{L}^2(R)}^2, \tag{29}$$

Notice that we also have a non-negativity constraint, as opposed to (25). If $\xi^\star$ solves (25) and happens to be non-negative, then we have that $\xi^\star$ solves (29) as well and the first equality of (28) is satisfied by definition. We show the second equality of (28) by direct computation, which also establishes the non-negativity of $\xi^\star$ simultaneously.

Apply Lem. 10 with

$$f(x,y) := \begin{cases} P_X(x)/R_X(x) & \text{if } R_X(x) > 0 \\ 1 & \text{otherwise} \end{cases}.$$

so that

$$\begin{aligned}
\xi^\star(x,y) &= \mathcal{C}_X^R(\mathbf{1} - f)(x,y) + f(x,y) \\
&= \left[ \sum_{z \in \mathcal{Y}} f(x,z)R_{Y|X}(z|x) - f(x,y) \right] + f(x,y) \\
&= f(x,y')
\end{aligned}$$

for any $y' \in \mathcal{Y}$. Thus, the likelihood ratio of $Q^\star$ with respect to $R$ is a marginal reweighting. Accordingly,

$$Q^\star(x,y) = \xi^\star(x,y)R(x,y) = \begin{cases} P_X(x)R_{Y|X}(y|x) & \text{if } R_X(x) > 0 \\ 0 & \text{if } R_X(x) = 0 \end{cases},$$

completing the proof. □

**Proposition 12.** *Assume that $P_X \ll R_X$ and $P_Y \ll R_Y$. Define*

$$Q^\star := \arg\min_{Q \in \Pi_X} \chi^2(Q\|R), \quad P^\star := \arg\min_{Q \in \Pi_Y} \chi^2(Q\|R). \tag{30}$$

*Then, it holds that*

$$\begin{aligned}
Q^\star(x,y) &= \begin{cases} P_X(x)R_{Y|X}(y|x) & \text{if } R_X(x) > 0 \\ 0 & \text{if } R_X(x) = 0 \end{cases} \\
P^\star(x,y) &= \begin{cases} P_Y(y)R_{X|Y}(x|y) & \text{if } R_Y(y) > 0 \\ 0 & \text{if } R_Y(y) = 0 \end{cases}.
\end{aligned} \tag{31}$$

*Proof.* The first equality of (31) follows by the claim of Lem. 11. The second equality follows by repeating the argument of Lem. 10 and Lem. 11 with $(X,x)$ and $(Y,y)$ swapped. □

### C.2 Proof of Main Results

We may now control the errors of the ratio of marginals using the projection interpretation established in the previous sections. Recall the event $\mathcal{S}$ as defined in Tab. 1. The following result, the monotonicity of the marginal violation terms in terms of KL, will be useful in the bound.

**Proposition 13.** *[Nutz, 2021, Proposition 6.10] Under the event $\mathcal{S}$, it holds that*

$$\mathrm{KL}(P_{n,X}^{(0)}\|P_X) \geq \mathrm{KL}(P_Y\|P_{n,Y}^{(1)}) \geq \mathrm{KL}(P_{n,X}^{(2)}\|P_X) \geq \dots$$

We give the following result for $\mathcal{X}$ and the analogous claim holds on $\mathcal{Y}$.

**Proposition 14.** *Assume that $P_{n,X}(x) > 0$ for all $x \in \mathcal{X}$. It holds that*

$$\max_{x \in \mathcal{X}} \left| \frac{P_X(x)}{P_{n,X}^{(k-1)}(x)} - 1 \right| \leq \begin{cases} \max\{n-1, 1\} & \text{if } k = 1 \\ \max\{1/p_\star^2 - 1, 1\} & \text{if } k > 1 \end{cases}. \tag{32}$$

*In addition, we have that*

$$\max_{x \in \mathcal{X}} \left| \frac{P_X(x)}{P_{n,X}^{(k-1)}(x)} - 1 \right| \leq \begin{cases} n\sqrt{\frac{1}{2} \, \mathrm{KL}(P_{n,X} \| P_X)} & \text{if } k = 1 \\ \frac{1}{p_\star^2} \sqrt{\frac{1}{2} \, \mathrm{KL}(P_{n,X} \| P_X)} & \text{if } k > 1 \end{cases}.$$

*Moreover, when* $\mathrm{KL}(P_{n,X} \| P_X) \leq p_\star^2 / 2$, *we have*

$$\max_{x \in \mathcal{X}} \left| \frac{P_X(x)}{P_{n,X}^{(k-1)}(x)} - 1 \right| \leq \frac{2}{p_\star} \sqrt{\frac{1}{2} \, \mathrm{KL}(P_{n,X} \| P_X)}. \tag{33}$$

*Proof.* We first show that $P_n^{(k-1)}(x) \geq 1/n$ for $k = 1$ and $P_n^{(k-1)}(x) \geq p_\star^2$ for $k > 1$. In the case that $k = 1$, the result follows directly from the event $\mathcal{S}$. For $k > 1$ such that $k$ is odd, we have that for $x \in \mathcal{X}$,

$$P_{n,X}^{(k-1)}(x) = \sum_{y \in \mathcal{Y}} P_n^{(k-1)}(x, y) = \sum_{y \in \mathcal{Y}} \frac{P_Y(y)}{P_{n,Y}^{(k-2)}(y)} P_n^{(k-2)}(x, y)$$

$$\geq p_\star \sum_{y \in \mathcal{Y}} P_n^{(k-2)}(x, y) = p_\star P_{n,X}^{(k-2)}(x) = p_\star P_X(x) \geq p_\star^2.$$

The result for $k$ even can be proven similarly. We now proceed to prove the inequalities given in the statement, which will rely on the lower bound above.

**Proving the first inequality.** Then, for any $x \in \mathcal{X}$,

$$\left| \frac{P_X(x)}{P_{n,X}^{(k-1)}(x)} - 1 \right| = \max \left\{ \frac{P_X(x)}{P_{n,X}^{(k-1)}(x)} - 1, 1 - \frac{P_X(x)}{P_{n,X}^{(k-1)}(x)} \right\} \leq \begin{cases} \max\{n - 1, 1\} & \text{if } k = 1 \\ \max\{1/p_\star^2 - 1, 1\} & \text{if } k > 1 \end{cases},$$

which is the desired result for the first inequality.

**Proving the second and third inequalities.** Consider an odd $k \geq 1$. By the definition of total variation distance, it holds that

$$\max_{x \in \mathcal{X}} \left| P_X(x) - P_{n,X}^{(k-1)}(x) \right| \leq \mathrm{TV}(P_{n,X}^{(k-1)}, P_X).$$

According to Pinsker's inequality, we have that $\mathrm{TV}(P_{n,X}^{(k-1)}, P_X) \leq \sqrt{\frac{1}{2} \, \mathrm{KL}(P_{n,X}^{(k-1)} \| P_X)}$, and so we have that

$$\max_{x \in \mathcal{X}} \left| P_X(x) - P_{n,X}^{(k-1)}(x) \right| \leq \sqrt{\frac{1}{2} \, \mathrm{KL}(P_{n,X}^{(k-1)} \| P_X)} \leq \sqrt{\frac{1}{2} \, \mathrm{KL}(P_{n,X}^{(0)} \| P_X)},$$

where the last inequality follows by the monotonicity of Sinkhorn iterations given in Prop. 13. We apply the lower bounds to write

$$\max_{x \in \mathcal{X}} \left| \frac{P_X(x)}{P_{n,X}^{(k-1)}(x)} - 1 \right| \leq \begin{cases} n\sqrt{\frac{1}{2} \, \mathrm{KL}(P_{n,X} \| P_X)} & \text{if } k = 1 \\ \frac{1}{p_\star^2} \sqrt{\frac{1}{2} \, \mathrm{KL}(P_{n,X} \| P_X)} & \text{if } k > 1 \end{cases}.$$

Finally, when $\sqrt{\frac{1}{2} \, \mathrm{KL}(P_{n,X} \| P_X)} \leq p_\star / 2$, we have that $\max_{x \in \mathcal{X}} \left| P_X(x) - P_{n,X}^{(k-1)}(x) \right| \leq p_\star / 2$ and thus

$$\min_{x \in \mathcal{X}} P_{n,X}^{(k-1)}(x) \geq \min_{x \in \mathcal{X}} P_X(x) - \max_{x \in \mathcal{X}} \left| P_{n,X}^{(k-1)}(x) - P_X(x) \right| \geq \frac{p_\star}{2}.$$

Hence,

$$\max_{x \in \mathcal{X}} \left| \frac{P_X(x)}{P_{n,X}^{(k-1)}(x)} - 1 \right| \leq \frac{\max_{x \in \mathcal{X}} \left| P_{n,X}^{(k-1)}(x) - P_X(x) \right|}{\min_{x \in \mathcal{X}} P_{n,X}^{(k-1)}(x)} \leq \frac{2}{p_\star} \sqrt{\frac{1}{2} \, \mathrm{KL}(P_{n,X} \| P_X)}.$$

Now, for $k$ even, set $k = 2t$ for $t \geq 0$. We have that

$$\max_{y \in \mathcal{Y}} \left| P_{n,Y}^{(2t-1)}(y) - P_Y(y) \right| \leq \text{TV}(P_{n,Y}^{(2t-1)}, P_Y) \leq \sqrt{\frac{1}{2} \text{KL}(P_Y \| P_{n,Y}^{(2t-1)})}.$$

Invoke Prop. 13 once again to achieve

$$\sqrt{\frac{1}{2} \text{KL}(P_Y \| P_{n,Y}^{(2t-1)})} \leq \sqrt{\frac{1}{2} \text{KL}(P_{n,X} \| P_X)},$$

which completes the proof. $\qquad\qquad\square$

# D  Statistical Analysis of Balancing Estimators

This section contains the proof of the main result, namely Thm. 1. We first consolidate notation and then give a broad outline of the proof for readability. Let the expectation of a function $h$ under a probability measure $Q$ on $\mathcal{X} \times \mathcal{Y}$ by denoted by

$$Q(h) = \sum_{x \in \mathcal{X}, y \in \mathcal{Y}} h(x, y) Q(x, y)$$

so that

$$\varphi_n^{(k)} = P_n^{(k)}(h), \quad \varphi = P(h),$$

and

$$\mathbb{G}_n^{(k)}(h) = \sqrt{n}[P_n^{(k)} - P](h) = \sqrt{n}(P_n^{(k)}(h) - P(h)). \tag{34}$$

Recalling in addition that $\mathcal{C}_k = \mathcal{C}_X$ for $k$ odd and $\mathcal{C}_k = \mathcal{C}_Y$ for $k$ even. The event

$$\mathcal{S} := \{ \text{Supp}(P_{n,X}) = \text{Supp}(P_X) \text{ and } \text{Supp}(P_{n,Y}) = \text{Supp}(P_Y) \}, \tag{35}$$

is used for purely technical reasons in many results.

**Proof Outline.** We first establish that the recursion formula

$$[P_n^{(k)} - P](h) = [P_n^{(k-1)} - P](\mathcal{C}_k h) + V_n^{(k-1)}(\mathcal{C}_k h)$$

holds in Prop. 15, where

$$V_n^{(k-1)}(h) = \begin{cases} \sum_{x,y} \left( \frac{P_X}{P_{n,X}^{(k-1)}}(x) - 1 \right) h(x,y) P_n^{(k-1)}(x,y) & k \text{ odd} \\ \sum_{x,y} \left( \frac{P_Y}{P_{n,Y}^{(k-1)}}(y) - 1 \right) h(x,y) P_n^{(k-1)}(x,y) & k \text{ even} \end{cases}. \tag{36}$$

The quantity $V_n^{(k-1)}(\mathcal{C}_k h)$ describes an error term that accumulates for each iteration of balancing, which explains why $k$ must be scaled appropriately against $n$ to ensure the error does not accumulate too fast. Applying the recursion repeatedly to the balanced sequence $(P_n^{(k)})_{k \geq 1}$ and unrolling the recursion, we see that when $k$ is odd,

$$\begin{aligned} [P_n^{(k)} - P](h) &= [P_n^{(k-1)} - P](\mathcal{C}_X h) + V_n^{(k-1)}(\mathcal{C}_X h) \\ &= [P_n^{(k-2)} - P](\mathcal{C}_Y \mathcal{C}_X h) + V_n^{(k-2)}(\mathcal{C}_Y \mathcal{C}_X h) + V_n^{(k-1)}(\mathcal{C}_X h) \\ &= \underbrace{[P_n^{(0)} - P](\mathcal{C}_1 \dots \mathcal{C}_k h)}_{\text{first-order term}} + \underbrace{\sum_{\ell=1}^{k} V_n^{(\ell-1)}(\mathcal{C}_\ell \dots \mathcal{C}_k h)}_{\text{higher-order term}} \end{aligned} \tag{37}$$

Additionally, let $h_{\ell,k} := \mathcal{C}_\ell \dots \mathcal{C}_k h$, so that the first-order term can be written as $P_n^{(0)}(h_{1,k}) - P(h_{1,k})$ higher-order term can also be written as $\sum_{\ell=1}^{k} V_n^{(\ell-1)}(h_{\ell,k})$. Because our original goal is to upper bound the mean squared error, we use the expansion above to write

$$\begin{aligned} \mathbb{E} \left| P_n^{(k)}(h) - P(h) \right|^2 &\leq \mathbb{E} \left| P_n^{(0)}(h_{1,k}) - P(h_{1,k}) \right|^2 \\ &+ 2\mathbb{E} \left| P_n^{(0)}(h_{1,k}) - P(h_{1,k}) \right| \left| \sum_{\ell=1}^{k} V_n^{(\ell-1)}(h_{\ell,k}) \right| + \mathbb{E} \left| \sum_{\ell=1}^{k} V_n^{(\ell-1)}(h_{\ell,k}) \right|^2 \end{aligned}$$

Regarding the first term, we have that $\mathbb{E} \left| P_n^{(0)}(h_{1,k}) - P(h_{1,k}) \right|^2 = \sigma_k^2 / n$, which is the dominant term in Thm. 1. Thus, the remaining challenge of the proof will be to upper bound the cross term and squared term and show its dependence on $n$. The dominant term of these two will be the cross term, as we will essentially show that $|P_n^{(0)}(h_{1,k}) - P(h_{1,k})|$ is $O(n^{-1/2})$ with high probability and that $|\sum_{\ell=1}^{k} V_n^{(\ell-1)}(h_{\ell,k})|$ is in fact $O(n^{-1})$ with high probability. As stated in Sec. 3, a key intermediate result in controlling the higher-order term is Prop. 14, whose proof is given in Appx. C. The remaining subsections walk through these steps in detail.

## D.1 Recursion of Estimation Error

We first recall that the sequence $(P_n^{(k)})_{k\geq 1}$ can be computed with the following formula:

$$P_n^{(0)}(x,y) := P_n(x,y) \text{ and } P_n^{(k)}(x,y) := \begin{cases} \frac{P_X}{P_{n,X}^{(k-1)}}(x)P_n^{(k-1)}(x,y) & k \text{ odd} \\ \frac{P_Y}{P_{n,Y}^{(k-1)}}(y)P_n^{(k-1)}(x,y) & k \text{ even} \end{cases}. \tag{38}$$

Prop. 15 establishes the conditions under which these steps are well-defined (i.e. $P_{n,X}^{(k-1)}(x) > 0$ and $P_{n,Y}^{(k-1)}(y) > 0$).

**Proposition 15.** *Let $(P_n^{(k)})_{k\geq 1}$, be a sequence computed according to (8). These iterations are well-defined under the event $\bar{S}$, and for $\mathbb{G}_n^{(k)}$ defined in (34) and $V_n^{(k)}$ defined in (36), it holds that*

$$\mathbb{G}_n^{(k)}(h) = \mathbb{G}_n^{(k-1)}(h) + \sqrt{n}V_n^{(k-1)}(h). \tag{39}$$

*and*

$$\mathbb{G}_n^{(k)}(h) = \mathbb{G}_n^{(k-1)}(\mathcal{C}_k h) + \sqrt{n}V_n^{(k-1)}(\mathcal{C}_k h). \tag{40}$$

*Proof.* First, assume that $P_{n,X}^{(k-1)}(x) > 0$ and $P_{n,Y}^{(k-1)}(y) > 0$ for all $x \in \mathcal{X}$ and $y \in \mathcal{Y}$ so that we may establish the recursion, which we will show by induction toward the end of the proof.

Consider the following steps in the case that $k$ is odd:

$$P_n^{(k)}(h)$$

$$= \sum_{x,y} h(x,y)P_n^{(k)}(x,y) = \sum_{x,y} h(x,y)\frac{P_X}{P_{n,X}^{(k-1)}}(x)P_n^{(k-1)}(x,y) \qquad \text{by (38) for } k \text{ odd}$$

$$= \sum_{x,y} 1 \cdot h(x,y)P_n^{(k-1)}(x,y) + \sum_{x,y} \left[\frac{P_X}{P_{n,X}^{(k-1)}}(x) - 1\right] \cdot h(x,y)P_n^{(k-1)}(x,y)$$

$$= P_n^{(k-1)}(h) + V_n^{(k-1)}(h).$$

Arguing analogously for $k$ even and subtracting $P(h)$ on both sides, we have that

$$[P_n^{(k)} - P](h) = [P_n^{(k-1)} - P](h) + V_n^{(k-1)}(h). \tag{41}$$

We refer to this as the "uncentered" recursion, which proves (39).

We can then establish the following "centered" recursion using the following decomposition in the case of $k$ odd.

$$[P_n^{(k)} - P](h)$$
$$= [P_n^{(k)} - P](\mathcal{C}_X h) + [P_n^{(k)} - P](\mu_X h) \qquad\qquad h = \mathcal{C}_X h + \mu_X h$$
$$= [P_n^{(k-1)} - P](\mathcal{C}_X h) + V_n^{(k-1)}(\mathcal{C}_X h) + [P_n^{(k)} - P](\mu_X h) \qquad \text{apply (41) to } \mathcal{C}_X h$$
$$= [P_n^{(k-1)} - P](\mathcal{C}_X h) + V_n^{(k-1)}(\mathcal{C}_X h). \qquad\qquad P_n^{(k)}(\mu_X h) = P(\mu_X h)$$

The last line follows because $\mu_X h$ is only a function on $\mathcal{X}$, and due to the definition of the marginal rebalancing iterations, $P_{n,X}^{(k)} = P_X$. This gives the desired formula by substituting (34).

We proceed to show that the iterations are well-defined. We will in fact show that $P_{n,X}^{(k-1)}(x) > 0$ and $P_{n,Y}^{(k-1)}(y) > 0$ for all $x \in \mathcal{X}$ and $y \in \mathcal{Y}$. For $k = 1$, $P_{n,X}^{(0)}(x) = P_{n,X}(x) > 0$ and $P_{n,Y}^{(0)}(y) = P_{n,Y}(y) > 0$ for all $x \in \mathcal{X}$ and $y \in \mathcal{Y}$ this holds under the event $S$ by assumption. We argue by induction that this holds for all $k > 1$. Assume that the claim is true for $\{1, \ldots, k-1\}$, and that $k$ is even. Then,

$$P_{n,X}^{(k-1)}(x) = P_X(x) > 0,$$

$$P_{n,Y}^{(k-1)}(y) = \sum_{x \in \mathcal{X}} P_n^{(k-1)}(x,y) = \sum_{x \in \mathcal{X}} \frac{P_X}{P_{n,X}^{(k-2)}}(x)P_n^{(k-2)}(x,y)$$

$$\geq \min_{x \in \mathcal{X}} \frac{P_X}{P_{n,X}^{(k-2)}}(x) \cdot P_{n,Y}^{(k-2)}(y) > 0$$

as $P_{n,X}^{(k-2)}(x) > 0$ and $P_{n,Y}^{(k-2)}(y) > 0$ by the inductive hypothesis. Arguing analogously for $k$ odd achieves the claim. $\square$

## D.2 Technical Tools & Intermediate Results

Having established the backbone of the argument, we collect in this subsection some useful tools that are used in the remainder of the proofs.

The following result follows from the method of types in information theory and will be helpful in deriving the dependence of the higher-order term on $n$.

**Theorem 16.** *[Cover, 1999, Theorem 11.2.1] Let $\nu$ be a discrete probability measure supported on $m$ atoms. Let $U_1, \ldots, U_n \overset{\text{i.i.d.}}{\sim} \nu$ and $\nu_n$ be the associated empirical measure. Then, we have for any $\epsilon > 0$ that*

$$\mathbb{P}\left(\mathrm{KL}(\nu_n \| \nu) \geq \epsilon\right) \leq 2^{-n\left(\epsilon - m\frac{\log(n+1)}{n}\right)}.$$

We then provide a result that counts the number of terms that appear when repeatedly centering via the operators $\mathcal{C}_1, \ldots, \mathcal{C}_k$. This formalizes the pattern

$$\mathcal{C}_X = I - \mu_X$$
$$\mathcal{C}_Y \mathcal{C}_X = I - \mu_X - \mu_Y + \mu_Y \mu_X$$
$$\mathcal{C}_X \mathcal{C}_Y \mathcal{C}_X = I - \mu_X - \mu_Y + \mu_Y \mu_X + \mu_X \mu_Y - \mu_X \mu_Y \mu_X,$$

and so on. This will be useful when bounding $h_{\ell,k}$ uniformly.

**Lemma 17.** *For any $k \geq 1$ and $\ell \in \{1, \ldots, k\}$,*

$$\mathcal{C}_\ell \ldots \mathcal{C}_k = I - \sum_{\tau=0}^{(k-\ell-1)/2} (\mu_X \mu_Y)^\tau \mu_X - \sum_{\tau=0}^{(k-\ell-1)/2} (\mu_Y \mu_X)^\tau \mu_Y$$
$$+ \sum_{\tau=1}^{(k-\ell)/2} (\mu_X \mu_Y)^\tau + \sum_{\tau=1}^{(k-\ell)/2} (\mu_Y \mu_X)^\tau + (-1)^{k-\ell+1} \mu_\ell \ldots \mu_k,$$

*where the sum $\sum_{\tau=i}^{j}$ is 0 when $i > j$ and is $\sum_{\tau=i}^{\lfloor j \rfloor}$ when $j$ is not an integer by convention.*

*Proof.* We prove the claim by backward induction on $\ell$, for the case that $k$ is odd. In the case $\ell = k$, the claim holds because $\mathcal{C}_k = I - \mu_k$. Next, for any $\ell < k$, assume that the stated result holds for $\{\ell+1, \ldots, k\}$. Then, if $\ell$ is also odd (so that $\mu_\ell = \mu_X$),

$$\mathcal{C}_\ell \ldots \mathcal{C}_k = \mathcal{C}_\ell \mathcal{C}_{\ell+1} \ldots \mathcal{C}_k$$
$$= I - \sum_{\tau=0}^{(k-\ell-2)/2} (\mu_X \mu_Y)^\tau \mu_X - \sum_{\tau=0}^{(k-\ell-2)/2} (\mu_Y \mu_X)^\tau \mu_Y$$
$$+ \sum_{\tau=1}^{(k-\ell-1)/2} (\mu_X \mu_Y)^\tau + \sum_{\tau=1}^{(k-\ell-1)/2} (\mu_Y \mu_X)^\tau + \mu_Y \underbrace{\cdots}_{k-\ell \text{ terms}} \mu_X$$
$$- \mu_X + \sum_{\tau=0}^{(k-\ell-2)/2} (\mu_X \mu_Y)^\tau \mu_X + \sum_{\tau=0}^{(k-\ell-2)/2} \mu_X (\mu_Y \mu_X)^\tau \mu_Y$$
$$- \sum_{\tau=1}^{(k-\ell-1)/2} (\mu_X \mu_Y)^\tau - \sum_{\tau=1}^{(k-\ell-1)/2} \mu_X (\mu_Y \mu_X)^\tau - (\mu_X \mu_Y)^{(k-\ell)/2} \mu_X$$

The red terms and blue terms cancel out to zero. This leaves

$$\mathcal{C}_\ell \ldots \mathcal{C}_k = I - \sum_{\tau=0}^{(k-\ell-2)/2} (\mu_X \mu_Y)^\tau \mu_X - \sum_{\tau=0}^{(k-\ell-2)/2} (\mu_Y \mu_X)^\tau \mu_Y$$
$$+ \sum_{\tau=1}^{(k-\ell-1)/2} (\mu_Y \mu_X)^\tau + (\mu_Y \mu_X)^{(k-\ell)/2}$$
$$+ \sum_{\tau=0}^{(k-\ell-2)/2} \mu_X (\mu_Y \mu_X)^\tau \mu_Y + (-1)^{k-\ell+1} \mu_\ell \ldots \mu_k$$

wherein we combine the red terms and re-index the blue terms to get

$$\mathcal{C}_\ell \dots \mathcal{C}_k = I - \sum_{\tau=0}^{(k-\ell-2)/2} (\mu_X \mu_Y)^\tau \mu_X - \sum_{\tau=0}^{(k-\ell-2)/2} (\mu_Y \mu_X)^\tau \mu_Y$$
$$+ \sum_{\tau=1}^{(k-\ell)/2} (\mu_Y \mu_X)^\tau + \sum_{\tau=1}^{(k-\ell)/2} (\mu_X \mu_Y)^\tau + (-1)^{k-\ell+1} \mu_\ell \dots \mu_k.$$

Finally, because $k - \ell$ is even when $k$ is odd and $\ell$ is odd, we can set the upper bound of the first two sums to $(k - \ell - 1)/2$ without changing the number of terms. This proves the desired result. The result can be proved similarly when $\ell$ is even. As a result, we have proved the claim for any odd k and $\ell \leq k$. Similar arguments can be used for the case of $k$ even and $\ell \leq k$. $\qquad\square$

### D.3 Analysis of Higher-Order Term

Returning to the outline at the start of this section, we may now bound the higher-order remainder term in (37), namely

$$\sum_{\ell=1}^k V_n^{(\ell-1)}(h_{\ell,k}) = \sum_{\ell=1}^k V_n^{(\ell-1)}(\mathcal{C}_\ell \dots \mathcal{C}_k h),$$

depends on controlling the quantity $V_n^{(k-1)}$ in the summation, which we recall for convenience:

$$V_n^{(k-1)}(h) = \begin{cases} \sum_{x,y} \left( \frac{P_X}{P_{n,X}^{(k-1)}}(x) - 1 \right) h(x,y) P_n^{(k-1)}(x,y) & k \text{ odd} \\ \sum_{x,y} \left( \frac{P_Y}{P_{n,Y}^{(k-1)}}(y) - 1 \right) h(x,y) P_n^{(k-1)}(x,y) & k \text{ even} \end{cases}. \tag{42}$$

Because we have established uniform control over the functions $P_X/P_{n,X}^{(k-1)} - 1$ and $P_Y/P_{n,Y}^{(k-1)} - 1$, via Prop. 14 in Appx. C we can now bound the full remainder in Prop. 20.

We also make use of the following intermediate result, which controls how large the $\ell_\infty$-norm of the function $h$ can grow after centering.

**Lemma 18.** $\|h_{\ell,k}\|_\infty \leq 2(k - \ell + 1) \|h\|_\infty$.

*Proof.* Apply Lem. 17 and the triangle inequality, so that we only need to count the number of terms that appear in the sums, adding 2 for the first and last term in the expression. We subtract 1 from the total, as one of either $(k - \ell)/2$ or $(k - \ell + 1)/2$ will be a fraction. This yields $2(k - \ell + 1)$ terms total, the desired result. $\qquad\square$

We upper bound the sum in Prop. 20. To do so, we introduce some notation. Consider $B_1$ and $B_2$ defined by

$$B_1 := M_1 \quad \text{and} \quad B_2 := \max_{2 \leq \ell \leq k} M_\ell \quad \text{for} \quad M_\ell := \begin{cases} \max_{x \in \mathcal{X}} \left| \frac{P_X(x)}{P_{n,X}^{(\ell-1)}(x)} - 1 \right| & \ell \text{ odd} \\ \max_{y \in \mathcal{Y}} \left| \frac{P_Y(y)}{P_{n,Y}^{(\ell-1)}(y)} - 1 \right| & \ell \text{ even} \end{cases}$$

for $k \geq 1$. We also enumerate the sample spaces as $\mathcal{X} = \{x_1, \dots, x_m\}$ and $\mathcal{Y} = \{y_1, \dots, y_m\}$, and define the function

$$\mathbb{1}_{jk}(x,y) := \begin{cases} \mathbb{1}\{x = x_j\} & k \text{ odd} \\ \mathbb{1}\{y = y_j\} & k \text{ even} \end{cases}.$$

This is an indicator function on the $j$-th element of either $\mathcal{X}$ or $\mathcal{Y}$ depending on whether $k$ is odd or even. Finally, for any function $h$, use (under the event $\mathcal{S}$) recall the empirical process notation

$$\mathbb{G}_n^{(k)}(h) := \sqrt{n} \left( P_n^{(k)}(h) - P(h) \right). \tag{43}$$

Using this notation, we can rewrite the recursion in terms of the quantity $\mathbb{G}_n^{(k)}(h)$ itself. This is established in the following lemma.

**Lemma 19.** *For $k$ odd, it holds that*

$$\mathbb{G}_n^{(k)}(h) = \mathbb{G}_n^{(k-1)}(\mathcal{C}_X h) + \sum_{j=1}^{m} \left[ \frac{P_X(x_j)}{P_{n,X}^{(k-1)}(x_j)} - 1 \right] \mathbb{G}_n^{(k-1)}(\mathcal{C}_X h\, \mathbf{1}_{jk}),$$

*whereas for $k$ even, it holds that*

$$\mathbb{G}_n^{(k)}(h) = \mathbb{G}_n^{(k-1)}(\mathcal{C}_Y h) + \sum_{j=1}^{m} \left[ \frac{P_Y(y_j)}{P_{n,Y}^{(k-1)}(y_j)} - 1 \right] \mathbb{G}_n^{(k-1)}(\mathcal{C}_Y h\, \mathbf{1}_{jk}),$$

*Proof.* We give the proof for $k$ odd. By (40) from Prop. 15 and by the definition of $\mathbb{G}_n^{(k)}(h)$, we need only show that $P(\mathcal{C}_X h \mathbf{1}_{jk}) = 0$. Indeed,

$$\mathbb{E}\left[ \mathcal{C}_X h \mathbf{1}_{jk} | X \right](x) = \begin{cases} \mathbb{E}\left[ \mathcal{C}_X h | X \right](x_j) & \text{if } x = x_j \\ 0 & \text{if } x \neq x_j \end{cases}.$$

But $\mathbb{E}\left[ \mathcal{C}_X h | X \right](x_j) = 0$ by definition of $\mathcal{C}_X$. Taking an expectation over $P_X$ gives that $P(\mathcal{C}_X h \mathbf{1}_{jk}) = 0$, which implies the desired result. The proof for $k$ even follows symmetrically. $\square$

The higher-order term in (37), can be bounded using Prop. 20.

**Proposition 20.** *For any $k \geq 1$, the following holds under the event $\mathcal{S}$:*

$$\sqrt{n} \sum_{\ell=1}^{k} |V_n^{(\ell-1)}(\mathcal{C}_\ell \ldots \mathcal{C}_k h)| \leq \sum_{j=1}^{m} \left( B_1 |\mathbb{G}_n^{(0)}(h_{1,k} \mathbf{1}_{j\ell})| + B_2 \sum_{\ell=2}^{k} |\mathbb{G}_n^{(0)}(h_{\ell,k} \mathbf{1}_{j\ell})| \right)$$
$$+ m B_2 \|h\|_\infty \sqrt{n} k(k-1)[B_1 + B_2(k+1)/3].$$

*Proof.* First, for any $\ell \in \{1, \ldots, k\}$, recall the notation $h_{\ell,k} := \mathcal{C}_\ell \ldots \mathcal{C}_k h$. By (39) from Prop. 15 and by Lem. 19, we have that for $\ell$ odd,

$$\sqrt{n} V_n^{(\ell-1)}(h_{\ell,k}) = \sum_{j=1}^{m} \left[ \frac{P_X}{P_{n,X}^{(\ell-1)}}(x_j) - 1 \right] \mathbb{G}_n^{(\ell-1)}(h_{\ell,k} \mathbf{1}_{j\ell}). \tag{44}$$

Using the statement above, we have that

$$\sqrt{n} |V_n^{(\ell-1)}(h_{\ell,k})| \leq M_\ell \sum_{j=1}^{m} |\mathbb{G}_n^{(\ell-1)}(h_{\ell,k} \mathbf{1}_{j\ell})|.$$

The bound above holds for $\ell$ even as well. Then, using the (39) from Prop. 15 again, we have that for $\ell \geq 2$,

$$[P_n^{(\ell-1)} - P](h_{\ell,k} \mathbf{1}_{j\ell}) = [P_n^{(\ell-2)} - P](h_{\ell,k} \mathbf{1}_{j\ell}) + V_n^{(\ell-2)}(h_{\ell,k} \mathbf{1}_{j\ell})$$

which implies that

$$\begin{aligned} |\mathbb{G}_n^{(\ell-1)}(h_{\ell,k} \mathbf{1}_{j\ell})| &\leq |\mathbb{G}_n^{(\ell-2)}(h_{\ell,k} \mathbf{1}_{j\ell})| + \sqrt{n} |V_n^{(\ell-2)}(h_{\ell,k} \mathbf{1}_{j\ell})| \\ &\leq |\mathbb{G}_n^{(0)}(h_{\ell,k} \mathbf{1}_{j\ell})| + \sqrt{n} |V_n^{(0)}(h_{\ell,k} \mathbf{1}_{j\ell})| + \ldots + \sqrt{n} |V_n^{(\ell-2)}(h_{\ell,k} \mathbf{1}_{j\ell})| \\ &\leq |\mathbb{G}_n^{(0)}(h_{\ell,k} \mathbf{1}_{j\ell})| + M_1 \sqrt{n} P_n^{(0)}(|h_{\ell,k}| \mathbf{1}_{j\ell}) + \ldots + M_\ell \sqrt{n} P_n^{(\ell-2)}(|h_{\ell,k}| \mathbf{1}_{j\ell}) \\ &\leq |\mathbb{G}_n^{(0)}(h_{\ell,k} \mathbf{1}_{j\ell})| + 2 \|h\|_\infty \sqrt{n} [B_1 + B_2(\ell-1)](k-\ell+1), \end{aligned} \tag{45}$$

by Lem. 18 and $M_1 \leq B_1$ and $M_\ell \leq B_2$ for $\ell \geq 2$. Summing these bounds, we have that

$$\sqrt{n} \sum_{\ell=1}^{k} |V_n^{(\ell-1)}(h_{\ell,k})|$$

$$\leq M_1 \sum_{j=1}^{m} |\mathbb{G}_n^{(0)}(h_{1,k}\mathbf{1}_{j\ell})| + \sum_{\ell=2}^{k} M_\ell \sum_{j=1}^{m} |\mathbb{G}_n^{(\ell-1)}(h_{\ell,k}\mathbf{1}_{j\ell})|$$

$$\leq B_1 \sum_{j=1}^{m} |\mathbb{G}_n^{(0)}(h_{1,k}\mathbf{1}_{j\ell})| + B_2 \sum_{\ell=2}^{k} \sum_{j=1}^{m} |\mathbb{G}_n^{(\ell-1)}(h_{\ell,k}\mathbf{1}_{j\ell})|$$

$$\leq B_1 \sum_{j=1}^{m} |\mathbb{G}_n^{(0)}(h_{1,k}\mathbf{1}_{j\ell})| \ +$$

$$B_2 \sum_{\ell=2}^{k} \sum_{j=1}^{m} \left( |\mathbb{G}_n^{(0)}(h_{\ell,k}\mathbf{1}_{j\ell})| + 2\|h\|_\infty \sqrt{n} \left[B_1 + B_2(\ell-1)\right](k-\ell+1)\right) \quad \text{apply (45)}$$

$$= \sum_{j=1}^{m} \left( B_1 |\mathbb{G}_n^{(0)}(h_{1,k}\mathbf{1}_{j\ell})| + B_2 \sum_{\ell=2}^{k} |\mathbb{G}_n^{(0)}(h_{\ell,k}\mathbf{1}_{j\ell})| \right) \ +$$

$$2mB_2 \|h\|_\infty \sqrt{n} \sum_{\ell=2}^{k} \left[B_1 + B_2(\ell-1)\right](k-\ell+1),$$

because $|\mathcal{X}| = m$. We sum up the last term:

$$\sum_{\ell=2}^{k} \left[B_1 + B_2(\ell-1)\right](k-\ell+1) = B_1 \sum_{\ell=1}^{k-1}(k-\ell) + B_2 \sum_{\ell=1}^{k-1} \ell(k-\ell)$$

$$= \frac{k(k-1)}{2} \left[B_1 + B_2(k+1)/3\right].$$

completing the proof. $\qquad \square$

### D.4 Proof of Main Results

We can now show the main result of this section: the bound on the mean squared error of the rebalanced estimator. Recall the event
$$\mathcal{S} := \{\text{Supp}(P_{n,X}) = \text{Supp}(P_X) \text{ and } \text{Supp}(P_{n,Y}) = \text{Supp}(P_Y)\} \qquad (46)$$
as introduced in (35). To remind the reader of the high-level steps of the proof, we may decompose the error on the event $\mathcal{S}$ we used the estimator
$$\tilde{\varphi}_n^{(k)} := \varphi_n^{(k)}\mathbb{1}_{\mathcal{S}} + \varphi_n^{(0)}\mathbb{1}_{\mathcal{S}^c}$$
so we decompose on the event $\mathcal{S}$ to write
$$\mathbb{E}_P\left[\left(\tilde{P}_n^{(k)}(h) - P(h)\right)^2\right] = \mathbb{E}_P\left[(P_n(h) - P(h))^2 \mathbf{1}_{\mathcal{S}^c}\right] + \mathbb{E}_P\left[(P_n^{(k)}(h) - P(h))^2 \mathbf{1}_{\mathcal{S}}\right]. \qquad (47)$$

Then, we use the upcoming Prop. 21 to bound the first term, which will in turn require showing that $\mathcal{S}$ occurs with high probability. As for the second term, we will apply Prop. 15 and the derivation (37) to write
$$\mathbb{E}_P\left[(P_n^{(k)}(h) - P(h))^2 \mathbf{1}_{\mathcal{S}}\right] = \mathbb{E}_P\left[T_1^2 \mathbf{1}_{\mathcal{S}}\right] + 2\mathbb{E}_P\left[T_1 T_2 \mathbf{1}_{\mathcal{S}}\right] + \mathbb{E}_P\left[T_2^2 \mathbf{1}_{\mathcal{S}}\right] \qquad (48)$$
for
$$T_1 := [P_n^{(0)} - P](\mathcal{C}_1 \ldots \mathcal{C}_k h) \text{ and } T_2 := \sum_{\ell=1}^{k} V_n^{(\ell-1)}(\mathcal{C}_\ell \ldots \mathcal{C}_k h). \qquad (49)$$

By definition, we have that $\mathbb{E}_P\left[T_1^2 \mathbf{1}_{\mathcal{S}}\right] \leq \mathbb{E}_P\left[T_1^2\right] = \sigma_k^2/n$. It then remains to bound the cross term $\mathbb{E}_P\left[T_1 T_2 \mathbf{1}_{\mathcal{S}}\right]$ and squared term $\mathbb{E}_P\left[T_2^2 \mathbf{1}_{\mathcal{S}}\right]$. This is accomplished by Lem. 23 and Lem. 22, respectively.

**Proposition 21.** *It holds that $P(\mathcal{S}^c) \leq 2m(1 - p_\star)^n$. Moreover, for any $\delta \in (0,1)$, we have*

$$\mathbb{E}_P\left[(P_n(h) - P(h))^2 \mathbf{1}_{\mathcal{S}^c}\right] \leq 4\,\|h\|_\infty^2 \min\left\{2m(1-p_\star)^n, \delta\right\} + \frac{2\log(2/\delta)}{n}\,\|h\|_\infty^2\, 2m(1-p_\star)^n.$$

*Proof.* Define $\mathcal{F}_X := \{\mathrm{Supp}(P_{n,X}) \neq \mathrm{Supp}(P_X)\}$ and $\mathcal{F}_Y := \{\mathrm{Supp}(P_{n,Y}) \neq \mathrm{Supp}(P_Y)\}$, so that $\mathcal{S}^c = \mathcal{F}_X \cup \mathcal{F}_Y$. We first control the probability of $\mathcal{F}_X$. Let $F_j := \{P_{n,X}(x_j) = 0\}$ for $j \in [m]$. We then obtain $\mathcal{F}_X = \cup_{j=1}^m F_j$, which implies by the union bound that

$$P(\mathcal{F}_X) \leq \sum_{j=1}^m P(F_j) = \sum_{j=1}^m (1 - P_X(x_j))^n \leq m(1 - p_\star)^n.$$

Similarly, we have that $P(\mathcal{F}_Y) \leq m(1 - p_\star)^n$ and thus $P(\mathcal{S}^c) \leq 2m(1 - p_\star)^n$, which gives the first claim.

To control the expectation, consider any $\delta > 0$, and define the event

$$\mathcal{E}_\delta := \left\{\left|P_n^{(0)}(h) - P(h)\right| \leq \sqrt{\frac{2\log(2/\delta)}{n}}\,\|h\|_\infty\right\}.$$

By Hoeffding's inequality, it holds that $P(\mathcal{E}_\delta) \geq 1 - \delta$. Furthermore, we get

$$\mathbb{E}[\mathbf{1}_{\mathcal{S}^c}(P_n^{(0)}(h) - P(h))^2] = \mathbb{E}[\mathbf{1}_{\mathcal{S}^c}\mathbf{1}_{\mathcal{E}_\delta^c}(P_n^{(0)}(h) - P(h))^2] + \mathbb{E}[\mathbf{1}_{\mathcal{S}^c}\mathbf{1}_{\mathcal{E}_\delta}(P_n^{(0)}(h) - P(h))^2]$$

$$\leq 4\,\|h\|_\infty^2\,\mathbb{E}[\mathbf{1}_{\mathcal{S}^c}\mathbf{1}_{\mathcal{E}_\delta^c}] + \frac{2\log(2/\delta)}{n}\,\|h\|_\infty^2\,\mathbb{E}[\mathbf{1}_{\mathcal{S}^c}\mathbf{1}_{\mathcal{E}_\delta}]$$

$$\leq 4\,\|h\|_\infty^2\,\min\{P(\mathcal{S}^c), P(\mathcal{E}_\delta^c)\} + \frac{2\log(2/\delta)}{n}\,\|h\|_\infty^2\,P(\mathcal{S}^c)$$

$$\leq 4\,\|h\|_\infty^2\,\min\{2m(1-p_\star)^n, \delta\} + \frac{2\log(2/\delta)}{n}\,\|h\|_\infty^2\, 2m(1-p_\star)^n.$$

$\square$

In order to bound the terms appearing in (48), we introduce the events $\mathcal{E}_1^\delta$, $\mathcal{E}_2^\delta$, and $\mathcal{E}_3^\delta$, defined by

$$\mathcal{E}_1^\delta := \left\{\max\left\{\mathrm{KL}(P_{n,X}\|P_X), \mathrm{KL}(P_{n,Y}\|P_Y)\right\} \leq \frac{1}{n}\log_2\frac{2}{\delta} + m\frac{\log(n+1)}{n}\right\}$$

$$\mathcal{F}_\ell^\delta := \left\{|\mathbb{G}_n^{(0)}(h_{\ell,k}\,\mathbf{1}_{j\ell})| \leq \sqrt{2\log(2mk/\delta)}\,2(k-\ell+1)\,\|h\|_\infty\right\}, \quad \ell = 1, \ldots, k, \ j = 1, \ldots, m$$

$$\mathcal{E}_2^\delta := \bigcap_{\ell=1}^k \mathcal{F}_\ell^\delta$$

$$\mathcal{E}_3^\delta := \left\{|\mathbb{G}_n^{(0)}(h_{1,k})| \leq \sqrt{2\log(2/\delta)}\,2k\,\|h\|_\infty\right\}.$$

The events are constructed such that $\mathbb{P}(\mathcal{E}_1^\delta) \geq 1 - \delta$, $\mathbb{P}(\mathcal{E}_2^\delta) \geq 1 - \delta$, and $\mathbb{P}(\mathcal{E}_3^\delta) \geq 1 - \delta$, as we used in the upcoming proofs of Lem. 23, Lem. 22, and Thm. 24.

**Lemma 22** (Squared term bound). *Let $T_2$ be defined as in (49). For any $\delta > 0$, assuming that $n \geq 2[\log_2(2/\delta) + m\log(n+1)]/p_\star^2$, we have that*

$$\mathbb{E}_P\left[T_2^2 \mathbf{1}_{\mathcal{S}}\right] \leq \frac{2\,\|h\|_\infty^2\, m^2 k^2}{p_\star^2}\left[\log_2(2/\delta) + m\log(n+1)\right]^{2 - \mathbf{1}\{k=1\}} \times$$

$$\left[\left(4n + \frac{k-1}{p_\star^2}\left(n + 2 + \frac{k+1}{p_\star^2}\right)\right)^2 \delta + \frac{8}{n^2}\left(\sqrt{2\log\frac{2mk}{\delta}}(k+1) + \frac{(k-1)(k+4)}{p_\star^2}\right)^2\right].$$

*Proof.* The following computations are done under the event $\mathcal{S}$. First, apply Prop. 20 to write

$$|T_2| \leq \frac{1}{\sqrt{n}}\sum_{j=1}^m\left(B_1\,|\mathbb{G}_n^{(0)}(h_{1,k}\,\mathbf{1}_{j\ell})| + B_2\sum_{\ell=2}^k |\mathbb{G}_n^{(0)}(h_{\ell,k}\,\mathbf{1}_{j\ell})|\right) +$$

$$mB_2\,\|h\|_\infty\,k(k-1)[B_1 + B_2(k+1)/3]. \tag{50}$$

We decompose on the event $\mathcal{E}_1^\delta \cap \mathcal{E}_2^\delta$. Note that by Thm. 16, we have that $\mathbb{P}(\mathcal{E}_1^\delta) \geq 1 - \delta$. It follows from Hoeffding's inequality, the union bound, and boundedness of $\|h_{\ell,k}\,\mathbf{1}_{j\ell}\|$ by Lem. 18 that $\mathbb{P}(\mathcal{E}_2^\delta) \geq 1 - \delta$ As a result, $\mathbb{P}(\mathcal{E}_1^\delta \cap \mathcal{E}_2^\delta) \geq 1 - 2\delta$.

**Bound $|T_2|$ under the event $\mathcal{S}\setminus(\mathcal{E}_1^\delta \cap \mathcal{E}_2^\delta)$.** In this case, we apply (32) from Prop. 14 to get $B_1 \leq n$ and $B_2 \leq 1/p_\star^2$, along with the universal bounds from Lem. 18:

$$\frac{1}{\sqrt{n}}\,|\mathbb{G}_n^{(0)}(h_{1,k}\,\mathbf{1}_{j\ell})| \leq 2\,\|h_{1,k}\|_\infty \leq 4k\,\|h\|_\infty$$

$$\frac{1}{\sqrt{n}}\sum_{\ell=2}^{k}|\mathbb{G}_n^{(0)}(h_{\ell,k}\,\mathbf{1}_{j\ell})| \leq 2\sum_{\ell=2}^{k}\|h_{\ell,k}\|_\infty \leq \sum_{\ell=2}^{k}4(k-\ell+1)\,\|h\|_\infty = 2k(k-1)\,\|h\|_\infty$$

so that by plugging into (50),

$$|T_2| \leq \|h\|_\infty\, mk\left[4n + \frac{k-1}{p_\star^2}\left(n + 2 + \frac{k+1}{3p_\star^2}\right)\right],$$

and in turn,

$$\mathbb{E}_P\left[T_2^2 \mathbb{1}_{\mathcal{S}\setminus(\mathcal{E}_1^\delta \cap \mathcal{E}_2^\delta)}\right] \leq 2\,\|h\|_\infty^2\, m^2 k^2\left[4n + \frac{k-1}{p_\star^2}\left(n + 2 + \frac{k+1}{3p_\star^2}\right)\right]^2 \delta. \tag{51}$$

**Bound $|T_2|$ under the event $\mathcal{S} \cap \mathcal{E}_1^\delta \cap \mathcal{E}_2^\delta$.** In this case, we may use that $n \geq 2[\log_2(2/\delta) + m\log(n+1)]/p_\star^2$ apply (33) from Prop. 14 to get

$$\max\{B_1, B_2\} \leq \frac{2}{p_\star}\sqrt{\frac{1}{2}\,\mathrm{KL}(P_{n,X}\|P_X)} \leq \frac{1}{p_\star\sqrt{n}}\sqrt{2\log_2(2/\delta) + 2m\log(n+1)}$$

and the bounds based on $\mathcal{E}_2^\delta$ which give

$$|\mathbb{G}_n^{(0)}(h_{1,k}\,\mathbf{1}_{j\ell})| \leq \sqrt{2\log\frac{2mk}{\delta}}\,2k\,\|h\|_\infty$$

$$\sum_{\ell=2}^{k}|\mathbb{G}_n^{(0)}(h_{\ell,k}\,\mathbf{1}_{j\ell})| \leq \sum_{\ell=2}^{k}\sqrt{2\log\frac{2mk}{\delta}}\,2(k-\ell+1)\,\|h\|_\infty \leq \sqrt{2\log\frac{2mk}{\delta}}\,k(k-1)\,\|h\|_\infty,$$

By plugging into (50),

$$|T_2| \leq \frac{2m\,\|h\|_\infty\,\sqrt{2\log(2mk/\delta)\,[2\log_2(2/\delta) + 2m\log(n+1)]}}{np_\star}k(k+1) + \tag{52}$$

$$\frac{m\,\|h\|_\infty\,[2\log_2(2/\delta) + 2m\log(n+1)]}{3np_\star^2}k(k-1)(k+4) \tag{53}$$

$$\leq \frac{4mk\,\|h\|_\infty\,[\log_2(2/\delta) + 2m\log(n+1)]^{1-\mathbb{1}\{k=1\}/2}}{np_\star^2} \times \tag{54}$$

$$\left[p_\star\sqrt{2\log(2mk/\delta)}(k+1) + (k-1)(k+4)\right]. \tag{55}$$

In turn,

$$\mathbb{E}_P\left[T_2^2 \mathbb{1}_{\mathcal{S}\setminus(\mathcal{E}_1^\delta \cap \mathcal{E}_2^\delta)}\right] \leq \frac{16\,\|h\|_\infty^2\, m^2 k^2\,[\log_2(2/\delta) + m\log(n+1)]^{2-\mathbb{1}\{k=1\}}}{n^2 p_\star^4} \times$$

$$\left[p_\star\sqrt{2\log(2mk/\delta)}(k+1) + (k-1)(k+4)\right]^2. \tag{56}$$

Combining together both (56) and (51) and using that $[\log_2(2/\delta) + 2m\log(n+1)] \geq 1$, we have that

$$\mathbb{E}_P\left[T_2^2\mathbb{1}_{\mathcal{S}}\right] \leq \frac{2\,\|h\|_\infty^2\, m^2 k^2}{p_\star^2}\,[\log_2(2/\delta) + m\log(n+1)]^{2-\mathbb{1}\{k=1\}} \times$$

$$\left[\left(4n + \frac{k-1}{p_\star^2}\left(n + 2 + \frac{k+1}{p_\star^2}\right)\right)^2 \delta + \frac{8}{n^2}\left(\sqrt{2\log(2mk/\delta)}(k+1) + \frac{(k-1)(k+4)}{p_\star^2}\right)^2\right],$$

the result as desired. $\qquad\square$

**Lemma 23** (Cross term bound). *Let $T_1$ and $T_2$ be defined as in (49). For any $\delta > 0$, assuming that $n \geq 2[\log_2(2/\delta) + m\log(n+1)]/p_\star^2$, we have that*

$$\mathbb{E}_P\left[T_1 T_2 \mathbb{1}_{\mathcal{S}}\right]$$

$$\leq \frac{2mk^2 \|h\|_\infty^2 \sqrt{2\log(2/\delta)} \left[\log_2(2/\delta) + 2m\log(n+1)\right]^{1-\mathbb{1}\{k=1\}/2}}{p_\star^2} \times$$

$$\left[\frac{p_\star \sqrt{2\log(2mk/\delta)}(k+1) + (k-1)(k+4)}{n^{3/2}} + 6\left(4np_\star^2 + (k-1)\left(n+2+\frac{k+1}{p_\star^2}\right)\right)\delta\right],$$

*Proof.* The following computations are done under the event $\mathcal{S}$. First, apply Prop. 20 to write

$$|T_1 T_2| \leq \frac{1}{\sqrt{n}} |\mathbb{G}_n^{(0)}(h_{1,k})| \left[\frac{1}{\sqrt{n}} \sum_{j=1}^m \left(B_1 |\mathbb{G}_n^{(0)}(h_{1,k}\,\mathbf{1}_{j\ell})| + B_2 \sum_{\ell=2}^k |\mathbb{G}_n^{(0)}(h_{\ell,k}\,\mathbf{1}_{j\ell})|\right) + \right.$$

$$\left. mB_2 \|h\|_\infty k(k-1)[B_1 + B_2(k+1)/3]\right]. \tag{57}$$

We decompose on the event $\mathcal{E}_1^\delta \cap \mathcal{E}_2^\delta \cap \mathcal{E}_3^\delta$. Note that by Thm. 16 and that $n \geq \log_2(2/\delta) + m\log(n+1)$, we have that $\mathbb{P}(\mathcal{E}_1^\delta) \geq 1 - \delta$. It follows by Hoeffding's inequality and the union bound that $\mathbb{P}(\mathcal{E}_2^\delta) \geq 1 - \delta$. Similarly, we also have by Hoeffding's inequality that $\mathbb{P}(\mathcal{E}_3^\delta) \geq 1 - \delta$. As a result, $\mathbb{P}(\mathcal{E}_1^\delta \cap \mathcal{E}_2^\delta \cap \mathcal{E}_3^\delta) \geq 1 - 3\delta$.

**Bound $|T_1 T_2|$ under the event** $\mathcal{S} \setminus (\mathcal{E}_1^\delta \cap \mathcal{E}_2^\delta \cap \mathcal{E}_3^\delta)$. In this case, we apply (32) from Prop. 14 to get $B_1 \leq n$ and $B_2 \leq 1/p_\star^2$, along with the universal bounds from Lem. 18:

$$\frac{1}{\sqrt{n}} |\mathbb{G}_n^{(0)}(h_{1,k})| \leq 2\|h_{1,k}\|_\infty \leq 4k\|h\|_\infty$$

$$\frac{1}{\sqrt{n}} |\mathbb{G}_n^{(0)}(h_{1,k}\,\mathbf{1}_{j\ell})| \leq 2\|h_{1,k}\|_\infty \leq 4k\|h\|_\infty$$

$$\frac{1}{\sqrt{n}} \sum_{\ell=2}^k |\mathbb{G}_n^{(0)}(h_{\ell,k}\,\mathbf{1}_{j\ell})| \leq 2\sum_{\ell=2}^k \|h_{\ell,k}\|_\infty \leq \sum_{\ell=2}^k 4(k-\ell+1)\|h\|_\infty = 2k(k-1)\|h\|_\infty,$$

so that by plugging into (57),

$$|T_1 T_2| \leq 4k^2 \|h\|_\infty^2 m\left[4n + \frac{k-1}{p_\star^2}\left(n+2+\frac{k+1}{3p_\star^2}\right)\right],$$

and in turn,

$$\mathbb{E}_P\left[T_1 T_2 \mathbb{1}_{\mathcal{S}\setminus(\mathcal{E}_1^\delta \cap \mathcal{E}_2^\delta \cap \mathcal{E}_3^\delta)}\right] \leq \frac{12k^2 \|h\|_\infty^2 m}{p_\star^2}\left[4np_\star^2 + (k-1)\left(n+2+\frac{k+1}{3p_\star^2}\right)\right]\delta. \tag{58}$$

**Bound $|T_1 T_2|$ under the event** $\mathcal{S} \cap \mathcal{E}_1^\delta \cap \mathcal{E}_2^\delta \cap \mathcal{E}_3^\delta$. In this case, we may use that $n \geq 2[\log_2(2/\delta) + m\log(n+1)]/p_\star^2$ apply (33) from Prop. 14 to get

$$\max\{B_1, B_2\} \leq \frac{2}{p_\star}\sqrt{\frac{1}{2}\mathrm{KL}(P_{n,X}\|P_X)} \leq \frac{1}{\sqrt{n}}\frac{1}{p_\star}\sqrt{2\log_2(2/\delta) + 2m\log(n+1)}$$

and the bounds based on $\mathcal{E}_2^\delta \cap \mathcal{E}_2^\delta \cap \mathcal{E}_3^\delta$ which give

$$|\mathbb{G}_n^{(0)}(h_{1,k})| \leq \sqrt{2\log(2/\delta)}2k\|h\|_\infty$$

$$|\mathbb{G}_n^{(0)}(h_{1,k}\,\mathbf{1}_{j\ell})| \leq \sqrt{2\log(2mk/\delta)}2k\|h\|_\infty$$

$$\sum_{\ell=2}^k |\mathbb{G}_n^{(0)}(h_{\ell,k}\,\mathbf{1}_{j\ell})| \leq \sum_{\ell=2}^k \sqrt{2\log\frac{2mk}{\delta}}2(k-\ell+1)\|h\|_\infty \leq \sqrt{2\log\frac{2mk}{\delta}}k(k-1)\|h\|_\infty,$$

By plugging into (57),

$$|T_2| \leq \frac{m \, \|h\|_\infty \sqrt{2\log(2mk/\delta)} \, [2\log_2(2/\delta) + 2m\log(n+1)]}{np_\star} k(k+1) \, +$$

$$\frac{m \, \|h\|_\infty \, [2\log_2(2/\delta) + 2m\log(n+1)]}{3np_\star^2} k(k-1)(k+4)$$

$$\leq \frac{mk \, \|h\|_\infty \, [\log_2(2/\delta) + 2m\log(n+1)]^{1-\mathbb{1}\{k=1\}/2}}{np_\star^2} \times$$

$$\left[ p_\star \sqrt{2\log(2mk/\delta)}(k+1) + (k-1)(k+4) \right]$$

$$|T_1 T_2| \leq \frac{2mk^2 \, \|h\|_\infty^2 \sqrt{2\log(2/\delta)} \, [\log_2(2/\delta) + 2m\log(n+1)]^{1-\mathbb{1}\{k=1\}/2}}{n^{3/2}p_\star^2} \times$$

$$\left[ p_\star \sqrt{2\log(2mk/\delta)}(k+1) + (k-1)(k+4) \right],$$

In turn,

$$\mathbb{E}_P \left[ T_2^2 \mathbb{1}_{\mathcal{S} \setminus (\mathcal{E}_1^\delta \cap \mathcal{E}_2^\delta \cap \mathcal{E}_3^\delta)} \right] \leq \frac{2mk^2 \, \|h\|_\infty^2 \sqrt{2\log(2/\delta)} \, [\log_2(2/\delta) + 2m\log(n+1)]^{1-\mathbb{1}\{k=1\}/2}}{n^{3/2}p_\star^2} \times$$

$$\left[ p_\star \sqrt{2\log(2mk/\delta)}(k+1) + (k-1)(k+4) \right], \tag{59}$$

Combining together both (59) and (58) and using that $[\log_2(2/\delta) + 2m\log(n+1)] \geq 1$, we have that

$$\mathbb{E}_P \left[ T_1 T_2 \mathbb{1}_{\mathcal{S}} \right]$$

$$\leq \frac{2mk^2 \, \|h\|_\infty^2 \sqrt{2\log(2/\delta)} \, [\log_2(2/\delta) + 2m\log(n+1)]^{1-\mathbb{1}\{k=1\}/2}}{p_\star^2} \times$$

$$\left[ \frac{p_\star \sqrt{2\log(2mk/\delta)}(k+1) + (k-1)(k+4)}{n^{3/2}} + 6\left(4np_\star^2 + (k-1)\left(n+2+\frac{k+1}{p_\star^2}\right)\right)\delta \right],$$

the result as desired. $\qquad \square$

We now combine the previous results to prove Thm. 24.

**Theorem 24.** *For a sequence of rebalanced distributions* $(\tilde{P}_n^{(k)})_{k \geq 1}$, *there exists an absolute constant* $C > 0$ *such that when* $n \geq C[\log_2(2n/p_\star) + m\log(n+1)]/p_\star^2$,

$$\mathbb{E}_P[(\tilde{P}_n^{(k)}(h) - P(h))^2] \leq \frac{\sigma_k^2}{n} + \frac{CB}{n^{3/2}}, \tag{60}$$

*where*

$$B = \frac{\sqrt{\log(2n/p_\star)}m^2 k^4 \, \|h\|_\infty^2}{p_\star^2} \left( \log_2 \frac{2n}{p_\star} + m\log(n+1) \right)^{2-\mathbb{1}\{k\}} \left( \log \frac{2mkn}{p_\star} + \frac{(k-1)^2}{p_\star^2} \right).$$

*Proof.* We apply the decomposition (47), and subsequently handle the second term using bounds on the terms in (48). Set $\delta = p_\star^4/n^4$. We apply Lem. 22 and Lem. 23 with this choice of $\delta$, so that there exists an absolute constants $\tilde{C}$, $C_1$, and $C_2$ such that

$$\mathbb{E}_P \left[ T_1 T_2 \mathbb{1}_{\mathcal{S}} \right] \leq C_1 \frac{\|h\|_\infty^2 m^2 k^3 \sqrt{\log(2n/p_\star)}}{n^{3/2}p_\star^2} \, [\log_2(2n/p_\star) + m\log(n+1)]^{1-\mathbb{1}\{k=1\}/2} \times$$

$$\left( \log \frac{2mnk}{p_\star} + \frac{k-1}{p_\star^2} \right)$$

$$\mathbb{E}_P \left[ T_2^2 \mathbb{1}_{\mathcal{S}} \right] \leq C_2 \frac{\|h\|_\infty^2 m^2 k^4}{n^2 p_\star^2} \, [\log_2(2n/p_\star) + m\log(n+1)]^{2-\mathbb{1}\{k=1\}} \times$$

$$\left( \log \frac{2mnk}{p_\star} + \frac{(k-1)^2}{p_\star^2} \right),$$

when $n \geq \tilde{C}[\log_2(2n/p_\star) + m \log(n+1)]/p_\star^2$. This then implies that there is an absolute constant $C_3$ such that

$$\mathbb{E}_P \left[ \left( \tilde{P}_n^{(k)}(h) - P(h) \right)^2 \right]$$

$$\leq \mathbb{E}_P \left[ (P_n^{(0)}(h) - P(h))^2 \, \mathbb{1}_{\mathcal{S}^c} \right] + \frac{\sigma_k^2}{n} +$$

$$\frac{C_3 \, \|h\|_\infty^2 \, m^2 k^4 \sqrt{\log(2n/p_\star)}}{n^{3/2} p_\star^2} \left[ \log_2 \frac{2n}{p_\star} + m \log(n+1) \right]^{2 - \mathbb{1}\{k=1\}} \left( \log \frac{2mnk}{p_\star} + \frac{(k-1)^2}{p_\star^2} \right).$$

Next, we apply Prop. 21 with the same choice of $\delta$. Because $2[\log_2(2/\delta) + m \log(n+1)] \geq \log(m/\delta)$ and $-\log(1 - p_\star) \geq p_\star \geq p_\star^2$, we have that $n \geq \log(\delta/m)/\log(1 - p_\star)$, which implies that $m(1 - p_\star)^n \leq \delta$. Combining with the display above, we have that there exists an absolute constant $C > 0$ such that

$$\mathbb{E}_P \left[ \left( \tilde{P}_n^{(k)}(h) - P(h) \right)^2 \right] \leq \frac{\sigma_k^2}{n} + \frac{C \, \|h\|_\infty^2 \, m^2 k^4 \sqrt{\log(2n/p_\star)}}{n^{3/2} p_\star^2}$$

$$\times \, [\log_2(2/\delta) + m \log(n+1)]^{2 - \mathbb{1}\{k=1\}} \left( \log \frac{2mnk}{p_\star} + \frac{(k-1)^2}{p_\star^2} \right),$$

which is the claimed result. $\qquad\square$

While not shown in the main text, similar techniques to those used above can also control the bias of $\tilde{P}_n^{(k)}(h)$ as in Thm. 25. Interestingly, this bias is of order $O(n^{-2})$ which confirms the intuition that even thought $\tilde{P}_n^{(k)}(h)$ may be biased, the dominant term is the variance.

**Theorem 25.** *For a sequence of rebalanced distributions $(P^{(k)})_{k \geq 1}$, there exists an absolute constant $C > 0$ such that when $n \geq C[\log_2(2n/p_\star) + m \log(n+1)]/p_\star^2$,*

$$\left| \mathbb{E}_P[\tilde{P}_n^{(k)}(h) - P(h)] \right|^2 \leq \frac{CB}{n^2}, \tag{61}$$

*where $B$ is as defined in Thm. 24.*

*Proof.* First, apply the decomposition (47) so that

$$\left| \mathbb{E}_P \left[ \tilde{P}_n^{(k)}(h) - P(h) \right] \right| \leq |\mathbb{E}_P \left[ (P_n(h) - P(h)) \, \mathbb{1}_{\mathcal{S}^c} \right]| + |\mathbb{E}_P \left[ (P_n^{(k)}(h) - P(h)) \, \mathbb{1}_{\mathcal{S}} \right]|.$$

By using the argument of Prop. 21, we have that

$$|\mathbb{E}_P \left[ P_n(h) - P(h) \right] \mathbb{1}_{\mathcal{S}^c}| \leq 2 \, \|h\|_\infty \min \{ 2m(1 - p_\star)^n, \delta \} + \sqrt{\frac{2 \log(2/\delta)}{n}} \, \|h\|_\infty \, 2m(1 - p_\star)^n.$$

Then, by the recursion formula Equation (37), we have that

$$\sqrt{n} \, |\mathbb{E}_P \left[ (P_n^{(k)}(h) - P(h)) \, \mathbb{1}_{\mathcal{S}} \right]|$$

$$= |\mathbb{E}_P \left[ \mathbb{G}_n^{(k)}(h) \mathbb{1}_{\mathcal{S}} \right]| = \left| \mathbb{E}_P \left[ (1 - \mathbb{1}_{\mathcal{S}^c}) \mathbb{G}_n^{(0)}(\mathcal{C}_1 \dots \mathcal{C}_k h) + \sqrt{n} \mathbb{1}_{\mathcal{S}} \sum_{\ell=1}^k V_n^{(\ell-1)}(\mathcal{C}_\ell \dots \mathcal{C}_k h) \right] \right|.$$

Because $\mathbb{G}_n^{(0)}(\mathcal{C}_1 \dots \mathcal{C}_k h)$ has zero mean, it follows that

$$\sqrt{n} \, |\mathbb{E}_P \left[ (P_n^{(k)}(h) - P(h)) \, \mathbb{1}_{\mathcal{S}} \right]| \leq |\mathbb{E}_P \left[ \mathbb{1}_{\mathcal{S}^c} \mathbb{G}_n^{(0)}(\mathcal{C}_1 \dots \mathcal{C}_k h) \right]| + \sqrt{n} \, |\mathbb{E}_P \left[ \mathbb{1}_{\mathcal{S}} T_2 \right]|$$

We have by Hoeffding's inequality that $\mathbb{P}(\mathcal{E}_3^\delta) \geq 1 - \delta$, and that by Lem. 18 that $\mathbb{G}_n^{(0)}(\mathcal{C}_1 \dots \mathcal{C}_k h) \leq 4k\sqrt{n} \, \|h\|_\infty$ universally. As a result, applying Prop. 21 once again,

$$|\mathbb{E}_P \left[ \mathbb{1}_{\mathcal{S}^c} \mathbb{G}_n^{(0)}(\mathcal{C}_1 \dots \mathcal{C}_k h) \right]|$$

$$\leq \left| \mathbb{E}_P \left[ \mathbb{1}_{\mathcal{S}^c} \mathbb{1}_{\mathcal{E}_3^\delta} \mathbb{G}_n^{(0)}(\mathcal{C}_1 \dots \mathcal{C}_k h) \right] \right| + \left| \mathbb{E}_P \left[ \mathbb{1}_{\mathcal{S}^c} \mathbb{1}_{\mathcal{E}_3^\delta} \mathbb{G}_n^{(0)}(\mathcal{C}_1 \dots \mathcal{C}_k h) \right] \right|$$

$$\leq 4k\sqrt{n} \, \|h\|_\infty \min \{ 2m(1 - p_\star)^n, \delta \} + \sqrt{2 \log(2/\delta)} 2k \, \|h\|_\infty \, 2m(1 - p_\star)^n.$$

Using a similar argument to Lem. 22, we have that under $\mathcal{S}\backslash(\mathcal{E}_1^\delta \cap \mathcal{E}_2^\delta)$ (which occurs with probability no more than $2\delta$),

$$|T_2| \leq \|h\|_\infty mk \left[ 4n + \frac{k-1}{p_\star^2} \left( n + 2 + \frac{k+1}{3p_\star^2} \right) \right],$$

and that under $\mathcal{S} \cap \mathcal{E}_1^\delta \cap \mathcal{E}_2^\delta$ (which occurs with probability at least $1 - 2\delta$),

$$|T_2| \leq \frac{4mk \|h\|_\infty \left[ \log_2(2/\delta) + 2m \log(n+1) \right]^{1-\mathbb{1}\{k=1\}/2}}{np_\star^2}$$
$$\left[ p_\star \sqrt{2\log(2mk/\delta)}(k+1) + (k-1)(k+4) \right].$$

Applying the decomposition $|\mathbb{E}_P \left[ \mathbb{1}_\mathcal{S} T_2 \right]| \leq \left| \mathbb{E}_P \left[ \mathbb{1}_{\mathcal{S}\backslash(\mathcal{E}_1^\delta \cap \mathcal{E}_2^\delta)} T_2 \right] \right| + \left| \mathbb{E}_P \left[ \mathbb{1}_{\mathcal{S} \cap \mathcal{E}_1^\delta \cap \mathcal{E}_2^\delta} T_2 \right] \right|$ and setting $\delta = \frac{p_\star^2}{n^2}$ achieves the desired result. $\qquad\square$

### D.5  Misspecified Marginal Distributions

We now adapt the main results to cases in which the marginal distributions $(P_X, P_Y)$ are misspecified, in that the user is provided marginal distributions $(\hat{P}_{X,\epsilon}, \hat{P}_{Y,\epsilon})$ which satisfy the following structure.

**Assumption 1.** *There exist fixed probability mass functions $\hat{P}_X$ and $\hat{P}_Y$ for some $\epsilon \in [0,1)$,*

$$\hat{P}_{X,\epsilon} = (1-\epsilon)P_X + \epsilon\hat{P}_X \text{ and } \hat{P}_{Y,\epsilon} = (1-\epsilon)P_Y + \epsilon\hat{P}_Y.$$

*We also have the existence of the positive quantity*

$$\hat{p}_\star := \min\{\min_x \hat{P}_X(x), \min_y \hat{P}_Y(y)\} > 0.$$

Given the existence of $\hat{p}_\star > 0$, we may also define

$$\hat{p}_{\star,\epsilon} = \min\{\min_x \hat{P}_{X,\epsilon}(x), \min_y \hat{P}_{Y,\epsilon}(y)\} \geq \epsilon\hat{p}_\star + (1-\epsilon)p_\star > 0.$$

To be precise, the iterations of balancing follow $\hat{P}_n^{(0)} = P_n$ and

$$\hat{P}_n^{(k)}(x,y) := \begin{cases} \frac{\hat{P}_{X,\epsilon}(x)}{\hat{P}_{n,X}^{(k-1)}(x)} \cdot \hat{P}_n^{(k-1)}(x,y) & k \text{ odd} \\ \frac{\hat{P}_{Y,\epsilon}(y)}{\hat{P}_{n,Y}^{(k-1)}(y)} \cdot \hat{P}_n^{(k-1)}(x,y) & k \text{ even} \end{cases}. \tag{62}$$

We start by deriving a result similar to Prop. 15. Since $\epsilon < 1$, the (possibly misspecified) target marginals $\hat{P}_{X,\epsilon}(x) > 0$ and $\hat{P}_{Y,\epsilon}(y) > 0$ for all $x \in \mathcal{X}$ and $y \in \mathcal{Y}$. Define

$$\hat{V}_n^{(k-1)}(h) := \begin{cases} \sum_{x,y} \left( \frac{\hat{P}_{X,\epsilon}}{\hat{P}_{n,X}^{(k-1)}}(x) - 1 \right) h(x,y)\hat{P}_n^{(k-1)}(x,y) & k \text{ odd} \\ \sum_{x,y} \left( \frac{\hat{P}_{Y,\epsilon}}{\hat{P}_{n,Y}^{(k-1)}}(y) - 1 \right) h(x,y)\hat{P}_n^{(k-1)}(x,y) & k \text{ even} \end{cases} \tag{63}$$

and

$$\hat{\mathbb{G}}_n^{(k)}(h) := \sqrt{n} \left( \hat{P}_n^{(k)}(h) - P(h) \right).$$

The format of this section will be to derive results analogous to the building blocks of the previous section. From that point, the computations from Appx. D.4 will achieve the desired result. For the sake of comparison to Thm. 1 we consider error terms containing $\epsilon$ only by their dependence on $(\epsilon, k, n, \hat{p}_{\star,\epsilon})$.

### D.5.1 Intermediate Results

**Proposition 26.** *Let $(\hat{P}_n^{(k)})_{k\geq 1}$ be a sequence computed according to (62). Define*

$$c^2 = \max\left\{\chi^2(\hat{P}_X\|P_X), \chi^2(\hat{P}_Y\|P_Y)\right\}.$$

*These iterations are well-defined under the event $\mathcal{S}$, and for $\mathbb{G}_n^{(k)}$ defined in (43), it holds that*

$$\hat{\mathbb{G}}_n^{(k)}(h) = \hat{\mathbb{G}}_n^{(k-1)}(h) + \sqrt{n}\hat{V}_n^{(k-1)}(h) \tag{64}$$

*and*

$$\hat{\mathbb{G}}_n^{(k)}(h) = \hat{\mathbb{G}}_n^{(k-1)}(\mathcal{C}_k h) + \sqrt{n}\hat{V}_n^{(k-1)}(\mathcal{C}_k h) + \begin{cases} \sqrt{n}[\hat{P}_{X,\epsilon} - P_X](\mu_X h) & \text{if } k \text{ odd} \\ \sqrt{n}[\hat{P}_{Y,\epsilon} - P_Y](\mu_Y h) & \text{if } k \text{ even} \end{cases}. \tag{65}$$

*Furthermore,*

$$\left|\hat{\mathbb{G}}_n^{(k)}(h)\right| \leq \left|\hat{\mathbb{G}}_n^{(k-1)}(\mathcal{C}_k h)\right| + \sqrt{n}\left|\hat{V}_n^{(k-1)}(\mathcal{C}_k h)\right| + c\|h\|_{\mathbf{L}^2(P)}\sqrt{n}\epsilon$$

$$= \left|\hat{\mathbb{G}}_n^{(k-1)}(\mathcal{C}_k h)\right| + \sqrt{n}\left|\hat{V}_n^{(k-1)}(\mathcal{C}_k h)\right| + O\left(\sqrt{n}\epsilon\right). \tag{66}$$

*Proof.* The proof that $\hat{P}_{n,X}^{(k-1)}(x) > 0$ and $\hat{P}_{n,Y}^{(k-1)}(y) > 0$ for all $x \in \mathcal{X}$ and $y \in \mathcal{Y}$ follows the exact same steps as in the proof of Prop. 15. We take this for granted and establish the recursion.

Consider the following steps in the case that $k$ is odd:

$$\hat{P}_n^{(k)}(h) = \sum_{x,y} h(x,y)\hat{P}_n^{(k)}(x,y) = \sum_{x,y} h(x,y)\frac{\hat{P}_{X,\epsilon}}{\hat{P}_{n,X}^{(k-1)}}(x)\hat{P}_n^{(k-1)}(x,y)$$

$$= \sum_{x,y} 1 \cdot h(x,y)\hat{P}_n^{(k-1)}(x,y) + \sum_{x,y}\left[\frac{\hat{P}_{X,\epsilon}}{\hat{P}_{n,X}^{(k-1)}}(x) - 1\right] \cdot h(x,y)\hat{P}_n^{(k-1)}(x,y)$$

$$= \hat{P}_n^{(k-1)}(h) + \hat{V}_n^{(k-1)}(h).$$

Subtracting $P(h)$ on both sides, we have that

$$[\hat{P}_n^{(k)} - P](h) = [\hat{P}_n^{(k-1)} - P](h) + \hat{V}_n^{(k-1)}(h). \tag{67}$$

This proves the uncentered recursion formula given in (64). We then show the centered version.

$$[\hat{P}_n^{(k)} - P](h)$$
$$= [\hat{P}_n^{(k)} - P](\mathcal{C}_X h) + [\hat{P}_n^{(k)} - P](\mu_X h)$$
$$= [\hat{P}_n^{(k)} - P](\mathcal{C}_X h) + [\hat{P}_{X,\epsilon} - P_X](\mu_X h)$$
$$= [\hat{P}_n^{(k-1)} - P](\mathcal{C}_X h) + \hat{V}_n^{(k-1)}(\mathcal{C}_X h) + [\hat{P}_{X,\epsilon} - P_X](\mu_X h).$$

Next, we bound the additional error term. By the Cauchy-Schwarz inequality,

$$[\hat{P}_{X,\epsilon} - P_X](\mu_X h) \leq \left\|\frac{\hat{P}_{X,\epsilon}}{P_X} - \mathbf{1}\right\|_{\mathbf{L}^2(P_X)} \cdot \|\mu_X h\|_{\mathbf{L}^2(P_X)}$$

$$= \sqrt{\chi^2(\hat{P}_{X,\epsilon}\|P_X)} \cdot \|\mu_X h\|_{\mathbf{L}^2(P_X)}$$

$$\leq \sqrt{\chi^2(\hat{P}_{X,\epsilon}\|P_X)} \cdot \|h\|_{\mathbf{L}^2(P)},$$

as $\mu_X$ is an orthogonal projection in $\mathbf{L}^2(P)$. Using convexity of $f$-divergences, we have that

$$\chi^2(\hat{P}_{X,\epsilon}\|P_X) \leq \epsilon\chi^2(\hat{P}_X\|P_X) + (1-\epsilon)\chi^2(P_X\|P_X) = \epsilon\chi^2(\hat{P}_X\|P_X).$$

This achieves the desired result. $\qquad\square$

Using similar ideas, we then prove an analog of Lem. 19.

**Lemma 27.** *For $k$ odd, it holds that*

$$\sqrt{n}\hat{V}_n^{(k-1)}(\mathcal{C}_X h) = \sum_{j=1}^{m} \left( \frac{\hat{P}_{X,\epsilon}}{\hat{P}_{n,X}^{(k-1)}}(x_j) - 1 \right) \hat{\mathbb{G}}_n^{(k-1)}(\mathcal{C}_X h \mathbf{1}_{jk}),$$

*whereas for $k$ even, it holds that*

$$\sqrt{n}\hat{V}_n^{(k-1)}(\mathcal{C}_Y h) = \sum_{j=1}^{m} \left( \frac{\hat{P}_{Y,\epsilon}}{\hat{P}_{n,Y}^{(k-1)}}(x_j) - 1 \right) \hat{\mathbb{G}}_n^{(k-1)}(\mathcal{C}_Y h \mathbf{1}_{jk}).$$

*Proof.* We give the proof for $k$ odd. We claim that we need only show that $P(\mathcal{C}_X h \mathbf{1}_{jk}) = 0$. This would show that

$$\begin{aligned}
\sqrt{n}\hat{V}_n^{(k-1)}(\mathcal{C}_X h) &= \sum_{x,y} \left( \frac{\hat{P}_{X,\epsilon}}{\hat{P}_{n,X}^{(k-1)}}(x) - 1 \right) [\mathcal{C}_X h](x,y)\hat{P}_n^{(k-1)}(x,y) \\
&= \sqrt{n}\sum_{j=1}^{m}\sum_{x,y} \left( \frac{\hat{P}_{X,\epsilon}}{\hat{P}_{n,X}^{(k-1)}}(x) - 1 \right) [\mathcal{C}_X h \mathbf{1}_{jk}](x,y)\hat{P}_n^{(k-1)}(x,y) \\
&= \sqrt{n}\sum_{j=1}^{m}\sum_{x,y} \left( \frac{\hat{P}_{X,\epsilon}}{\hat{P}_{n,X}^{(k-1)}}(x_j) - 1 \right) [\mathcal{C}_X h \mathbf{1}_{jk}](x,y)\hat{P}_n^{(k-1)}(x,y) \\
&= \sum_{j=1}^{m} \left( \frac{\hat{P}_{X,\epsilon}}{\hat{P}_{n,X}^{(k-1)}}(x_j) - 1 \right) \sqrt{n}\sum_{x,y}[\mathcal{C}_X h \mathbf{1}_{jk}](x,y)\hat{P}_n^{(k-1)}(x,y) \\
&= \sum_{j=1}^{m} \left( \frac{\hat{P}_{X,\epsilon}}{\hat{P}_{n,X}^{(k-1)}}(x_j) - 1 \right) \hat{\mathbb{G}}_n^{(k-1)}(\mathcal{C}_X h \mathbf{1}_{jk}),
\end{aligned}$$

where $P(\mathcal{C}_X h \mathbf{1}_{jk}) = 0$ is employed in the last step. Now the result follows from (65) in Prop. 26 and the definition of $\hat{\mathbb{G}}_n^{(k)}(h)$. To prove the claim, as in Lem. 19, write

$$\mathbb{E}\left[\mathcal{C}_X h \mathbf{1}_{jk} | X\right](x) = \begin{cases} \mathbb{E}\left[\mathcal{C}_X h | X\right](x_j) & \text{if } x = x_j \\ 0 & \text{if } x \neq x_j \end{cases}.$$

But $\mathbb{E}\left[\mathcal{C}_X h | X\right](x_j) = 0$ by definition of $\mathcal{C}_X$. Taking an expectation over $P_X$ gives that $P(\mathcal{C}_X h \mathbf{1}_{jk}) = 0$, which implies the desired result. The proof for $k$ even follows symmetrically. $\square$

For the remainder of the argument, we see that (66) can be unrolled so that

$$\left| \hat{\mathbb{G}}_n^{(k)}(h) \right| \leq \underbrace{\left| \mathbb{G}_n^{(0)}(\mathcal{C}_1 \ldots \mathcal{C}_k h) \right|}_{\text{first-order term}} + \underbrace{\sqrt{n}\sum_{\ell=1}^{k} \left| \hat{V}_n^{(\ell-1)}(\mathcal{C}_\ell \ldots \mathcal{C}_k h) \right|}_{\text{higher-order term}} + \underbrace{O(k\sqrt{n}\epsilon)}_{\text{misspecification}}, \tag{68}$$

where we use that $\mathbb{G}_n^{(0)} = \hat{\mathbb{G}}_n^{(0)}$.

Next, we need to bound $\left| \hat{V}_n^{(\ell-1)}(\mathcal{C}_\ell \ldots \mathcal{C}_k h) \right|$, in particular accounting for the marginal violation term. We follow similar steps as in the analysis of the higher-order term in Appx. D.3.

**Proposition 28.** *Assume that $P_{n,X}(x) > 0$ for all $x \in \mathcal{X}$. It holds that*

$$\max_{x \in \mathcal{X}} \left| \frac{\hat{P}_{X,\epsilon}(x)}{\hat{P}_{n,X}^{(k-1)}(x)} - 1 \right| \leq \begin{cases} \max\{n-1, 1\} & \text{if } k = 1 \\ \max\{1/\hat{p}_{\star,\epsilon}^2 - 1, 1\} & \text{if } k > 1. \end{cases} \tag{69}$$

*In addition, we have that*

$$\max_{x \in \mathcal{X}} \left| \frac{\hat{P}_{X,\epsilon}(x)}{\hat{P}_{n,X}^{(k-1)}(x)} - 1 \right| \leq \begin{cases} O\left(n\sqrt{\log\frac{1}{1-\epsilon}}\right) + n\sqrt{\frac{1}{2}\,\mathrm{KL}(P_{n,X}\|P_X)} & \text{if } k = 1 \\ O\left(\frac{1}{\hat{p}_{\star,\epsilon}^2}\sqrt{\log\frac{1}{1-\epsilon}}\right) + \frac{1}{\hat{p}_{\star,\epsilon}^2}\sqrt{\frac{1}{2}\,\mathrm{KL}(P_{n,X}\|P_X)} & \text{if } k > 1 \end{cases},$$

*Moreover, when* $\mathrm{KL}(P_{n,X}\|P_X) \leq \frac{\hat{p}_{\star,\epsilon}^2}{8}$ *and* $\epsilon \leq 1 - \exp\left(-\frac{\hat{p}_{\star,\epsilon}^2}{8}\right)$, *we have*

$$\max_{x \in \mathcal{X}} \left| \frac{\hat{P}_{X,\epsilon}(x)}{\hat{P}_{n,X}^{(k-1)}(x)} - 1 \right| \leq O\left(\frac{1}{\hat{p}_{\star,\epsilon}}\sqrt{\log \frac{1}{1-\epsilon}}\right) + \frac{2}{\hat{p}_{\star,\epsilon}}\sqrt{\frac{1}{2}\mathrm{KL}(P_{n,X}\|P_X)}. \tag{70}$$

*Proof.* First, observe that $\hat{P}_{n,X}^{(0)}(x) = P_{n,X}^{(0)}(x) \geq 1/n$ under the event $\mathcal{S}$. For $k > 1$ such that $k$ is odd, we have that for $x \in \mathcal{X}$,

$$\hat{P}_{n,X}^{(k-1)}(x) = \sum_{y \in \mathcal{Y}} \hat{P}_n^{(k-1)}(x,y) = \sum_{y \in \mathcal{Y}} \frac{\hat{P}_{Y,\epsilon}(y)}{\hat{P}_{n,Y}^{(k-2)}(y)} \hat{P}_n^{(k-2)}(x,y)$$

$$\geq \hat{p}_{\star,\epsilon} \sum_{y \in \mathcal{Y}} \hat{P}_n^{(k-2)}(x,y) = \hat{p}_{\star,\epsilon} \hat{P}_{n,X}^{(k-2)}(x) = \hat{p}_{\star,\epsilon}\hat{P}_{X,\epsilon}(x) \geq \hat{p}_{\star,\epsilon}^2.$$

The result for $k$ even can be proven similarly. We now prove the inequalities listed in the statement using on the lower bounds above.

**Proving the first inequality.** For any $x \in \mathcal{X}$,

$$\left| \frac{\hat{P}_{X,\epsilon}(x)}{\hat{P}_{n,X}^{(k-1)}(x)} - 1 \right| = \max\left\{ \frac{\hat{P}_{X,\epsilon}(x)}{\hat{P}_{n,X}^{(k-1)}(x)} - 1, 1 - \frac{\hat{P}_{X,\epsilon}(x)}{\hat{P}_{n,X}^{(k-1)}(x)} \right\} \leq \begin{cases} \max\{n-1,1\} & \text{if } k=1 \\ \max\{1/\hat{p}_{\star,\epsilon}^2 - 1, 1\} & \text{if } k>1 \end{cases},$$

which is the desired result.

**Proving the second and third inequalities.** Consider an odd $k \geq 1$. By the definition of total variation distance, it holds that

$$\max_{x \in \mathcal{X}} \left| \hat{P}_{X,\epsilon}(x) - \hat{P}_{n,X}^{(k-1)}(x) \right| \leq \mathrm{TV}(\hat{P}_{n,X}^{(k-1)}, \hat{P}_{X,\epsilon}).$$

According to Pinsker's inequality, we have that $\mathrm{TV}(\hat{P}_{n,X}^{(k-1)}, \hat{P}_{X,\epsilon}) \leq \sqrt{\frac{1}{2}\mathrm{KL}(\hat{P}_{n,X}^{(k-1)}\|\hat{P}_{X,\epsilon})}$, and so we have that

$$\max_{x \in \mathcal{X}} \left| \hat{P}_{X,\epsilon}(x) - \hat{P}_{n,X}^{(k-1)}(x) \right| \leq \sqrt{\frac{1}{2}\mathrm{KL}(\hat{P}_{n,X}^{(k-1)}\|\hat{P}_{X,\epsilon})} \leq \sqrt{\frac{1}{2}\mathrm{KL}(P_{n,X}^{(0)}\|\hat{P}_{X,\epsilon})},$$

where the last inequality follows by the monotonicity of Sinkhorn iterations given in Prop. 13. Notice that the remaining term is $\mathrm{KL}(P_{n,X}^{(0)}\|\hat{P}_{X,\epsilon}) = \mathrm{KL}(P_{n,X}\|\hat{P}_{X,\epsilon})$, which may not decay to zero as $n \to \infty$. Because $\epsilon < 1$, write

$$\mathrm{KL}(P_{n,X}\|\hat{P}_{X,\epsilon}) = \sum_{x \in \mathcal{X}} P_{n,X}(x) \log \frac{P_{n,X}(x)}{(1-\epsilon)P_X(x) + \epsilon\hat{P}_X(x)}$$

$$\leq \sum_{x \in \mathcal{X}} P_{n,X}(x) \log \frac{P_{n,X}(x)}{(1-\epsilon)P_X(x)}$$

$$= \mathrm{KL}(P_{n,X}\|P_X) + \log \frac{1}{1-\epsilon}$$

$$\implies \sqrt{\frac{1}{2}\mathrm{KL}(P_{n,X}\|\hat{P}_{X,\epsilon})} \leq \sqrt{\frac{1}{2}\mathrm{KL}(P_{n,X}\|P_X)} + \sqrt{\frac{1}{2}\log \frac{1}{1-\epsilon}}.$$

We can then apply the lower bounds

$$\max_{x \in \mathcal{X}} \left| \frac{\hat{P}_{X,\epsilon}(x)}{\hat{P}_{n,X}^{(k-1)}(x)} - 1 \right| \leq \begin{cases} n\left(\sqrt{\frac{1}{2}\mathrm{KL}(P_{n,X}\|P_X)} + \sqrt{\frac{1}{2}\log \frac{1}{1-\epsilon}}\right) & \text{if } k=1 \\ \frac{1}{\hat{p}_{\star,\epsilon}^2}\left(\sqrt{\frac{1}{2}\mathrm{KL}(P_{n,X}\|P_X)} + \sqrt{\frac{1}{2}\log \frac{1}{1-\epsilon}}\right) & \text{if } k>1 \end{cases}.$$

Finally, combining the arguments above, we have that

$$\max_{x \in \mathcal{X}} \left| \hat{P}_{X,\epsilon}(x) - \hat{P}_{n,X}^{(k-1)}(x) \right| \leq \sqrt{\frac{1}{2}\mathrm{KL}(P_{n,X}\|P_X)} + \sqrt{\frac{1}{2}\log \frac{1}{1-\epsilon}}$$

$$\leq \frac{\hat{p}_{\star,\epsilon}}{4} + \frac{\hat{p}_{\star,\epsilon}}{4} = \frac{\hat{p}_{\star,\epsilon}}{2},$$

where the last step invoked the assumption that

$$\mathrm{KL}(P_{n,X}\|P_X) \leq \frac{\hat{p}_{\star,\epsilon}^2}{8} \quad \text{and} \quad \epsilon \leq 1 - \exp\left(-\frac{\hat{p}_{\star,\epsilon}^2}{8}\right).$$

This means that

$$\min_{x \in \mathcal{X}} \hat{P}_{n,X}^{(k-1)}(x) \geq \min_{x \in \mathcal{X}} \hat{P}_{X,\epsilon}(x) - \max_{x \in \mathcal{X}} \left|\hat{P}_{n,X}^{(k-1)}(x) - \hat{P}_{X,\epsilon}(x)\right| \geq \frac{\hat{p}_{\star,\epsilon}}{2}.$$

Hence,

$$\max_{x \in \mathcal{X}} \left|\frac{\hat{P}_{X,\epsilon}(x)}{\hat{P}_{n,X}^{(k-1)}(x)} - 1\right| \leq \frac{\max_{x \in \mathcal{X}} \left|\hat{P}_{n,X}^{(k-1)}(x) - \hat{P}_{X,\epsilon}(x)\right|}{\min_{x \in \mathcal{X}} \hat{P}_{n,X}^{(k-1)}(x)} \leq \frac{2}{\hat{p}_{\star,\epsilon}} \sqrt{\frac{1}{2} \mathrm{KL}(P_{n,X}\|\hat{P}_{X,\epsilon})}.$$

Now, for $k$ even, set $k = 2t$ for $t \geq 0$. We have that

$$\max_{y \in \mathcal{Y}} \left|\hat{P}_{n,Y}^{(2t-1)}(y) - \hat{P}_{Y,\epsilon}(y)\right| \leq \mathrm{TV}(\hat{P}_{n,Y}^{(2t-1)}, \hat{P}_{Y,\epsilon}) \leq \sqrt{\frac{1}{2} \mathrm{KL}(\hat{P}_{Y,\epsilon}\|\hat{P}_{n,Y}^{(2t-1)})}.$$

Invoke Prop. 13 once again to achieve

$$\sqrt{\frac{1}{2} \mathrm{KL}(\hat{P}_{Y,\epsilon}\|\hat{P}_{n,Y}^{(2t-1)})} \leq \sqrt{\frac{1}{2} \mathrm{KL}(P_{n,X}\|\hat{P}_{X,\epsilon})} \leq \sqrt{\frac{1}{2} \mathrm{KL}(P_{n,X}\|P_X)} + \sqrt{\frac{1}{2} \log \frac{1}{1-\epsilon}},$$

which completes the proof. $\qquad\square$

Proceeding with similar steps, define the quantities

$$\hat{B}_1 := \hat{M}_1 \quad \text{and} \quad \hat{B}_2 := \max_{2 \leq \ell \leq k} \hat{M}_\ell \quad \text{for} \quad \hat{M}_\ell := \begin{cases} \max_{x \in \mathcal{X}} \left|\frac{\hat{P}_{X,\epsilon}(x)}{\hat{P}_{n,X}^{(\ell-1)}(x)} - 1\right| & \ell \text{ odd} \\ \max_{y \in \mathcal{Y}} \left|\frac{\hat{P}_{Y,\epsilon}(y)}{\hat{P}_{n,Y}^{(\ell-1)}(y)} - 1\right| & \ell \text{ even} \end{cases}.$$

We must now establish an analog of Prop. 20.

**Proposition 29.** *For any $k \geq 1$, the following holds under the event $\mathcal{S}$:*

$$\sqrt{n} \sum_{\ell=1}^{k} \left|\hat{V}_n^{(\ell-1)}(\mathcal{C}_\ell \ldots \mathcal{C}_k h)\right| \leq \sum_{j=1}^{m} \left(\hat{B}_1 |\mathbb{G}_n^{(0)}(h_{1,k}\mathbf{1}_{j\ell})| + \hat{B}_2 \sum_{\ell=2}^{k} |\mathbb{G}_n^{(0)}(h_{\ell,k}\mathbf{1}_{j\ell})|\right)$$
$$+ m\hat{B}_2 \|h\|_\infty \sqrt{n} k(k-1)[\hat{B}_1 + \hat{B}_2(k+1)/3].$$

*Proof.* This proof largely follows the argument of Prop. 20, while accounting for the misspecified marginal error. Using again the notation $h_{\ell,k} := \mathcal{C}_\ell \ldots \mathcal{C}_k h$, it follows from Lem. 27 that, for odd $\ell$,

$$\sqrt{n}\hat{V}_n^{(\ell-1)}(h_{\ell,k}) = \sum_{j=1}^{m} \left[\frac{\hat{P}_{X,\epsilon}}{\hat{P}_n^{(\ell-1)}}(x_j) - 1\right] \hat{\mathbb{G}}_n^{(\ell-1)}(h_{\ell,k}\mathbf{1}_{j\ell}) \leq \hat{M}_\ell \sum_{j=1}^{m} \left|\hat{\mathbb{G}}_n^{(\ell-1)}(h_{\ell,k}\mathbf{1}_{j\ell})\right|.$$

The bound above holds for $\ell$ even as well. Then, using (64) from Prop. 26 along with the triangle inequality, we have that for $\ell \geq 2$,

$$\left|[\hat{P}_n^{(\ell-1)} - P](h_{\ell,k}\mathbf{1}_{j\ell})\right| \leq \left|\hat{P}_n^{(\ell-2)} - P](h_{\ell,k}\mathbf{1}_{j\ell})\right| + \left|\hat{V}_n^{(\ell-2)}(h_{\ell,k}\mathbf{1}_{j\ell})\right|$$

which implies that

$$\left|\hat{\mathbb{G}}_n^{(\ell-1)}(h_{\ell,k}\mathbf{1}_{j\ell})\right| \tag{71}$$
$$\leq \left|\hat{\mathbb{G}}_n^{(\ell-2)}(h_{\ell,k}\mathbf{1}_{j\ell})\right| + \sqrt{n}\left|\hat{V}_n^{(\ell-2)}(h_{\ell,k}\mathbf{1}_{j\ell})\right|$$
$$\leq \left|\hat{\mathbb{G}}_n^{(0)}(h_{\ell,k}\mathbf{1}_{j\ell})\right| + \sqrt{n}\left|\hat{V}_n^{(0)}(h_{\ell,k}\mathbf{1}_{j\ell})\right| + \ldots + \sqrt{n}\left|\hat{V}_n^{(\ell-2)}(h_{\ell,k}\mathbf{1}_{j\ell})\right|$$
$$\leq \left|\hat{\mathbb{G}}_n^{(0)}(h_{\ell,k}\mathbf{1}_{j\ell})\right| + \hat{M}_1\sqrt{n}\hat{P}_n^{(0)}(|h_{\ell,k}|\mathbf{1}_{j\ell}) + \ldots + \hat{M}_\ell\sqrt{n}\hat{P}_n^{(\ell-2)}(|h_{\ell,k}|\mathbf{1}_{j\ell})$$
$$\leq \left|\hat{\mathbb{G}}_n^{(0)}(h_{\ell,k}\mathbf{1}_{j\ell})\right| + 2\|h\|_\infty \sqrt{n}\left[\hat{B}_1 + \hat{B}_2(\ell-1)\right](k-\ell+1), \tag{72}$$

by Lem. 18 and $\hat{M}_1 \le \hat{B}_1$ and $\hat{M}_\ell \le \hat{B}_2$ for $\ell \ge 2$. The bound above holds trivially for $\ell = 1$. Summing these bounds over $\ell$ and $j$, we have that

$$\sqrt{n} \sum_{\ell=1}^{k} \left| \hat{V}_n^{(\ell-1)}(h_{\ell,k}) \right|$$

$$\le \hat{M}_1 \sum_{j=1}^{m} |\mathbb{G}_n^{(0)}(h_{1,k}\mathbf{1}_{j\ell})| + \sum_{\ell=2}^{k} \hat{M}_\ell \sum_{j=1}^{m} \left| \hat{\mathbb{G}}_n^{(\ell-1)}(h_{\ell,k}\mathbf{1}_{j\ell}) \right|$$

$$\le \hat{B}_1 \sum_{j=1}^{m} |\mathbb{G}_n^{(0)}(h_{1,k}\mathbf{1}_{j\ell})| + \hat{B}_2 \sum_{\ell=2}^{k} \sum_{j=1}^{m} \left| \hat{\mathbb{G}}_n^{(\ell-1)}(h_{\ell,k}\mathbf{1}_{j\ell}) \right|$$

$$\le \hat{B}_1 \sum_{j=1}^{m} |\mathbb{G}_n^{(0)}(h_{1,k}\mathbf{1}_{j\ell})|$$

$$+ \hat{B}_2 \sum_{\ell=2}^{k} \sum_{j=1}^{m} \left( \left| \hat{\mathbb{G}}_n^{(0)}(h_{\ell,k}\mathbf{1}_{j\ell}) \right| + 2\|h\|_\infty \sqrt{n} \left[ \hat{B}_1 + \hat{B}_2(\ell-1) \right] (k-\ell+1) \right) \quad \text{apply (72)}$$

$$= \sum_{j=1}^{m} \left( \hat{B}_1 |\mathbb{G}_n^{(0)}(h_{1,k}\mathbf{1}_{j\ell})| + \hat{B}_2 \sum_{\ell=2}^{k} |\mathbb{G}_n^{(0)}(h_{\ell,k}\mathbf{1}_{j\ell})| \right)$$

$$+ 2m\hat{B}_2 \|h\|_\infty \sqrt{n} \sum_{\ell=2}^{k} \left[ \hat{B}_1 + \hat{B}_2(\ell-1) \right] (k-\ell+1),$$

because $|\mathcal{X}| = m$. We sum up the last term:

$$\sum_{\ell=2}^{k} \left[ \hat{B}_1 + \hat{B}_2(\ell-1) \right] (k-\ell+1) = \hat{B}_1 \sum_{\ell=1}^{k-1}(k-\ell) + \hat{B}_2 \sum_{\ell=1}^{k-1} \ell(k-\ell)$$

$$= \frac{k(k-1)}{2} \left[ \hat{B}_1 + \hat{B}_2(k+1)/3 \right],$$

which completes the proof. $\qquad \square$

### D.5.2 Mean Squared Error Bound

Ultimately, we wish to construct an upper bound for

$$\mathbb{E}_P \left[ \left( \hat{P}_n^{(k)}(h) - P(h) \right)^2 \mathbf{1}_{\mathcal{S}} \right] + \mathbb{E}_P \left[ (P_n(h) - P(h))^2 \mathbf{1}_{\mathcal{S}^c} \right], \tag{73}$$

as the method returns $P_n(h)$ when $\mathcal{S}$ is not satisfied. The first term will be controlled by intermediate tools developed above. The second term that includes $\mathcal{S}^c$ is no different from the one analyzed in Prop. 21. We handle the second term first. Recall from Prop. 21 that for any $\delta \in (0, 1)$,

$$\mathbb{E}_P \left[ (P_n(h) - P(h))^2 \mathbf{1}_{\mathcal{S}^c} \right] \le$$

$$4 \|h\|_\infty^2 \min\{2m(1-p_\star)^n, \delta\} + \frac{2\log(2/\delta)}{n} \|h\|_\infty^2 \, 2m(1-p_\star)^n. \tag{74}$$

Repeat the argument from the proof of Thm. 24: because $2[\log_2(2/\delta) + m\log(n+1)] \ge \log(m/\delta)$ and $-\log(1-p_\star) \ge p_\star \ge p_\star^2$, we have that

$$n \ge 2[\log_2(2/\delta) + m\log(n+1)]/p_\star^2 \implies n \ge \log(\delta/m)/\log(1-p_\star). \tag{75}$$

This in turn implies that $m(1-p_\star)^n \le \delta$, and gives as a condition on the sample size $n$. Further in the analysis, we will set $\delta = (\hat{p}_{\star,\epsilon}/n)^4$, so right-hand side of (74) can then be upper bounded further, resulting in

$$\mathbb{E}_P \left[ (P_n(h) - P(h))^2 \mathbf{1}_{\mathcal{S}^c} \right] \le 4\|h\|_\infty^2 \, \delta \left( 2 + \frac{\log(2/\delta)}{n} \right) = \tilde{O}\left( \frac{\hat{p}_{\star,\epsilon}^4}{n^4} \right),$$

a higher-order term compared to other components of the bound.

Next, we must control the left-hand side of (73). We perform the decomposition based on (68):

$$\mathbb{E}_P\left[\left(\hat{P}_n^{(k)}(h) - P(h)\right)^2 \mathbb{1}_S\right]$$

$$\leq \mathbb{E}_P\left[T_1^2 \mathbb{1}_S\right] + 2\mathbb{E}_P\left[\left|T_1\hat{T}_2\right|\mathbb{1}_S\right] + \mathbb{E}_P\left[\hat{T}_2^2 \mathbb{1}_S\right] \tag{76}$$

$$+ O(k\sqrt{\epsilon}) \cdot \mathbb{E}_P\left[\left(|T_1| + |\hat{T}_2|\right)\mathbb{1}_S\right] + O(k^2\epsilon) \tag{77}$$

for

$$T_1 := [P_n - P](\mathcal{C}_1 \ldots \mathcal{C}_k h) \text{ and } \hat{T}_2 := \sum_{\ell=1}^{k}\left|\hat{V}_n^{(\ell-1)}(\mathcal{C}_\ell \ldots \mathcal{C}_k h)\right|. \tag{78}$$

Recall the events $\mathcal{E}_1^\delta$ and $\mathcal{E}_2^\delta$ and $\mathcal{E}_3^\delta$ from Appx. D.4. To perform this computation efficiently, we will split the bounds on each term into two components. In particular, we will show that

- Under the event $S \cap \mathcal{E}_1^\delta \cap \mathcal{E}_2^\delta : \left|\hat{T}_2\right| \leq \mathcal{T}_2 + E_2$,

- Under the event $S \backslash (\mathcal{E}_1^\delta \cap \mathcal{E}_2^\delta) : \left|\hat{T}_2\right| \leq \mathcal{T}_2^c + E_2^c$,

- Under the event $S \cap \mathcal{E}_3^\delta : |T_1| \leq \mathcal{T}_1$,

- Under the event $S \backslash \mathcal{E}_3^\delta : |T_1| \leq \mathcal{T}_1^c$,

where any term denoted with "$E$" will represent all error terms that include $\epsilon$ and will be written in big-$O$ notation. There are no errors for the bounds on $T_1$, as this term does not depend on the misspecified marginals. The idea is that for the "$\mathcal{T}_2$" terms we may reuse the bounds derived in Appx. D.4 by simply replacing $p_\star$ with $\hat{p}_{\star,\epsilon}$. This is due to the fact that the dependence of the analogous terms from Appx. D.4 depend on $p_\star$ only through Prop. 14; similarly, the corresponding terms in this section depend on $\hat{p}_{\star,\epsilon}$ through Prop. 28. We return to the terms in (76) and (77).

Decomposing on $\mathcal{E}_3^\delta$ will result in a bound of the form

$$O(k\sqrt{\epsilon}) \cdot \mathbb{E}_P\left[|T_1|\mathbb{1}_S\right] \leq O(k\sqrt{\epsilon}) \cdot (\delta\mathcal{T}_1^c + \mathcal{T}_1).$$

Decomposing on $\mathcal{E}_1^\delta \cap \mathcal{E}_2^\delta$ will result in a bounds of the form

$$\mathbb{E}_P\left[\hat{T}_2^2\mathbb{1}_S\right] \leq 2\delta(\mathcal{T}_2^c)^2 + \mathcal{T}_2^2 + \tilde{O}\left(\delta\left((E_2^c)^2 + E_2^c\mathcal{T}_2^c\right) + \left(E_2^2 + E_2\mathcal{T}_2\right)\right)$$

$$O(k\sqrt{\epsilon}) \cdot \mathbb{E}_P\left[|\hat{T}_2|\mathbb{1}_S\right] \leq O(k\sqrt{\epsilon}) \cdot (\delta\left(\mathcal{T}_2^c + E_2^c\right) + \mathcal{T}_2 + E_2).$$

Finally, decomposing on $\mathcal{E}_1^\delta \cap \mathcal{E}_2^\delta \cap \mathcal{E}_3^\delta$ will result in a bound of the form

$$\mathbb{E}_P\left[\left|T_1\hat{T}_2\right|\mathbb{1}_S\right] \leq 3\delta\mathcal{T}_1^c\mathcal{T}_2^c + \mathcal{T}_1\mathcal{T}_2 + \tilde{O}\left(\delta\mathcal{T}_1^c E_2^c + \mathcal{T}_1 E_2\right).$$

The leading terms $2\delta(\mathcal{T}_2^c)^2 + \mathcal{T}_2^2$ and $3\delta\mathcal{T}_1^c\mathcal{T}_2^c + \mathcal{T}_1\mathcal{T}_2$ from both bounds should have the exact same form as the terms in Lem. 22 and Lem. 23, with $p_\star$ replaced by $\hat{p}_{\star,\epsilon}$, thus retaining the same dependence on $(n, k)$. By setting $\delta = \hat{p}_{\star,\epsilon}^4/n^4$, we will achieve a similar result to Thm. 24, i.e., that

$$\mathbb{E}_P\left[\left(\hat{P}_n^{(k)}(h) - P(h)\right)^2\mathbb{1}_S\right]$$

$$\leq \frac{\sigma_k^2}{n} + \tilde{O}\left(\frac{k^6}{n^{3/2}}\right)$$

$$+ \tilde{O}\left((\hat{p}_{\star,\epsilon}/n)^4\left(E_2^c(E_2^c + \mathcal{T}_2^c)\right) + E_2\left(E_2 + \mathcal{T}_2\right) + (\hat{p}_{\star,\epsilon}/n)^4\mathcal{T}_1^c E_2^c + \mathcal{T}_1 E_2\right). \tag{79}$$

$$+ \tilde{O}\left(k\sqrt{\epsilon}\left((\hat{p}_{\star,\epsilon}/n)^4\mathcal{T}_1^c + \mathcal{T}_1 + (\hat{p}_{\star,\epsilon}/n)^4\left(\mathcal{T}_2^c + E_2^c\right) + \mathcal{T}_2 + E_2\right) + k^2\epsilon\right). \tag{80}$$

It remains to quantify the $\tilde{O}$ terms by computing the order of the 6 constants $(\mathcal{T}_2, E_2, \mathcal{T}_2^c, E_2^c, \mathcal{T}_1, \mathcal{T}_1^c)$. We follow similar steps to Lem. 22 and Lem. 23 to achieve this.

**Lemma 30.** *For $\delta = (\hat{p}_{\star,\epsilon}/n)^4$, assume that $n \geq 8[\log_2(2/\delta) + m\log(n+1)]/\hat{p}_{\star,\epsilon}^2$ and $\epsilon \leq 1 - \exp\left(-\frac{\hat{p}_{\star,\epsilon}^2}{8}\right)$. Then, it holds that*

$$\mathcal{T}_2^c = \tilde{O}\left(\frac{k^2}{\hat{p}_{\star,\epsilon}^2}\left(n + \frac{k}{\hat{p}_{\star,\epsilon}^2}\right)\right), \quad E_2^c = 0$$

$$\mathcal{T}_2 = \tilde{O}\left(\frac{k^3}{n\hat{p}_{\star,\epsilon}^2}\right), \quad E_2 = \tilde{O}\left(\frac{k^3}{\hat{p}_{\star,\epsilon}^2}\left(\sqrt{\frac{1}{n}\log\frac{1}{1-\epsilon}} + \log\frac{1}{1-\epsilon}\right)\right).$$

*Proof.* The following computations are done under the event $\mathcal{S}$. First, apply Prop. 29 to write

$$\sqrt{n}\left|\hat{T}_2\right| \leq \sum_{j=1}^{m}\left(\hat{B}_1\,|\mathbb{G}_n^{(0)}(h_{1,k}\,\mathbf{1}_{j\ell})| + \hat{B}_2\sum_{\ell=2}^{k}|\mathbb{G}_n^{(0)}(h_{\ell,k}\,\mathbf{1}_{j\ell})|\right)$$

$$+ m\hat{B}_2\,\|h\|_\infty\,k(k-1)[\hat{B}_1 + \hat{B}_2(k+1)/3]. \tag{81}$$

We decompose on the event $\mathcal{E}_1^\delta \cap \mathcal{E}_2^\delta$.

**Bound $|T_2|$ under the event $\mathcal{S}\backslash(\mathcal{E}_1^\delta \cap \mathcal{E}_2^\delta)$.** In this case, we apply (69) from Prop. 28 to get $\hat{B}_1 \leq n$ and $\hat{B}_2 \leq 1/\hat{p}_{\star,\epsilon}^2$, along with the universal bounds from Lem. 18:

$$\frac{1}{\sqrt{n}}\,|\mathbb{G}_n^{(0)}(h_{1,k}\,\mathbf{1}_{j\ell})| \leq 2\,\|h_{1,k}\|_\infty \leq 4k\,\|h\|_\infty$$

$$\frac{1}{\sqrt{n}}\sum_{\ell=2}^{k}|\mathbb{G}_n^{(0)}(h_{\ell,k}\,\mathbf{1}_{j\ell})| \leq 2\sum_{\ell=2}^{k}\|h_{\ell,k}\|_\infty \leq \sum_{\ell=2}^{k}4(k-\ell+1)\,\|h\|_\infty = 2k(k-1)\,\|h\|_\infty$$

so that by plugging into (81),

$$\left|\hat{T}_2\right| \leq \underbrace{\|h\|_\infty\,mk\left[4n + \frac{k-1}{\hat{p}_{\star,\epsilon}^2}\left(n + 2 + \frac{k+1}{3\hat{p}_{\star,\epsilon}^2}\right)\right]}_{\mathcal{T}_2^c} + \underbrace{0}_{E_2^c}.$$

**Bound $|T_2|$ under the event $\mathcal{S} \cap \mathcal{E}_1^\delta \cap \mathcal{E}_2^\delta$.** In this case, we may use that $n \geq 8/\hat{p}_{\star,\epsilon}^2$ (because $[\log_2(2/\delta) + m\log(n+1)] \geq 1$ for $\delta \in (0,1)$) and apply (70) from Prop. 28 to get

$$\max\left\{\hat{B}_1, \hat{B}_2\right\} \leq O\left(\frac{1}{\hat{p}_{\star,\epsilon}}\sqrt{\log\frac{1}{1-\epsilon}}\right) + \frac{2}{\hat{p}_{\star,\epsilon}}\sqrt{\frac{2\log_2(2/\delta) + 2m\log(n+1)}{2n}}$$

The bounds based on $\mathcal{E}_2^\delta$ give

$$|\mathbb{G}_n^{(0)}(h_{1,k}\,\mathbf{1}_{j\ell})| \leq \sqrt{2\log\frac{2mk}{\delta}}\,2k\,\|h\|_\infty$$

$$\sum_{\ell=2}^{k}|\mathbb{G}_n^{(0)}(h_{\ell,k}\,\mathbf{1}_{j\ell})| \leq \sum_{\ell=2}^{k}\sqrt{2\log\frac{2mk}{\delta}}\,2(k-\ell+1)\,\|h\|_\infty \leq \sqrt{2\log\frac{2mk}{\delta}}\,k(k-1)\,\|h\|_\infty.$$

By plugging into (81), we can reuse the steps in the bound from (55) (for all terms without $\epsilon$) to write

$$\left|\hat{T}_2\right| \leq \frac{4mk\,\|h\|_\infty\,[\log_2(2/\delta) + 2m\log(n+1)]^{1-\mathbb{1}\{k=1\}/2}}{n\hat{p}_{\star,\epsilon}^2} \times$$

$$\left[\hat{p}_{\star,\epsilon}\sqrt{2\log(2mk/\delta)}(k+1) + (k-1)(k+4)\right] + E_2,$$

so that

$$\mathcal{T}_2 = \frac{4mk\,\|h\|_\infty\,[\log_2(2/\delta) + 2m\log(n+1)]^{1-\mathbb{1}\{k=1\}/2}}{n\hat{p}_{\star,\epsilon}^2}$$

$$\times\left[\hat{p}_{\star,\epsilon}\sqrt{2\log(2mk/\delta)}(k+1) + (k-1)(k+4)\right].$$

We compute $E_2$ by using that

$$\max\left\{\hat{B}_1, \hat{B}_2\right\} \leq O\left(\tfrac{1}{\hat{p}_{\star,\epsilon}}\sqrt{\log\tfrac{1}{1-\epsilon}}\right) + \tilde{O}\left(\tfrac{1}{\hat{p}_{\star,\epsilon}\sqrt{n}}\right)$$

$$|\mathbb{G}_n^{(0)}(h_{1,k}\,\mathbf{1}_{j\ell})| \leq \tilde{O}\left(k\right)$$

$$\sum_{\ell=2}^{k} |\mathbb{G}_n^{(0)}(h_{\ell,k}\,\mathbf{1}_{j\ell})| \leq \tilde{O}\left(k^2\right),$$

which gives

$$E_2 = \tilde{O}\left(\tfrac{k^3}{\hat{p}_{\star,\epsilon}^2}\left(\sqrt{\tfrac{1}{n}\log\tfrac{1}{1-\epsilon}} + \log\tfrac{1}{1-\epsilon}\right)\right).$$

$\square$

We now make the corresponding argument for the term $T_1$.

**Lemma 31.** *For* $\delta = (\hat{p}_{\star,\epsilon}/n)^4$, *it holds that*

$$\mathcal{T}_1^c = \tilde{O}(k), \quad \mathcal{T}_1 = \tilde{O}\left(\frac{k}{\sqrt{n}}\right).$$

*Proof.* The following computations are done under the event $\mathcal{S}$.

**Bound** $|T_1|$ **under the event** $\mathcal{S}\backslash\mathcal{E}_3^\delta$. Here we simply apply a universal bound on the empirical process term:

$$\frac{1}{\sqrt{n}}|\mathbb{G}_n^{(0)}(h_{1,k})| \leq 2\,\|h_{1,k}\|_\infty \leq 4k\,\|h\|_\infty,$$

so that $\mathcal{T}_1^c = 4k\,\|h\|_\infty$

**Bound** $|T_1|$ **under the event** $\mathcal{S}\cap\mathcal{E}_3^\delta$. Now, we may use the definition of the event $\mathcal{E}_3^\delta$ to achieve

$$\frac{1}{\sqrt{n}}|\mathbb{G}_n^{(0)}(h_{1,k})| \leq \sqrt{\frac{2\log(2/\delta)}{n}}\,2k\,\|h\|_\infty = \mathcal{T}_1.$$

$\square$

Knowing that $E_2^c = 0$, we simplify (79) and (79) to read

$$\tilde{O}\left(E_2\left(E_2 + \mathcal{T}_2 + \mathcal{T}_1\right)\right)$$

$$\tilde{O}\left(k\sqrt{\epsilon}\left((\hat{p}_{\star,\epsilon}/n)^4\mathcal{T}_1^c + \mathcal{T}_1 + (\hat{p}_{\star,\epsilon}/n)^4\mathcal{T}_2^c + \mathcal{T}_2 + E_2\right) + k^2\epsilon\right).$$

We now combine the bounds from the previous two lemmas to compute (79) and (80) to state the main result.

**Theorem 32.** *Let Asm. 1 be true with error* $\epsilon \in [0,1)$. *For a sequence of rebalanced distributions* $(\hat{P}_n^{(k)})_{k\geq 1}$, *there exists an absolute constant* $C > 0$ *such that when* $n \geq C[\log_2(2n/\hat{p}_{\star,\epsilon}) + m\log(n+1)]/\min\{p_\star, \hat{p}_{\star,\epsilon}\}^2$, *we have that*

$$\mathbb{E}_P\left[\left(\hat{P}_n^{(k)}(h) - P(h)\right)^2 \mathbb{1}_{\mathcal{S}}\right] + \mathbb{E}_P\left[\left(P_n(h) - P(h)\right)^2 \mathbb{1}_{\mathcal{S}^c}\right] \leq \frac{\sigma_k^2}{n} + \tilde{O}\left(\frac{k^6}{n^{3/2}}\right)$$

$$+ \tilde{O}\left(\frac{k^4}{\hat{p}_{\star,\epsilon}^2}\left(\sqrt{\tfrac{1}{n}\log\tfrac{1}{1-\epsilon}} + \log\tfrac{1}{1-\epsilon}\right)\left[\frac{k^2}{\hat{p}_{\star,\epsilon}^2}\left(\sqrt{\tfrac{1}{n}\log\tfrac{1}{1-\epsilon}} + \log\tfrac{1}{1-\epsilon} + \tfrac{1}{n}\right) + \tfrac{1}{\sqrt{n}}\right]\right)$$

$$+ \tilde{O}\left(k^2\left[\sqrt{\epsilon}\left(\tfrac{\hat{p}_{\star,\epsilon}^4}{n^4} + \tfrac{1}{\sqrt{n}} + \tfrac{\hat{p}_{\star,\epsilon}^2 k}{n^4}\left(n + \tfrac{k^2}{\hat{p}_{\star,\epsilon}^2}\right) + \tfrac{k^2}{\hat{p}_{\star,\epsilon}^2}\left[\tfrac{1}{n} + \sqrt{\tfrac{1}{n}\log\tfrac{1}{1-\epsilon}} + \log\tfrac{1}{1-\epsilon}\right]\right) + \epsilon\right]\right).$$

# E    Experimental Details

We provide the full experimental details of the experimental results from Sec. 4. We report additional evaluations on downstream tasks with linear probing and zero-shot retrieval. Finally, we give illustrations of the sensitivity to misspecified marginals, and of the convergence to the given marginals.

## E.1    Datasets

**Pre-Training Data.** The pre-training data was taken from the public ImageNet-Captions dataset [Fang et al., 2013]. We subset the dataset by selecting the 250 classes that were most frequent in the dataset, resulting in 174,594 images and associated Flickr captions. The exact images used and their associated captions are given in the code supplement.

**Evaluation Data.** We perform zero-shot classification (as described in Sec. 4), zero-shot retrieval, and linear probing with various image classification and image-caption datasets. We used the default class captions (for classification) and default linear probing parameters from the CLIP Benchmark repo. The datasets (test splits) used were:

- **CIFAR-10:** 10,000 colored natural images labeled with one of 10 classes.
- **CIFAR-100:** 10,000 colored natural images labeled with one of 100 classes.
- **STL-10:** 80,000 colored natural images labeled with one of 10 classes.
- **MS-COCO:** 41,000 colored natural images with associated captions.
- **Flickr8k:** 8,000 colored natural images with associated captions.
- **Rendered SST2:** 1,821 images of typed natural language with sentiment label (2 classes).
- **VOC2007:** 4,952 colored natural images labeled with one of 20 classes.
- **FGVC Aircraft:** 34,000 colored natural images labeled with one of 102 classes.

Evaluation scripts using the various embedding models (described below) are provided.

## E.2    Model Specification and Hyperparameters

**Architecture and Implementation.** The models considered CLIP models [Radford et al., 2021], and are specified by pairs of encoders $(f_\theta, g_\theta)$, representing images and text, respectively. The encoders decompose into $f_\theta = f_\theta^{\text{head}} \circ f_\theta^{\text{base}}$ (similarly for $g_\theta$) where $f_\theta^{\text{base}}$ denotes a base image encoder and $f_\theta^{\text{head}}$ denotes a trainable head model. The head models are feed-forward networks with two hidden layers, 256 hidden units, and 128-dimensional output representations. Their input dimensions may be 512 or 768, depending on whether a CLIP model or BERT/GPT-2 model is used as the base. For the image base/foundation models, we use the open-source OpenCLIP implementation of the ViT-B/32 model with the `laion2b_s34b_b79k` model tag. For the text encoder, we use the encoder of the variant of the ViT-B/32 with tag `datacomp_xl_s13b_b90k`. For the other text encoders the Huggingface implementations of GPT-2 and BERT were used.

**Optimizer.** For optimization, models were trained with stochastic gradient descent (SGD) with the learning rate tuned along the grid $\left\{1^{-3}, 3^{-3}, 1^{-2}, 3^{-2}, 1^{-1}\right\}$ and a fixed weight decay parameter of 0.01. Momentum-variants such as Adam [Kingma and Ba, 2015] were not used to isolate the effect of varying losses as described in Sec. 4.

## E.3    Compute Environment

Experiments were run on a CPU/GPU workstation with 12 virtual cores, 126G of memory, and four NVIDIA TITAN Xp GPUs with 12G memory each. The code was written in Python 3 and we use PyTorch for automatic differentiation. The OpenCLIP and CLIP Benchmark repositories were used for zero-shot evaluation.

### E.4 CLIP and Multi-CLIP

We considered in the contrastive learning example from Sec. 2 – see (6) in particular – a variant of the CLIP objective in which either zero, or one, or more than one balancing iterations are performed (see (6)), via optimizing

$$L_n^{(k)} = -\frac{1}{2} \sum_{i=1}^n \left[ \log Q_n^{(k)}(X_i, Y_i) + \log R_n^{(k)}(X_i, Y_i) \right]. \tag{82}$$

This contrasts the single-iteration variant $L_n^{(1)}$ which in fact reduces to the original CLIP loss. Because these iterations are applied in the objective, backpropagation occurs *through* each iteration.

In Fig. 3, we plot the zero-shot classification performance (in terms of average per-class recall) of the variants trained on $L_n^{(0)}$ (the normalized initial measure, *No Balancing*), $L_n^{(1)}$ (the original CLIP loss, *CLIP balancing*), and $L_n^{(2)}$ (the two-iteration CLIP loss, *Multi-CLIP balancing*). We also vary the quality of the text encoder $f_{\theta_T}$, observing an overall accuracy trend of GPT-2 $\prec$ BERT $\prec$ CLIP across variants, which is to be expected given the base representation quality of each model. Interestingly, there is an improvement stemming from performing multiple balancing iterations across choices of the text embedding, the batch size $m$, and the evaluation dataset.

### E.5 Metadata Curation

We considered in the metadata curation example from Sec. 2 how to use balancing to adjust the entire pre-training set, in the spirit of Xu et al. [2024]. The target marginal $P_Y$ is selected by choosing a threshold for which frequent keywords have their probability mass truncated, and the probability measure is normalized to sum to one. In Fig. 4, we show the observed marginal $P_{n,Y}$ and the target marginal $P_Y$ sorted in increasing order (left). The original marginal on $\mathcal{Y}$ has approximately 5 orders of magnitude of difference between the most and least probable keyword. After balancing, the target marginal has less than 2 orders of difference. To see how this affects downstream performance, we plot the zero-shot classification accuracy over training iterations in Fig. 4 (right) when using the original dataset (orange) and using the metadata-balanced dataset (blue). We observe moderate improvement especially in the small batch regime ($m = 512$) when curating the dataset.

### E.6 Additional Experiments

In this section, we provide 1) a synthetic data example that helps elucidate the role of the spectral decomposition introduced in Sec. 3, and 2) additional evaluations on downstream tasks such as zero-shot retrieval and linear probing. For the latter, we maintain the experimental settings as used in the zero-shot classification example from Sec. 4 (Fig. 3). That is, we train variants of CLIP models (see Sec. 2) on the ImageNet-Captions dataset [Fang et al., 2013]. As before, we use a fixed image/text encoder as a base vector representation and compose it with a trainable feed-forward neural network, i.e., $f_\theta = f_\theta^{\text{head}} \circ f^{\text{base}}$, for $\theta = \theta_I$ (images) or $\theta = \theta_T$ (text). For the base text embeddings, we maintain three levels of model quality: GPT-2 [Radford et al., 2019], BERT [Devlin et al., 2019], and CLIP-based encodings.

**Baseline Comparisons.** We present a synthetic data example to understand the role of the singular values $s_2, \ldots, s_m$ and compare our approach to simple baselines that make use of $(P_X, P_Y)$. We also consider misspecification of these target marginals, in that they are chosen by the user but are not the marginal distributions of the data-generating distribution $P$. First, while one can verify by hand that (14) is a distribution for which $s_2 = s$, we construct a more general example for $m \geq 2$. We leave the construction to the end of this example. For controllable misspecification, we define $\epsilon \in [0, 0.5]$ to be the *misspecification* level, so that the corrupted target marginals are set to be

$$\tilde{P}_X := (1 - \epsilon) P_X + \epsilon \hat{P}_X \text{ and } \tilde{P}_Y := (1 - \epsilon) P_Y + \epsilon \hat{P}_Y, \tag{83}$$

where $\hat{P}_X$ and $\hat{P}_Y$ are drawn independently and randomly from the $\text{Dirichlet}(\mathbf{1}_m)$ distribution (i.e. uniformly over the probability simplex on $m$ atoms). Finally, other than the empirical measure $P_n$, we define one additional baseline; the *importance weighted independently (IPWI)* estimator is defined as

$$P_n^{\text{IPWI}}(x, y) = \frac{\tilde{P}_X(x)}{P_{n,X}(x)} \frac{\tilde{P}_Y(y)}{P_{n,Y}(y)} P_n(x, y). \tag{84}$$

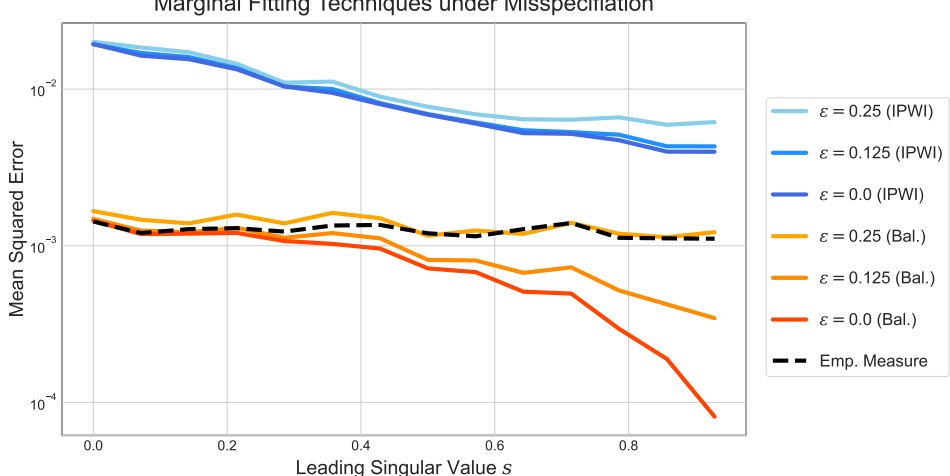

Figure 5: **Baseline Comparisons across Dependence and Misspecification Levels.** Each line refers to a combination of an estimation method (the empirical probability measure $P_n$, the estimator $P_n^{\mathrm{IPWI}}$ from (84), or the balancing estimator $P_n^{(k)}$ for $k = 8$) and a noise level on the provided marginals (see (83)). The $y$-axis shows the mean squared error of estimating a linear functional. The $x$-axis represents the dependence level $s = s_2$ (i.e. the leading singular value other than $s_1 = 1$).

This estimator simply reweighs all cells of the empirical measure by the likelihood ratio from each observed marginal to the target marginal. Note that the result may not even be a probability measure, as it may not sum to one. Observe the comparative performance in (see Fig. 5). We notice in particular that the naive $P_n^{\mathrm{IPWI}}$ is outperformed by empirical measure uniformly over $s$, as by applying both reweightings simultaneously, the estimator does not satisfy either marginal constraint. On the other hand, under the maximum amount of target marginal corruption ($\epsilon = 0.5$), the balancing-based estimator suffers an approximately half-order of magnitude in MSE. When $s \approx 1$, the MSE of the balancing estimator decreases significantly. We hypothesize that this is because the data sources $X$ and $Y$ are nearly a function of one another, and if this function is estimable to high precision by a small amount of data, then a single marginal can identify the entire joint distribution via pushforward calculations. That being said, it is important to note that the quantities $u_j$ and $v_j$ in (15) also depend on $s$, so it is difficult to control the singular values without controlling the respective bases.

As for the construction of the probability mass function and test function, let $\mathbf{I}_m$ and $\mathbf{1}_{m \times m}$ denote the identity matrix and matrix of ones in $\mathbb{R}^{m \times m}$. For any $s \in (0, 1)$ and $m \geq 1$, consider the probability mass matrix $P$ given by

$$P = \frac{1}{m} \left[ \frac{1}{m} \mathbf{1}_{m \times m} + s \left( \mathbf{I}_m - \frac{1}{m} \mathbf{1}_{m \times m} \right) \right].$$

The eigenvalues of the first matrix in the squared brackets are $(1, 0, \ldots, 0)$, as it is a rank 1 matrix for which $\mathbf{1}_m$ is an eigenvector. The second matrix in the square brackets is the centering matrix (the projection matrix that subtracts the mean of a vector's components from the entire vector). Multiplied by $s$, it has eigenvalues $(0, s, \ldots, s)$ where 0 is associated to the eigenvector $\mathbf{1}_m$. Thus, the matrix in its entirety has eigenvalues $(1/m, s/m, \ldots, s/m)$, where the scaling factor ensures that $P$ sums to one. The relation (13) holds for this choice of $P$ and uniform marginals, with $s_2 = \ldots = s_m = s$. Thus, by tuning $s$, we may control the level of dependence between $X$ and $Y$. Finally, because $\mathcal{X}$ and $\mathcal{Y}$ are finite, we can also specify the test function $h$ via an $m \times m$ table indexed by $i$ (meaning $x_i$) and $j$ (meaning $y_j$). We let $h(x_i, y_j) = |Z_{ij}|$ where the $Z_{ij}$ are independently drawn from a standard normal distribution. The resulting mean squared error is estimated with 200 seeds at $n = 300$.

**Zero-Shot Retrieval.** In this evaluation, we assess the ability of the learned representations to match queries from one modality to their counterparts in another modality. We are given a test sets $\mathcal{X}_{\text{test}} = \{x_1, \ldots, x_M\}$ of images and $\mathcal{Y}_{\text{test}} = \{y_1, \ldots, y_N\}$ of texts in natural language. We are also given a matrix of annotations $A \in \{0, 1\}^{M \times N}$ where $A_{ij} = 1$ if and only if $y_j$ is a "relevant" caption

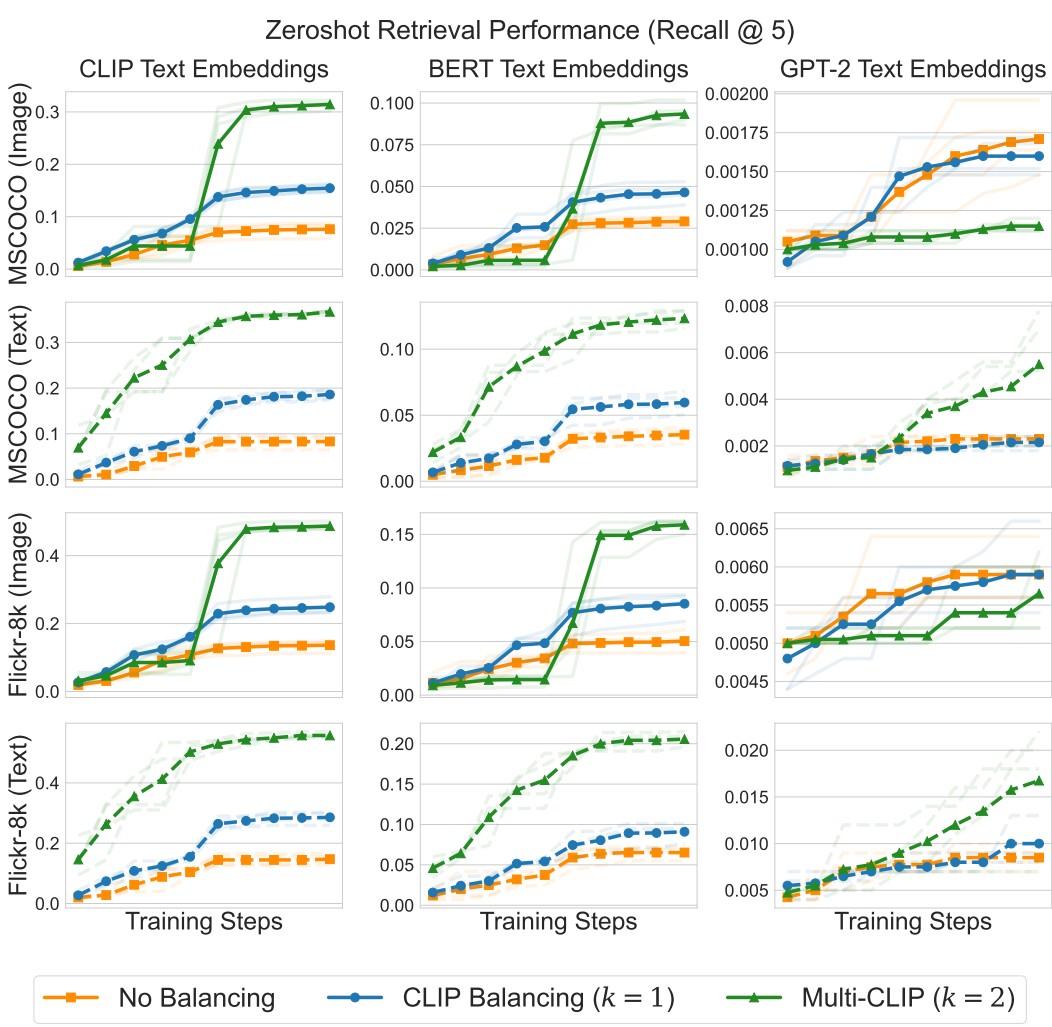

Figure 6: **Zero-Shot Retrieval Performance across Embeddings and Objectives.** The three vertical panels describe different choices of the text encoder $f_{\theta_T}$ which increases in quality from left to right; that is, pre-trained GPT-2, BERT, and CLIP embeddings, respectively. Rows indicate various datasets, either MS-COCO or Flickr8k. evaluated under recall at $K = 5$ for image and text retrieval, respectively. The $y$-axis of each plot indicates the metric (see (85)) for either image or text retrieval, whereas the $x$-axis indicates training iterations at batch size $512$.

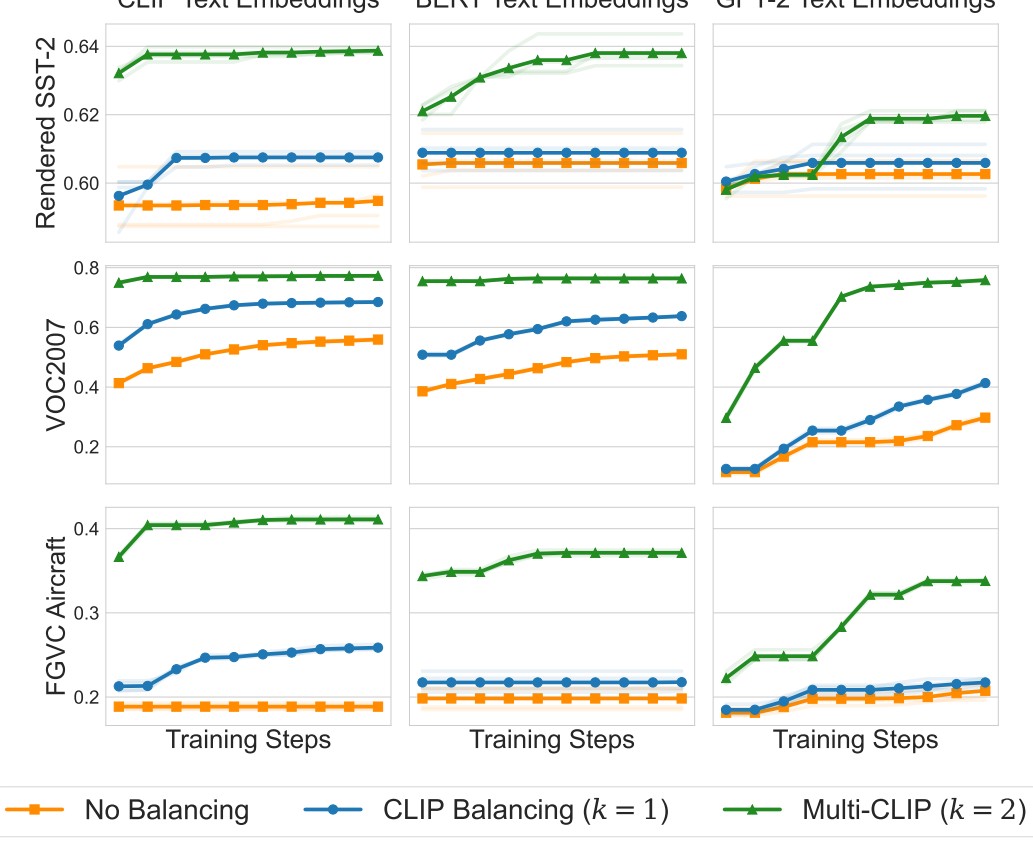

Figure 7: **Linear Probe Performance across Embeddings and Objectives.** The three vertical panels describe different choices of the text encoder $f_{\theta_T}$ which increases in quality from left to right; that is, pre-trained GPT-2, BERT, and CLIP embeddings, respectively. Rows indicate various evaluation datasets from Rendered SST2, VOC2007, and FGVC Aircraft. The $y$-axis of each plot indicates the average per-class recall, whereas the $x$-axis indicates training iterations at batch size 512.

for image $x_i$ (and vice versa). Given a particular query $y \in \mathcal{Y}_{\text{test}}$, we define the top-$K$ neighborhood of $y$ as

$$\mathcal{N}_K(y;\theta) = \arg\max_{S \subseteq [M]:|S|=K} \sum_{i \in S} \langle f_{\theta_I}(x_i), f_{\theta_T}(y) \rangle,$$

i.e. the images in the test set that have the closest embeddings under the given model. Then, we may define the *average recall at $K$ for image retrieval* metric as

$$\text{AverageRecall}_K(\theta) := \frac{1}{N} \sum_{j=1}^{N} \frac{\sum_{i \in \mathcal{N}_K(y_j;\theta)} A_{ij}}{\sum_{i' \in [M]} A_{i'j}}. \tag{85}$$

In words, the metric evaluates the retrieval system's ability to detect relevant items in the dataset, in this case by comparing the closeness of the image-text representations. We can analogously define the *average recall at $K$ for text retrieval* metric by swapping the role of $x$ and $y$ above. The results for both retrieval metrics on the MS-COCO [Lin et al., 2015] and Flickr8k [Hodosh et al., 2013] benchmarks are given in Fig. 6.

**Linear Probe.** Here, we evaluate the quality of the model's encoders by fine-tuning a single linear layer on top of the learned representations for a classification task. In the case of linear probing via image classification, we use only the image encoder $f_{\theta_I}$. We are given a training set $\{(x_1, c_1), \dots, (x_N, c_N)\}$ of image-label pairs, where each $c_i \in \{1, \dots, C\}$. We fix the model

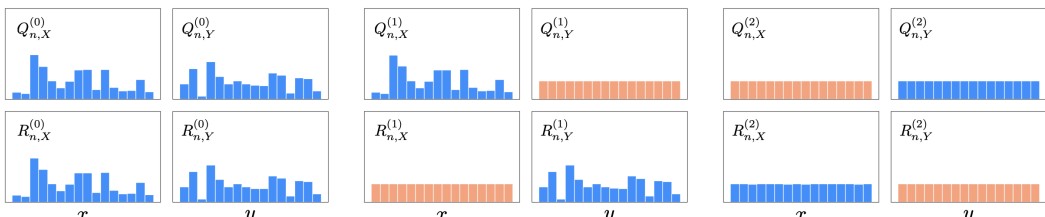

Figure 8: **Empirical Marginals of CLIP Contrast Matrix.** Depiction of the probability measures $Q_n^{(k)}$ and $R_n^{(k)}$ as described in (82) from Sec. 2. The orange bars correspond to the observed marginal after fitting to the target uniform distribution on the given iteration. **Left:** $Q_n^{(0)}$ and $R_n^{(0)}$, where neither marginal is set to uniform. **Center:** $Q_n^{(1)}$ and $R_n^{(1)}$, which corresponds to the original CLIP loss. **Right:** $Q_n^{(2)}$ and $R_n^{(2)}$, which correspond to two iterations of the balancing procedure within the loss. The blue bars are slightly non-uniform.

parameter $\theta_I$ and solve the regularized multinomial cross entropy (MCE) objective

$$\min_{W \in \mathbb{R}^{C \times r}} \left[ L_{\mathrm{MCE}}(W) := -\frac{1}{N} \sum_{i=1}^{N} [\mathrm{LogSoftmax}(W f_{\theta_I}(x_i))]_{c_i} + \frac{\lambda}{2} \|W\|_F^2 \right],$$

where $\lambda > 0$ is a regularization parameter, $\|\cdot\|_F$ denotes the Frobenius norm on $\mathbb{R}^{C \times r}$ and $\mathrm{LogSoftmax} : \mathbb{R}^C \to \mathbb{R}^C$ is given by $\mathrm{LogSoftmax}(z) = z - \log \sum_{j=1}^{C} \exp(z_j)$. This results in a classifier

$$g(x) := \arg\max_{j \in [C]} [W f_{\theta_I}(x)]_j,$$

which can then be evaluated using standard accuracy metrics on a held-out test set. The image classification results for the Rendered SST2 [Radford et al., 2021], VOC2007 [Everingham et al., 2007], and FGVC Aircraft [Maji et al., 2013] benchmarks are given in Fig. 7.

**Empirical Marginals in CLIP Balancing.** To further clarify how the iterative balancing procedure is baked into the CLIP losses, recall from (82) that the objectives decompose into two terms, which depend on $Q_n^{(k)}$ and $R_n^{(k)}$ which differ only based on whether balancing to fit $P_Y$ or to fit $P_X$ is applied first, respectively. Thus, for any model parameterized by $\theta$ and any number of iterations $k$, there are four marginal distributions of interest: $Q_{\theta,X}^{(k)}, Q_{\theta,Y}^{(k)}, R_{\theta,X}^{(k)}$, and $R_{\theta,Y}^{(k)}$. Based on the order of iterations, we have that $Q_{\theta,Y}^{(1)} = R_{\theta,Y}^{(2)} = P_Y$, and $R_{\theta,X}^{(1)} = Q_{\theta,X}^{(2)} = P_X$. This is illustrated in Fig. 8. We see that after only a few iterations, both marginal distributions converge to the uniform distribution.

