# OpenReview forum: "The Benefits of Balance: From Information Projections to Variance Reduction"
_NeurIPS.cc/2024/Conference — NeurIPS 2024 poster_

### Official Review · Reviewer_vTkj · 2024-07-11

**Soundness:** 3
**Presentation:** 3
**Contribution:** 3
**Rating:** 6
**Confidence:** 2

**Summary:**

This paper introduces a technique called iterative data balancing—altering data distributions to match predefined marginal distributions—that can lead to variance reduction in model predictions. The authors highlight its utility for self-supervised learning, which has been used to train several foundation models. The results demonstrate that iterative rebalancing of data leads to improvements in zero-shot learning performance and a reduction in variance among the empirical marginals with more than one iteration (k>1) of their technique.

**Strengths:**

The paper has theoretical contributions that include the derivation of non-asymptotic bounds that quantify the variance reduction achieved through their data balancing technique. The authors also present empirical studies that demonstrate the effectiveness of their proposed balancing technique. The authors discuss the utility of data balancing across different tasks, such as image-caption pair matching and self-supervised clustering, identifying the utility of their approach. Their approach has the potential for adoption in various domains, including in the training of foundation models.

**Weaknesses:**

The authors could expand the range of experiments to include a more diverse set of tasks, which in turn could enhance the generalization of the findings. Furthermore, their iterative data balancing technique relies heavily on predefined (uniform) target marginal distributions (see questions about this in next section). Finally, the iterative nature of the proposed data balancing technique may introduce significant computational demands. The paper could benefit by a more comprehensive overview of how the iterative technique computational overhead is impacted by very large datasets and/or models.

**Questions:**

1. In your work, your target marginals were uniform; how would your method respond to non-uniform marginals?
2. Your target marginals were accurately specified (as uniform). What if the target marginals of the two distributions were less accurately specified (i.e., the true underlying distributions are not well-known)? How do you think that this would influence the empirical results of your technique (e.g., zero-shot average per-class recall)?
3. Are there existing methods for variance reduction and data balancing? If so, why did you not include an empirical comparison to existing methods?
4. You mention that the zero-shot evaluation metrics are difficult to produce intervals for (i.e., you are missing error bars). Why is this the case?

**Limitations:**

I don't feel the authors addressed the limitations of their work significantly, aside from noting that their methods were different as those "in practice". As already mentioned in the questions section, how robust are your results to imbalance and uncertainty in the target marginals? Furthermore, are there situations where your technique fails to improve the empirical results? If so, what are they?

---

> ### Author Rebuttal · Authors · 2024-08-06
>
> Thank you for your insightful comments. We address them below.
>
> >**The authors could expand the range of experiments to include a more diverse set of tasks.**
>
> Thank you for raising this point. We also show performance on an image retrieval task in Figure 8 of the attached PDF using the Pascal VOC benchmark. We show mean average precision (MAP) across 30 queries in a 17k image database. Qualitatively similar findings hold, in that the additional iteration of balancing (over the original CLIP objective) yields performance improvements and warrants further investigation.
>
> >**Data balancing technique relies heavily on predefined (uniform) target marginal distributions...**
>
> We first emphasize that neither the theoretical results nor the method described require the marginals to be uniform. The matrix scaling/Sinkhorn algorithm (as balancing is called in other contexts) has been used across domains for arbitrary marginals. The choice of the uniform marginals in the experiments was motivated by the fact that leading SSL methods already in use currently can be shown to be exactly, mathematically equivalent to balancing with uniform marginals.
>
> However, as far as inaccurately specified marginal distributions, it is an interesting question what the balanced measure converges to ($n \rightarrow \infty$) when $(P_X, P_Y)$ are not the marginals of the true probability measure $P$. This is outside the scope of this work but is a natural follow-up investigation.
>
> >**Finally, the iterative nature of the proposed data balancing technique may introduce significant computational demands.**
>
> Thank you for identifying this point. The balancing technique is not a proposal of this paper; it is already in use in large foundation models such as CLIP and DINO, as described in Section 2. In particular, the approach applies to minibatches, and simply reduces to row scaling and column scaling of a matrix. The batch sizes for training the largest CLIP models are on the order of $10^4$, so taking row and column sums of this matrix is usually not the computational bottleneck of these training workloads.
>
> >**Are there existing methods for variance reduction and data balancing?**
>
> Thank you for raising this important point. Another approach for variance reduction through marginal fitting used in stochastic simulation and one-dimensional optimal transport is to use the [quantile transform](https://scikit-learn.org/stable/modules/generated/sklearn.preprocessing.quantile_transform.html). That is, assuming that $X_1, \ldots, X_n$ are real-valued, we may replace them by $F_n(X_1), \ldots, F_n(X_n)$, where $F_n$ is the empirical CDF (in other words, $F_n(X_i)$ is the rank of $X_i$ divided by $n$). These random variables are uniformly distributed. By applying $F_X^{-1}$, the resulting (discrete) inform variables are now approximately distributed according to $F_X$. This is an approximation of the inverse CDF sampling technique, as $F_X^{-1}(U)$ for a continuous uniform random variable $U$ will have CDF $F_X$. We call this the “Inverse Probability (IP) Weighting” method and compare it to the empirical distribution and the balanced distribution in a simulation (see Figure 9 of the attached PDF). Across sample sizes, balancing improves on mean squared estimation error. While the values near $s=0$ imply that there is “more information” in the marginals, the inherent variability of the distribution also increases, explaining how various methods do not achieve perfect variance reduction.
>
> >**You mention that the zero-shot evaluation metrics are difficult to produce intervals for. Why is this the case?**
>
> Thank you for identifying this point of improvement. We meant that incorporating error estimates based on bootstrapping the datasets themselves would be computationally infeasible, as each point on the curve requires the evaluation of a foundation model to be used in the CLIP Benchmark suite. However, we have incorporated multiple seeds to account for uncertainty across training runs. Please see both Figure 6 and Figure 8 of the attached PDF to see that qualitatively similar trends hold for the average performance and performance across individual seeds.
>
> >**Are there situations where your technique fails to improve the empirical results? If so, what are they?**
>
> As mentioned in the response to the Reviewer os3n, the learned representations based on more than 2 iterations of rebalancing (Figure 6 of the attached PDF) show that performance in zero-shot tasks may no longer improve after 2 iterations. Similarly, as seen in the simulation in Figure 9, other methods for balancing may perform as well or better when the random variables in $X$ and $Y$ are nearly independent. Please see the "Example" in the response to Reviewer os3n for an intuitive explanation of this simulation. We emphasize that the experiments are meant to be illustrative of the theory established in Sections 2 and 3, and we do not make claims of state-of-the-art empirical methods.

---

> > ### Comment · Reviewer_vTkj · 2024-08-11
> > **Rebuttal Read**
> >
> > Thank you for taking the time to provide a detailed rebuttal. You have clarified the points I inquired about.

---

### Official Review · Reviewer_os3n · 2024-07-12

**Soundness:** 3
**Presentation:** 2
**Contribution:** 3
**Rating:** 5
**Confidence:** 3

**Summary:**

This paper explores the use of data balancing in various self-supervised learning (SSL) frameworks. The authors argue that this iterative algorithm, which is typically used to avoid representation collapse in SSL models, also provides a benefit of reducing the variance of empirical functionals of the distribution over data sources. The paper establishes non-asymptotic bounds quantifying this variance reduction and relates them to the eigendecays of specific Markov operators.

**Strengths:**

1. the paper provides a new perspective on the benefits of data balancing
2. provide different examples of data balancing in practice and prove a non-asymptotic bound on the MSE of balanced estimators.
3. The findings may have implications for improving SSL models

**Weaknesses:**

1.the experiments are somewhat limited in scope
2.adding more visualizations or intuitive explanations may be better for understanding the key finding of the paper.
3.will the assumptions limit the applicability of the findings?

**Questions:**

1. Can the findings be extended to other areas?
2. Can the authors provide more details on how the assumptions, such as the spectral gap condition, hold or when will they not hold in practical scenarios? Are there specific types of data or models where these assumptions are more likely to be satisfied?
3. For Figure 2, Can the authors provide experiment results with more iterations?
4. Are there any other variance reduction techniques and how does data balancing compare to those techniques?

**Limitations:**

In checklist the author mentions that the work is primarily theoretical and has no societal impact. but it will be better to discuss the positive social impact. The author also mentions the limitation that the setting studied has some dissimilarities with practice but haven't addressed the limitations yet.

---

> ### Author Rebuttal · Authors · 2024-08-06
>
> Thank you for your helpful comments and suggestions. We address them below.
>
> The upcoming comments concern the interpretation of the spectral gap condition (and the second largest singular value $s_2$). To facilitate this discussion, we introduce a simple example by starting with an arbitrary value of $s = s_2 < 1$ and construct a probability distribution $P$ that satisfies the condition in Eq (8) of the paper.
>
> **Example:** For $m=2$, we have that $\mathcal{X} = \\{x_1, x_2\\}$ and $\mathcal{Y} = \\{y_1, y_2\\}$, so every element in $h \in L^2(P)$ can be represented by four numbers $(h(x_1, y_1), h(x_2, y_1), h(x_1, y_2), h(y_1, y_2))$. In the case of uniform marginals, we can verify directly that Eq (8) can be satisfied by setting $\alpha_1 = \beta_1 = (1, 1, 1, 1)$, $\alpha_2 = (1, -1, 1, -1)$, $\beta_2 = (1, 1, -1, -1)$ and $P(x_1, y_1)  = P(x_2, y_2) = (1+s)/4$ and $P(x_1, y_2) = P(x_2, y_1) = (1-s)/4$ . Thus, as $s \rightarrow 1$, the distribution becomes “fully dependent” as $Y$ and $X$ are completely determined by one another. As $s \rightarrow 0$, the $X$ and $Y$ are independent. Intuitively, the marginals would contain the most information when $s = 0$, as the distribution can be computed directly from them. This idea can be generalized to construct similar distributions for $m >2$, which we use for the simulations in the attached PDF.
>
> >**Can the authors provide more details on how the assumptions, such as the spectral gap condition, hold or when will they not hold in practical scenarios? Are there specific types of data or models where these assumptions are more likely to be satisfied? Will the assumptions limit the applicability of the findings?**
>
> As mentioned in Assumption 2, it will hold if $P(x, y) > 0$ for any $(x, y)$ such that $P_X(x) > 0$ and $P_Y(y) > 0$ by the Perron-Frobenius theorem, but may hold otherwise. As seen above, the $s=1$ cases are often pathological, such as having perfect dependence between $X$ and $Y$. The positivity assumption is standard in even classical references on this subject; see [Bickel, Ritov, Wellner (1991)](https://projecteuclid.org/journals/annals-of-statistics/volume-19/issue-3/Efficient-Estimation-of-Linear-Functionals-of-a-Probability-Measure-P/10.1214/aos/1176348251.full) Eq (P3) for a equivalent condition. Thus, it is safe to assume the condition will hold in all practical scenarios of interest.
>
> >**…adding more visualizations or intuitive explanations may be better for understanding the key finding of the paper.**
>
> Thank you for raising this point. Returning to the example above, we also see that because $\alpha_1 = \beta_1$, the angle between the subspaces can be measured by the angle between $\alpha_2$ and $\beta_2$. By direct computation, we can see that $\langle \alpha_2, \beta_2 \rangle = s$, which means that the cosine of the angle $a$ between the subspaces is $s = \cos a$. Thus, the geometric interpretation of $s$ is the cosine of the angle between $L^2(P_X)$ and $L^2(P_Y)$. This can be visualized in Figure 7 of the attached PDF. In fact, this holds even for non-uniform marginals. With $P(x_1) = p_X$ and $P(y_1) = p_Y$, a slightly more tedious computation will show that Eq (8) is satisfied when $\alpha_1 = \beta_1 = (1, 1, 1, 1)$, $\alpha_2 = (\sqrt{(1-p_X)/p_X}, -\sqrt{p_X/(1-p_X)}, (\sqrt{(1-p_X)/p_X}, -\sqrt{p_X/(1-p_X)})$, $\beta_2 = (\sqrt{(1-p_Y)/p_Y}, \sqrt{(1-p_Y)/p_Y}, -\sqrt{p_Y/(1-p_Y)}, -\sqrt{p_Y/(1-p_Y)})$, and $P(x_1, y_1)  = p_Xp_Y + s \sqrt{p_X(1-p_X)p_Y(1-p_Y)}$. In this case, we still have that $\langle \alpha_2, \beta_2 \rangle = s$, so the non-uniform marginals do not warp this angle.
>
> >**Are there any other variance reduction techniques and how does data balancing compare to those techniques?**
>
> Another approach for variance reduction through marginal fitting used in stochastic simulation and one-dimensional optimal transport is to use the [quantile transform](https://scikit-learn.org/stable/modules/generated/sklearn.preprocessing.quantile_transform.html). That is, assuming that $X_1, \ldots, X_n$ are real-valued, we may replace them by $F_n(X_1), \ldots, F_n(X_n)$, where $F_n$ is the empirical CDF (in other words, $F_n(X_i)$ is the rank of $X_i$ divided by $n$). These random variables are uniformly distributed. By applying $F_X^{-1}$, the resulting (discrete) inform variables are now approximately distributed according to $F_X$. This is an approximation of the inverse CDF sampling technique, as $F_X^{-1}(U)$ for a continuous uniform random variable $U$ will have CDF $F_X$. We call this the “Inverse Probability (IP) Weighting” method and compare it to the empirical distribution and the balanced distribution in a simulation (see Figure 9 of the attached PDF). Across sample sizes, balancing improves on mean squared estimation error. While the values near $s=0$ imply that there is “more information” in the marginals, the inherent variability of the distribution also increases, explaining how various methods do not achieve perfect variance reduction.
>
> >**the experiments are somewhat limited in scope… Can the findings be extended to other areas?**
>
> Thank you for raising this point. We also show performance on an image retrieval task in Figure 8 of the attached PDF using the Pascal VOC benchmark. We show mean average precision (MAP) across 30 queries in a 17k image database. Qualitatively similar findings hold, in that the additional iteration of balancing (over the original CLIP objective) yields performance improvements and warrants further investigation.
>
> >**For Figure 2, Can the authors provide experiment results with more iterations?**
>
> We show this experiment in Figure 6 of the attached PDF. Both from observing the performance of the $k=5$ variant as well as directly observing the marginals in Figure 3 of the manuscript, we see that the marginals stabilize very quickly (after $k=2$ iterations in most cases), and performance remains similar beyond this point.

---

> > ### Comment · Reviewer_os3n · 2024-08-11
> >
> > I thank the authors for detailed responses and additional experiments. I've raised the score of contribution.

---

> > > ### Author Response · Authors · 2024-08-12
> > > **Response to os3n**
> > >
> > > We are happy to hear that the response to the review met your expectations. We are a bit puzzled as it appears on our end that the score is the same as before (5: Borderline accept). Does it appear to be the same for you? Thank you once again.

---

### Official Review · Reviewer_G8ZX · 2024-08-05

**Soundness:** 3
**Presentation:** 2
**Contribution:** 2
**Rating:** 5
**Confidence:** 3

**Summary:**

This work focusses on data balancing strategies in context of self-supervised learning. The main claim of the paper is that data balancing, commonly used to avoid representation collapse, has a variance reduction effect. The authors introduce an upper bound on the MSE of a balancing estimator, relating it to empirical risk minimisation. The main paper covers the key elements of the proofs, which is given in detail (and is extensive) in the appendix. Experiments are conducted to illustrate the impact of data balancing on examples described in the paper.

**Strengths:**

This paper attempts to shed light on SSL training and the role of data balancing. The paper formalises the problem and develops extensive theory. The main results is pretty cool and insightful in the sense that the upper bound on the MSE shows that data balancing has a variance reduction effect. The topic is of interest to the community and the work is focussing on a poorly understood paradigm that is becoming dominant.

**Weaknesses:**

I have three main concerns with this work:

1/ The theory is *very* extensive. The Appendix contains several pages of proofs that are difficult to parse and come on top of the formalism presented in the main paper. It seems like the main body could be simplified and made more to the point to convey the main gist of the contribution and make it more accessible.

2/ It is unclear how the data balancing examples in Section 2 map to the formalism introduced in Section 3. For example, what would (4) look like for example 1 and example 2?

3/ It is unclear what the experiments bring to the table and how they provide evidence to the main result. Making the link more explicit and explaining what are the key take aways from these results would help the reader.

**Questions:**

I have the following questions for the authors:

- Line 54: What do you mean by "X and Y are forms of the data that are related to, but distinct from, the form of Z"" given that Z is equal to (X,Y)?
- p4, example 1: What would the target marginals correspond to? What is \psi_n^(k) and P_n^(k) here?
- p4, example 2: What would the target marginals correspond to? What is \psi_n^(k) and P_n^(k) here?
- Why do we need \tilde{\psi}_n^{(k)} in (12) and how does it relate to \psi_n^{(k)}?
- How does (15) relate to the clip example introduced earlier in the paper and why is this a valid and sensible simplification to study?
- Does the main result have implications in practice in terms of design of algorithm?

**Limitations:**

The authors provided a brief discussion about future work at the end of the paper. The work presented here is theoretical in nature, attempting to provide new insights in existing approaches. There are not immediate implications in practice, but authors could discuss the scope of their work in more detail.

---

> ### Author Rebuttal · Authors · 2024-08-06
>
> Thank you for your comments and questions. We address them below.
>
> >**The theory is very extensive. The Appendix contains several pages of proofs that are difficult to parse and come on top of the formalism presented in the main paper.**
>
> We provide the complete, self-contained, proofs of all of our theoretical results, for reproducibility. The main analytical appendices (B, C, and D) are split by topic and include an outline for readability. As per the NeurIPS Paper Checklist, "For each theoretical result, does the paper provide the full set of assumptions and a complete (and correct) proof?... The proofs can either appear in the main paper or the supplemental material, but if they appear in the supplemental material, the authors are encouraged to provide a short proof sketch to provide intuition." This sketch is given between lines 235 and 259. We are happy to incorporate any suggestions to make them more clear.
>
> >**It is unclear how the data balancing examples in Section 2 map to the formalism introduced in Section 3. For example, what would (4) look like for example 1 and example 2?**
>
> The connection between the examples and the analysis in Section 3 is first introduced in the introduction, between lines 59 and 81. The main notational difference is that quantities are often indexed by $\theta$ in Section 2 and 4 as they describe objectives that are evaluated at a specific parameter value. In Section 3, we drop this index as we are considering the statistical properties of balancing estimators (i.e. the loss that is actually optimized in an SSL scenario) at a fixed parameter value. Thus, the analogous value to $\psi\_n^{(k)}$ in (4) is always $\sum_{x, y} h_\theta(x, y) R_\theta^{(k)}(x, y)$, which is another way of writing (2).
>
> >**How does (15) relate to the clip example introduced earlier in the paper and why is this a valid and sensible simplification to study? Does the main result have implications in practice in terms of design of algorithm?**
>
> The objective in Eq (15) is not a simplification but a generalization of the CLIP objective. It exactly reduces to the CLIP objective when $k = 1$. As seen in experiments, zero-shot accuracy of learned representations can improve when considering more iterations (see Figure 2 of the manuscript and Figure 6 of the attached PDF). The main result is meant to provide theoretical insight into the reasons behind the success of SSL training procedures that are widely popular, but not well-understood (as outlined between lines 23 and 48).
>
> >**What do you mean by "$X$ and $Y$ are forms of the data that are related to, but distinct from, the form of $Z$" given that $Z$ is equal to $(X,Y)$?**
>
> $Z$ is not necessarily equal to $(X, Y)$, except in empirical risk minimization (line 53). We use the examples from Section 2; in the case of self-labeling, $Z$ represents an image, $X=Z$, and $Y$ is a learnable cluster representation (Example 1). For contrastive learning, similarly to empirical risk minimization, we have $Z = (X, Y)$ where $X$ is an image and $Y$ is a caption (Example 2). In Appendix E.4, we have yet another example of metadata curation, in which $Z$ is an image-caption pair, $X = Z$, and $Y$ is an associated keyword.
>
> >**example 1: What would the target marginals correspond to? What is $\psi_n^{(k)}$ and $P_n^{(k)}$ here?**
>
> The target marginals are discrete uniform measures on $\mathcal{X}$ and $\mathcal{Y}$ (line 148). $P_n^{(k)}$ would be analogous to $R_\theta^{(k)}$ (line 132), and $\psi_n^{(k)}$ is (throughout) analogous to (2). We say analogous because the starting measure $R_\theta^{(0)}$ comes from the output of a model parametrized by $\theta$, whereas $P_n^{(0)} = P_n$ is used in Section 3 to represent the empirical measure of randomly drawn data.
>
> >**example 2: What would the target marginals correspond to? What is $\psi_n^{(k)}$ and $P_n^{(k)}$ here?**
>
> The target marginals are discrete uniform measures on $\mathcal{X}$ and $\mathcal{Y}$ (line 167). The other quantities have the same interpretation as in the previous example and others.
>
> >**Why do we need $\tilde{\psi}_n^{(k)}$ in (12) and how does it relate to $\psi_n^{(k)}$?**
>
> Because balancing iterations are only well-defined under the event $\mathcal{S}$ (line 192), $\tilde{\psi}_n^{(k)}$ is defined simply to handle the technical consideration of $\mathcal{S}$ not being satisfied.

---

> > ### Comment · Reviewer_G8ZX · 2024-08-14
> > **Re: Rebuttals by authors**
> >
> > I thank the authors for their response. My concern about the clarity of the material presented remains and I would have liked the authors to discuss the practical implications of their results.
> >
> > However, I agree that, overall, the theory produced is non-trivial and appreciated the additional experiments conducted, which I found informative. I have raised my score as a result.

---

### Author Rebuttal · Authors · 2024-08-06

We thank the reviewers for their hard work reviewing our paper and providing concrete comments! We collect the broad points made below and address other reviewer concerns in the individual responses. To summarize, our paper provides three theoretical innovations:
1. The first quantitative and non-asymptotic analysis of the statistical effect of data balancing (a.k.a. matrix scaling or biproportional fitting), which spans self-supervised learning (SSL), optimal transport, and statistics.
2. A novel interpretation of contrastive objectives relating them to data balancing with subsequent algorithmic implications.
3. A mathematical recursion formula for balanced probability measures that can be used as an independent technical tool.

The main contributions of the paper are of a theoretical nature, for which we provide complete proofs in the Appendix sections, and we also provide numerical experiments as illustrations. While the paper should be assessed as such, we have addressed all feedback from the reviewers.

**The main concerns across reviews were the following.**

**Additional experimentation:** The reviewers requested additional experiments, including performance on tasks other than zero-shot image classification, comparison to other balancing approaches, and extensions of the balanced-CLIP variant for more than two iterations. These experiments are described in the responses and the associated results can be found in the attached PDF.

**Intuition behind theoretical assumptions:** The reviewers also requested clarity on theoretical assumptions, such as the spectral gap condition or whether uniform marginals are needed. We address these with both intuitive explanations and companion simulation experiments.

Please see the individual responses for more details. Thank you!

---

### Author Response · Authors · 2024-08-09
**Thank you for your feedback / Addressing additional concerns**

Dear reviewers,

As the discussion period draws to a close on August 13 (2 business days from now) we kindly request that you take a moment to review and acknowledge our responses to your questions and comments.

In response to the comments raised by Reviewers os3n and vTkj, we have added additional experiments including other tasks and other data balancing baselines. We have also addressed questions from Reviewers G8ZX, os3n, and vTkj about theoretical assumptions and the link between the formal statements in Section 3 and the examples in Sections 2 and 4. Please see the responses for details.

If any further concerns or questions arise that we could address, please do not hesitate to reach out. We appreciate your time and feedback.

Sincerely,

The authors

---

### Decision · Program_Chairs · 2024-09-25

**Decision:**

Accept (poster)

**Comment:**

This work studies data balancing in SSL and show that avoiding representation collapse also leads to a variance reduction. The paper establishes non-asymptotic bounds quantifying this variance reduction. The authors addressed the concerns raised by the reviewers and produced the additional experiments they requested. I found those additional results helpful; authors should incorporate the additional material in the final paper or Appendices. All reviewers voted for acceptance.